# FLOW ANNEALED IMPORTANCE SAMPLING BOOTSTRAP

**Laurence I. Midgley**[*]
University of Cambridge
InstaDeep
laurencemidgley@gmail.com

**Vincent Stimper**[*]
Max Planck Institute for Intelligent Systems
University of Cambridge
vs488@cam.ac.uk

**Gregor N. C. Simm**
University of Cambridge
gncs2@cam.ac.uk

**Bernhard Schölkopf**
Max Planck Institute
for Intelligent Systems
bs@tue.mpg.de

**José Miguel Hernández-Lobato**
University of Cambridge
jmh233@cam.ac.uk

## ABSTRACT

Normalizing flows are tractable density models that can approximate complicated target distributions, e.g. Boltzmann distributions of physical systems. However, current methods for training flows either suffer from mode-seeking behavior, use samples from the target generated beforehand by expensive MCMC methods, or use stochastic losses that have high variance. To avoid these problems, we augment flows with annealed importance sampling (AIS) and minimize the mass-covering $\alpha$-divergence with $\alpha = 2$, which minimizes importance weight variance. Our method, Flow AIS Bootstrap (FAB), uses AIS to generate samples in regions where the flow is a poor approximation of the target, facilitating the discovery of new modes. We apply FAB to multimodal targets and show that we can approximate them very accurately where previous methods fail. To the best of our knowledge, we are the first to learn the Boltzmann distribution of the alanine dipeptide molecule using only the unnormalized target density, without access to samples generated via Molecular Dynamics (MD) simulations: FAB produces better results than training via maximum likelihood on MD samples while using 100 times fewer target evaluations. After reweighting the samples, we obtain unbiased histograms of dihedral angles that are almost identical to the ground truth.

## 1 INTRODUCTION

Approximating intractable distributions is a challenging task whose solution has relevance in many real-world applications. A prominent example involves approximating the Boltzmann distribution of a given molecule. In this case, the unnormalized density can be obtained by physical modeling and is given by $e^{-u(\boldsymbol{x})}$, where $\boldsymbol{x}$ are the 3D atomic coordinates and $u(\cdot)$ returns the dimensionless energy of the system. Drawing independent samples from this distribution is difficult (Lelièvre et al., 2010). It is typically done by running expensive Molecular Dynamics (MD) simulations (Leimkuhler & Matthews, 2015), which yield highly correlated samples and require long simulation times.

An alternative is given by normalizing flows. These are tractable density models parameterized by neural networks. They can generate a batch of independent samples with a single forward pass and any bias in the samples can be eliminated by reweighting via importance sampling. Flows are called Boltzmann generators when they approximate Boltzmann distributions (Noé et al., 2019). Recently, there has been a growing interest in these methods (Dibak et al., 2022; Köhler et al., 2021; Liu et al., 2022) as they have the potential to avoid the limitations of MD simulations. Most current approaches to train Boltzmann generators rely on MD samples since these are required for the estimation of the flow parameters by maximum likelihood (ML) (Wu et al., 2020). Alternatively, flows can be trained without MD samples by minimizing the Kullback–Leibler (KL) divergence with respect to the target distribution. Wirnsberger et al. (2022) followed this approach to approximate the

---

[*]Equal contribution

Boltzmann distribution of atomic solids with up to 512 atoms. However, the KL divergence suffers from mode-seeking behavior, which severely deteriorates the performance of this approach with multimodal target distributions (Stimper et al., 2022). On the other hand, mass-covering objective such as the forward KL divergence suffer from the high variance of the samples from the flow.

To address these challenges, we present a new method for training flows: **F**low **A**IS **B**ootstrap[1] (**FAB**). Our main contributions are as follows:

1. We propose to use the $\alpha$-divergence with $\alpha = 2$ as our training objective, which is mass-covering and minimizes importance weight variance. At test time an importance sampling distribution with low $\alpha$-divergence (with $\alpha = 2$) may be used to approximate expectations with respect to the target with low variance. This objective is challenging to estimate during training. To approximate this objective we use annealed importance sampling (AIS) with the flow as the initial distribution and the target set to the minimum variance distribution for the estimation of the $\alpha$-divergence. AIS returns samples that provide a higher quality training signal than samples from the flow, as it focuses on the regions that contribute the most to the $\alpha$-divergence loss.

2. We reduce the computational cost of our method by introducing a scheme to re-use samples via a prioritized replay buffer.

3. We apply FAB to a toy 2D Gaussian mixture distribution, the 32 dimensional "Many Well" problem, and the Boltzmann distribution of alanine dipeptide. In these experiments, we outperform competing approaches and, to the best of our knowledge, we are the first to successfully train a Boltzmann generator on alanine dipeptide using only the unnormalized target density. In particular, we use over 100 times fewer target evaluations than a Boltzmann generator trained with MD samples while producing a better approximation to the target.

## 2 BACKGROUND

**Normalizing flows**     Given a random variable $\mathbf{z}$ with distribution $q(\mathbf{z})$, a normalizing flow (Tabak & Vanden-Eijnden, 2010; Rezende & Mohamed, 2015; Papamakarios et al., 2021) uses an invertible map $F : \mathbb{R}^d \rightarrow \mathbb{R}^d$ to transform $\mathbf{z}$ yielding the random variable $\mathbf{x} = F(\mathbf{z})$ with distribution

$$q(\mathbf{x}) = q(\mathbf{z}) \left| \det(J_F(\mathbf{z})) \right|^{-1} , \tag{1}$$

where $J_F(\mathbf{z}) = \partial F / \partial \mathbf{z}$ is the Jacobian of $F$. If we parameterize $F$, we can use the resulting model to approximate a target distribution $p$. To simplify our notation, we will assume the target density $p(\mathbf{x})$ is normalized, i.e., it integrates to 1, but the methods described here are equally applicable when this is not the case. If samples from the target distribution are available, the flow can be trained via ML. If only the target density $p(\mathbf{x})$ is given, the flow can then be trained by minimizing the reverse KL divergence[2] between $q$ and $p$, i.e., $\mathrm{KL}(q\|p) = \int_x q(\mathbf{x}) \log\{q(\mathbf{x})/p(\mathbf{x})\}\mathrm{d}\mathbf{x}$, which is estimated via Monte Carlo using samples from $q$.

**Alpha divergence**     An alternative to the KL divergence is the $\alpha$-divergence (Zhu & Rohwer, 1995; Minka, 2005; Müller et al., 2019; Bauer & Mnih, 2021; Campbell et al., 2021) defined by

$$D_\alpha(p\|q) = -\frac{\int_x p(\mathbf{x})^\alpha q(\mathbf{x})^{1-\alpha}\mathrm{d}\mathbf{x}}{\alpha(1-\alpha)} . \tag{2}$$

The $\alpha$-divergence is mode-seeking for $\alpha \le 0$ and mass-covering for $\alpha \ge 1$ (Minka, 2005), as shown in Figure 1. When $\alpha = 2$, minimizing the $\alpha$-divergence is equivalent to minimizing the variance of the importance sampling weights $w_{\mathrm{IS}}(\mathbf{x}) = p(\mathbf{x})/q(\mathbf{x})$, which is desirable if importance sampling will be used to eliminate bias in the samples from $q$ at test time.

**Annealed importance sampling**     AIS begins by sampling from an initial distribution $\mathbf{x}_1 \sim p_0 = q$, given by the flow in our case, and then transitioning via MCMC through a sequence of intermediate distributions, $p_1$ to $p_{M-1}$, to produce a sample $\mathbf{x}_M$ closer to the target distribution $g = p_M$ (Neal, 2001). Each transition generates an intermediate sample $\mathbf{x}_j$ by running a few steps of a Markov chain initialized with the previous intermediate sample $\mathbf{x}_{j-1}$ that leaves the intermediate

---

[1]FAB uses the flow in combination with AIS to estimate a loss in order to improve the flow. Thus we use *bootstrap* in the name of our method to mean "using one's existing resources to improve oneself".

[2]We refer to reverse KL divergence as just "KL divergence", following standard practice in literature.

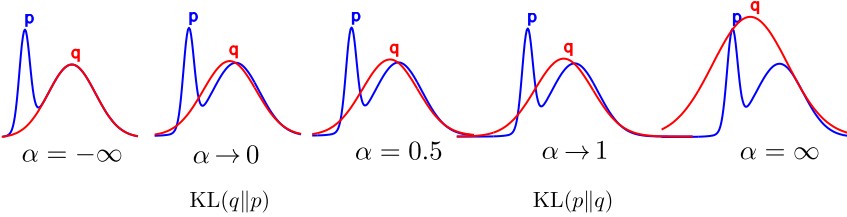

Figure 1: Illustration of unnormalized Gaussian approximating distributions $q$, shown in red, that minimize the $\alpha$-divergence for different values of $\alpha$ with respect to a bimodal target distribution $p$, shown in blue. The solutions $q$ are mode-seeking for values of $\alpha \leq 0$ and they are mass-covering for values of $\alpha \geq 1$. The cases $\alpha \to 0$ and $\alpha \to 1$ correspond to $\text{KL}(q\|p)$ and $\text{KL}(q\|p)$, respectively. Figure reproduced from (Minka, 2005).

distribution $p_{j-1}$ invariant. Each $p_j$ is defined by interpolating between the initial and target log densities: $\log p_i(\mathbf{x}) = \beta_i \log p_0(\mathbf{x}) + (1 - \beta_i) \log p_M(\mathbf{x})$, where $1 = \beta_0 > \beta_1 > ... > \beta_N = 0$. AIS provides an importance weight for the final resulting sample $\mathbf{x}_M$ given by

$$w_{\text{AIS}}(\mathbf{x}_M) = \frac{p_1(\mathbf{x}_1)}{p_0(\mathbf{x}_1)} \frac{p_2(\mathbf{x}_2)}{p_1(\mathbf{x}_2)} \cdots \frac{p_{M-1}(\mathbf{x}_{M-1})}{p_{M-2}(\mathbf{x}_{M-1})} \frac{p_M(\mathbf{x}_M)}{p_{M-1}(\mathbf{x}_M)}. \tag{3}$$

These weights exhibit variance reduction compared to their importance sampling counterparts $w_{\text{IS}}(\mathbf{x}) = p(\mathbf{x})/q(\mathbf{x})$ (Neal, 2001). The AIS samples and importance weights may then be used to estimate expectations over the target $g(\mathbf{x})$ using $\text{E}_{g(\mathbf{x})}[h(\mathbf{x})] = \text{E}_{\text{AIS}}[w_{\text{AIS}}(\mathbf{x})h(\mathbf{x})]$, where $h(\mathbf{x})$ is some function of interest. Hamiltonian Monte Carlo (HMC) is a suitable transition operator for implementing AIS in challenging problems (Neal, 1995; Sohl-Dickstein & Culpepper, 2012).

## 3 METHOD

### 3.1 FLOW ANNEALED IMPORTANCE SAMPLING BOOTSTRAP

FAB trains a flow $q$ to approximate a target $p$ by minimizing $D_{\alpha=2}(p\|q)$, which is estimated with AIS using $q$ as initial distribution and $p^2/q$ as target. The latter is the minimum variance importance sampling distribution for estimating the $D_{\alpha=2}(p\|q)$ loss. FAB performs a form of bootstrapping since it fits the flow $q$ using the samples generated by $q$ after these have been improved with AIS to fit $p^2/q$. Thereby, we train a mass-covering flow without access to samples from the target. Below we provide a brief derivation of our loss function and refer to Appendix B for the full derivation.

We consider $q$ to be specified by some parameters $\theta$ and write $q_\theta$ to make this explicit. We aim to tune $q_\theta$ by minimizing the loss function $\mathcal{L}(\theta) \propto D_{\alpha=2}(p\|q_\theta)$ where $\mathcal{L}(\theta)$. We can write our loss as an expectation over some distribution $g(\mathbf{x})$ by using importance sampling:

$$D_{\alpha=2}(p\|q_\theta) \propto \mathcal{L}(\theta) = \int \frac{p(\mathbf{x})^2}{q_\theta(\mathbf{x})} \, \mathrm{d}\mathbf{x} = \text{E}_{g(\mathbf{x})}\left[ \frac{p(\mathbf{x})^2}{q_\theta(\mathbf{x})g(\mathbf{x})} \right]. \tag{4}$$

We consider setting $g \propto p^2/q_\theta$ which minimizes[3] the variance in the estimation of $\mathcal{L}(\theta)$. Sampling directly from $g \propto p^2/q_\theta$ is intractable. Instead, we train the flow using an estimate of the loss based on samples generated by AIS when targeting $g \propto p^2/q_\theta$ and using $q_\theta$ as the initial distribution. These AIS samples have higher quality than those returned by the flow as they occur in regions where the integrand in Equation (4) takes high values. These are regions where $p$ and $q$ have high and low density, respectively. Another advantage of the AIS samples is that we can use the weights returned by AIS to obtain an unbiased estimate of $\mathcal{L}(\theta)$.

To obtain the gradient of Equation (4) with respect to $\theta$, let us denote $f_\theta(\mathbf{x}) = p(\mathbf{x})^2/q_\theta(\mathbf{x})$. Without loss of generality[4] we assume $\int f_\theta(\mathbf{x})\mathrm{d}\mathbf{x} = 1$ and then set $g(\mathbf{x}) = f_\theta(\mathbf{x})$. First, we write the gradient

---

[3]The importance sampling distribution $g$ that minimizes the variance in the estimation of $\mu = \int |f(\mathbf{x})|p(\mathbf{x})\mathrm{d}\mathbf{x}$ is given by $g(\mathbf{x}) \propto |f(\mathbf{x})|p(\mathbf{x})$ (Kahn & Marshall, 1953; Owen, 2013). We note this is different from the distribution that minimizes variance for self-normalized importance sampling, which is given by $g(\mathbf{x}) \propto |f(\mathbf{x}) - \mu|p(\mathbf{x})$ (Hesterberg, 1988; Owen, 2013).

[4]See Appendix B for the full derivation where we keep track of the normalizing constant.

as an expectation over $g$:

$$\nabla_\theta \mathcal{L}(\theta) = \mathrm{E}_{g(\mathbf{x})}\left[\frac{\nabla_\theta f_\theta(\mathbf{x})}{f_\theta(\mathbf{x})}\right] = \mathrm{E}_{g(\mathbf{x})}\left[\nabla_\theta \log f_\theta(\mathbf{x})\right] = -\mathrm{E}_{g(\mathbf{x})}\left[\nabla_\theta \log q_\theta(\mathbf{x})\right]. \tag{5}$$

We can then write this as an expectation over the AIS forward pass:

$$\nabla_\theta \mathcal{L}(\theta) = -\mathrm{E}_{\mathrm{AIS}}\left[w_{\mathrm{AIS}}\nabla_\theta \log q_\theta(\bar{\mathbf{x}}_{\mathrm{AIS}})\right], \tag{6}$$

where $\bar{\mathbf{x}}_{\mathrm{AIS}}$ and $w_{\mathrm{AIS}}$ are the samples and respective importance weights generated by AIS when targeting $g$. The *bar* superscript denotes stopped gradients in the AIS samples, $\bar{\mathbf{x}}_{\mathrm{AIS}}$, with respect to $\theta$. If we stop the gradients of $w_{\mathrm{AIS}}$ as well, we can then use the surrogate loss function $\mathcal{S}(\theta) = -\mathrm{E}_{\mathrm{AIS}}\left[\bar{w}_{\mathrm{AIS}}\log q_\theta(\bar{\mathbf{x}}_{\mathrm{AIS}})\right]$, which can be estimated by Monte Carlo and differentiated to obtain unbiased estimates of the gradient. In practice, we found that using the self-normalized importance weights greatly improved training stability. We refer to the surrogate loss with self-normalized importance weights as $\mathcal{S}'(\theta)$. Its estimate used for training is given by:

$$\mathcal{S}'(\theta) \approx -\sum_i^N \frac{\bar{w}_{\mathrm{AIS}}^{(i)}}{\sum_i^N \bar{w}_{\mathrm{AIS}}^{(i)}}\log q_\theta(\bar{\mathbf{x}}_{\mathrm{AIS}}^{(i)}), \tag{7}$$

where $\bar{w}_{\mathrm{AIS}}^{(i)}$ and $\bar{\mathbf{x}}_{\mathrm{AIS}}^{(i)}$ are $N$ samples and weights generated by AIS using $g = p^2/q_\theta$ as target distribution. When evaluating the gradient of Equation (7) with respect to $\theta$, gradients must be stopped during the computation of the AIS samples and weights. In practice, we obtain good performance with a relatively low number of intermediate AIS distributions, e.g., 1 for the Gaussian mixture model problem and 8 for the dipeptide problem, see the following section. Moreover, we can use AIS after training with target $p$ to further reduce variance when approximating expectations over $p$.

Here we have focused on the minimization of $D_{\alpha=2}(p\|q)$ using an AIS bootstrapping approach with $p^2/q$ as target distribution. However, our approach is general and could be used to minimize other objectives (Midgley et al., 2021) and to train other models, such as those that combine flows with stochastic sampling steps (Wu et al., 2020; Arbel et al., 2021; Matthews et al., 2022; Jing et al., 2022). We provide further discussion and examples related to this in Appendix C. This includes a derivation of a version of FAB that works for $\alpha$ divergence minimization with arbitrary values of $\alpha$.

In Appendix D we provide an analysis of the quality of the estimates of the gradient of $D_{\alpha=2}(p\|q)$ produced by FAB and by importance sampling with samples from $q$ or $p$. We focus on the FAB gradient in the form from Equation (6), as it is easy to analyze. First, we show that in a simple scenario where both $q$ and $p$ are 1D Gaussians, the signal-to-noise ratio of FAB with a small number of AIS distributions is far superior to that of estimating $D_{\alpha=2}(p\|q)$ using samples from $q$ or $p$.

We also study in Appendix D the performance of FAB as the dimensionality of the problem grows. Similar to the analysis of AIS with increasing dimensionality by Neal (2001), we consider a simple scenario where $p$ and $q$ are factorized and the AIS MCMC transitions are perfect (output independent samples that follow the corresponding intermediate distributions). We then show the following: 1) Estimating $D_{\alpha=2}$ with importance sampling using samples from $q$ or $p$ results in a variance of the gradient estimate that grows exponentially with respect to the dimensionality of the problem. 2) This variance remains constant in FAB when the number of AIS distributions increases by the same factor as the dimensionality. We provide an empirical analysis of the gradient variance in FAB under the aforementioned assumptions. If we increase the number of AIS distributions by the same factor by which the dimensionality increases, the SNR of the gradient estimate remains roughly constant. This suggests that FAB should scale well to higher dimensional problems, relative to training via estimation of the $D_{\alpha=2}$ loss by importance sampling with samples from $q$ or $p$. We acknowledge that our simplifying assumptions are strong and we leave a more general analysis to future work.

## 3.2 RE-USING SAMPLES THROUGH A REPLAY BUFFER

Although AIS is relatively cheap, it is still significantly more expensive than directly sampling from the flow as it requires additional flow and target evaluations. To speed up computations, we re-use AIS samples during the flow updates by making use of a *prioritized replay buffer* analogous to the one in (Mnih et al., 2015; Schaul et al., 2016).

Consider a replay buffer with a set of samples and corresponding AIS weights generated during a single run of AIS with target $g(\mathbf{x}) = p(\mathbf{x})^2/q_\theta(\mathbf{x})$, where $q_\theta(\mathbf{x})$ is the flow at a point in training

---

**Algorithm 1:** FAB for the minimization of $D_{\alpha=2}(p\|q)$ with a prioritized replay buffer

---

Initialize flow $q$ parameterized by $\theta$
Initialize replay buffer to a fixed maximum size
**for** *iteration = 1 to K* **do** `// Generate AIS samples and add to buffer`
  Sample $\mathbf{x}_q^{(1:M)}$ from $q_\theta$ and evaluate $\log q_\theta(\mathbf{x}_q^{(1:M)})$
  Obtain $\mathbf{x}_{\text{AIS}}^{(1:M)}$ and $\log w_{\text{AIS}}^{(1:M)}$ using AIS with target $p^2/q_\theta$ and seed $\mathbf{x}_q^{(1:M)}$ and $\log q_\theta(\mathbf{x}_q^{(1:M)})$
  Add $\mathbf{x}_{\text{AIS}}^{(1:M)}$, $\log w_{\text{AIS}}^{(1:M)}$ and $\log q_\theta(\mathbf{x}_{\text{AIS}}^{(1:M)})$ to replay buffer

  **for** *iteration = 1 to L* **do** `// Sample from buffer and update` $q_\theta$
    Sample $\mathbf{x}_{\text{AIS}}^{(1:N)}$ and $\log q_{\theta_{\text{old}}}(\mathbf{x}_{\text{AIS}}^{(1:N)})$ from buffer with probability proportional to $w_{\text{AIS}}^{(1:N)}$
    Calculate $\log w_{\text{correction}}^{(1:N)} = \log q_{\theta_{\text{old}}}(\mathbf{x}_{\text{AIS}}^{(1:N)}) - \text{stop-grad}(\log q_\theta(\mathbf{x}_{\text{AIS}}^{(1:N)}))$
    Update $\log w_{\text{AIS}}^{(1:N)}$ and $\log q_{\theta_{\text{old}}}(\mathbf{x}_{\text{AIS}}^{1:N})$ in buffer to $\log w_{\text{AIS}}^{(1:N)} + \log w_{\text{correction}}^{(1:N)}$ and $\log q_\theta(\mathbf{x}_{\text{AIS}}^{1:N})$
    Calculate loss $\mathcal{S}'(\theta) = -1/N \sum_i^N w_{\text{correction}}^{(i)} \log q_\theta(\mathbf{x}_{\text{AIS}}^{(i)})$
    Perform gradient descent on $\mathcal{S}'(\theta)$ to update $\theta$

---

specified by $\theta$. We can approximate the gradient of Equation (7) using

$$\nabla_\theta \mathcal{S}'(\theta) \approx -\nabla_\theta \frac{1}{N} \sum_{i=1}^{N} \log q_\theta(\mathbf{x}_i), \tag{8}$$

where $\mathbf{x}_1, \ldots, \mathbf{x}_N$ are sampled from the buffer with probability proportional to their AIS weights. However, if the buffer data points have been generated with a previous value of $\theta$, denoted $\theta_{\text{old}}$, we have to multiply their AIS weights with a correction factor $w_{\text{correction}} = q_{\theta_{\text{old}}}(\mathbf{x})/q_\theta(\mathbf{x})$ before sampling, which requires to additionally store $q_{\theta_{\text{old}}}(\mathbf{x})$ for each data point in the buffer. Note that $q_\theta$ is in the denominator of this correction factor because $g(\mathbf{x})$, the distribution we are sampling from, is inversely proportional to $q_\theta$.

The resulting procedure extracts data from the buffer in a prioritized manner: We sample according to $g$, which favors points with low $q_\theta$ and high $p$. As $q_\theta$ is updated to fit samples from the buffer, it will take higher values on those samples and their weights will gradually be decreased to encourage drawing alternative samples. The buffer allows us to re-use old AIS samples and does not require re-evaluating $p(\mathbf{x})$, which could be expensive in some cases.

A limitation of the above approach is that it requires updating the AIS weights for all data points in the buffer before sampling, which is expensive. To significantly speed up computations, we instead draw a minibatch from the buffer with probability proportional to the old AIS weights and then reweight each sample with the corresponding $w_{\text{correction}}$. Before updating $\theta$, we update the AIS weights for the sampled points in the buffer and replace the respective $q_{\theta_{\text{old}}}(\mathbf{x})$ values with $q_\theta(\mathbf{x})$. The pseudocode for the final procedure is shown in Algorithm 1. In practice, we found that sampling from the buffer without replacement worked better, at the cost of introducing bias into our gradient estimates. Lastly, we set a maximum length for the buffer, and once this is reached we discard the oldest samples each time new samples are added.

## 4 EXPERIMENTS

This section contains an experimental evaluation of our proposed method. The code is publicly available at `https://github.com/lollcat/fab-torch`. It is written in PyTorch and uses the `normflows` package to implement the flows (Stimper et al., 2023). The Appendix contains a detailed description of each experiment to guarantee reproducibility. In Appendix F we include an additional set of experiments on the 32-dimensional *"Many Well"* distribution given by the product of 16 copies of the 2-dimensional Double Well distribution from Noé et al. (2019); Wu et al. (2020).

### 4.1 MIXTURE OF GAUSSIANS IN 2D

First, we consider a synthetic problem where $p$ is a mixture of bivariate Gaussians with 40 mixture components. The two-dimensional nature of this problem allows us to easily visualize the results

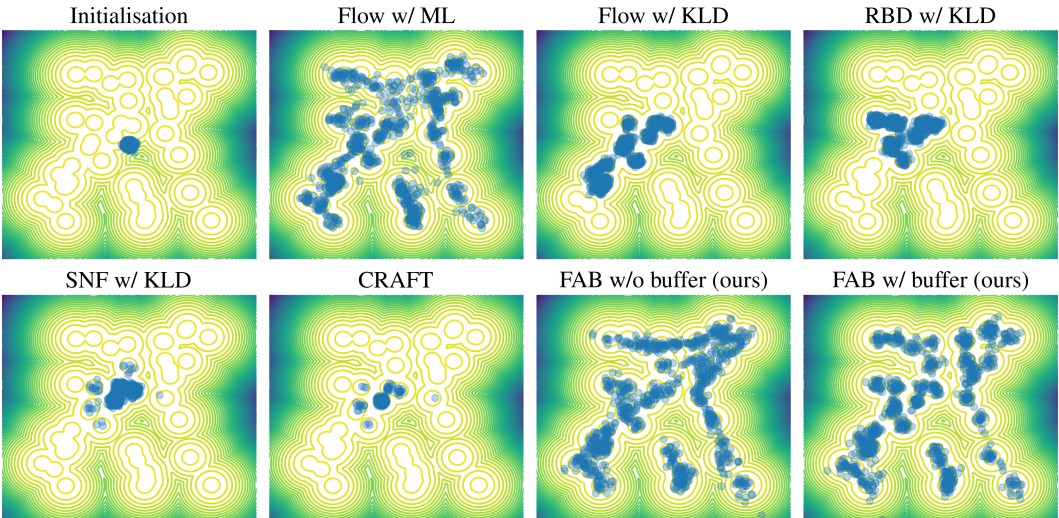

Figure 2: Contour lines for the target distribution $p$ and samples (blue discs) drawn from the approximation $q_\theta$ obtained by different methods on the mixture of Gaussians problem. The plot for the flow trained by $D_{\alpha=2}$, which had the worst performance, is shown in Appendix E.3.

of different methods while the multimodality of $p$ makes the problem relatively challenging. To increase the problem difficulty, we give the flow a pathological initialization where samples from $q_\theta$ concentrate in a small region of the sampling space, as illustrated in the top left plot in Figure 2.

We compare the following methods: 1) FAB with a replay buffer as shown in Algorithm 1; 2) FAB without a replay buffer, where we directly optimize Equation (7); 3) a flow model that minimizes $\mathrm{KL}(q_\theta\|p)$; 4) a flow with a Resampled Base Distribution (RBD) (Stimper et al., 2022) that minimizes $\mathrm{KL}(q_\theta\|p)$; 5) a Stochastic Normalizing Flow (SNF) model (Wu et al., 2020) that also minimizes $\mathrm{KL}(q_\theta\|p)$; 6) a Continual Repeated Flow Annealed Transport (CRAFT) model (Matthews et al., 2022) that minimizes a CRAFT specific version of KL divergence specified in Appendix C.2; 7) a flow model that minimizes $D_{\alpha=2}(p\|q_\theta)$ estimated using samples from $q_\theta$ (Müller et al., 2019), and 8) a flow trained by maximum likelihood (ML) using samples from $p$. In this toy problem, we have access to ground truth samples from the target, allowing us to train the flow by ML. However, we are interested in the case where samples from the target are not cheaply available. To denote this, we single out the results from this latter method in our tables with a horizontal dashed line.

All methods besides CRAFT[5] use the same parametric form for $q_\theta$ given by a Real NVP flow model with 15 layers (Dinh et al., 2017). For the FAB-based approaches, we run AIS with a single intermediate distribution ($\beta = 0.5$) and MCMC transitions with 1 Metropolis-Hastings step, being a Gaussian perturbation and then an accept-reject step. For FAB with prioritized buffer, we perform $L = 4$ gradient updates to $q_\theta$ per AIS sampling step. For the SNF and CRAFT, we do 1 Metropolis-Hastings step every 3 flow layers. For training the flow by ML, we draw new samples from the target for each loss estimation. All models are trained for $2 \cdot 10^7$ flow evaluations. Further details on the hyper-parameters and architectures used in each algorithm are provided in Appendix E.1.

Figure 2 shows that our two FAB-based methods and the flow trained by ML fit all modes in $p$. By contrast, the other alternative methods cover only a small subset of the modes. The reason for this is that such methods are trained only on samples from $q_\theta$ and the poor initialization of $q_\theta$ makes it unlikely that they will ever generate samples from undiscovered modes.

To evaluate the trained models we compute the effective sample size (ESS) obtained when doing importance sampling with $q_\theta$; the average log-likelihood of $q_\theta$ on samples from $p$; the forward KL divergence with respect to the target; and the mean absolute error (MAE) in the estimation of $\mathbb{E}_{p(\mathbf{x})}[f(\mathbf{x})]$ by importance sampling with $q_\theta$, where $f(\mathbf{x})$ is a toy quadratic function specified in

---

[5]CRAFT also uses an affine transform in the flow, but with an autoregressive dependency across dimensions (Kingma et al., 2016) instead of coupling.

Table 1: Results for the mixture of Gaussians problem. Our methods are marked in *italic*. Best results are emphasized in **bold**. Log-likelihood values for the first two methods are NaN because they assign zero density to samples from missing modes. Log-likelihood values for SNF and CRAFT are N/A because this method does not provide density values for the generated samples. The CRAFT implementation used does not allow for forward-KL to be estimated. Furthermore, the resampling step from CRAFT prevents the ESS from being estimated. Hence, these fields have N/A.

| | ESS (%) | $\mathrm{E}_{p(\mathbf{x})}\left[\log q(\mathbf{x})\right]$ | $\mathrm{KL}(p\|q)$ | MAE (%) | MAE w/o RW (%) |
|---|---|---|---|---|---|
| Flow w/ ML | $\mathbf{54.3 \pm 10.4}$ | $\mathbf{-7.18 \pm 0.05}$ | $\mathbf{0.31 \pm 0.05}$ | $\mathbf{9.0 \pm 0.3}$ | $\mathbf{6.1 \pm 2.0}$ |
| Flow w/ $D_{\alpha=2}$ | $0.7 \pm 0.3$ | $\mathrm{NaN} \pm \mathrm{NaN}$ | $\mathrm{NaN} \pm \mathrm{NaN}$ | $99.5 \pm 0.2$ | $99.6 \pm 0.1$ |
| Flow w/ KLD | $55.0 \pm 20.8$ | $\mathrm{NaN} \pm \mathrm{NaN}$ | $\mathrm{NaN} \pm \mathrm{NaN}$ | $26.0 \pm 3.1$ | $26.3 \pm 1.4$ |
| RBD w/ KLD | $37.9 \pm 16.3$ | $\mathrm{NaN} \pm \mathrm{NaN}$ | $\mathrm{NaN} \pm \mathrm{NaN}$ | $68.3 \pm 18.1$ | $92.1 \pm 1.9$ |
| SNF w/ KLD | $43.0 \pm 21.3$ | $\mathrm{N/A} \pm \mathrm{N/A}$ | $\mathrm{NaN} \pm \mathrm{NaN}$ | $64.6 \pm 20.6$ | $69.3 \pm 17.2$ |
| CRAFT | $\mathrm{N/A} \pm \mathrm{N/A}$ | $\mathrm{N/A} \pm \mathrm{N/A}$ | $\mathrm{N/A} \pm \mathrm{N/A}$ | $98.8 \pm 0.0$ | $99.1 \pm 0.0$ |
| *FAB w/o buffer* | $31.1 \pm 6.8$ | $-7.86 \pm 0.19$ | $1.00 \pm 0.19$ | $9.4 \pm 0.2$ | $\mathbf{4.4 \pm 0.7}$ |
| *FAB w/ buffer* | $\mathbf{61.9 \pm 8.0}$ | $\mathbf{-7.16 \pm 0.07}$ | $\mathbf{0.30 \pm 0.07}$ | $\mathbf{8.9 \pm 0.1}$ | $8.9 \pm 0.5$ |

Appendix E. We express the MAE as a percentage of the true expectation to ease interpretability. Finally, we also report the MAE that is obtained when we do not reweight samples according to the importance weights. We provide further details on the evaluation setup in Appendix E.2.

Table 1 shows for each method average results and corresponding standard errors over 3 random seeds. FAB with the buffer performs similarly to the benchmark of training the flow by ML. Both of these methods are the best performing ones with the highest ESS, the highest log-likelihood on samples from $p$ and the lowest forward KL divergence and lowest MAE. FAB without a replay buffer is the next best method while the other methods perform poorly. This is especially the case regarding the forward KL divergence and log-likelihood values on samples from $p$: the non FAB/ML methods assign zero density to points sampled from undiscovered modes of $p$, which is represented by writing NaN in the table. Note that the ESS for the SNF, RBD and the flow trained by minimizing $\mathrm{KL}(q_\theta\|p)$ are spurious as these methods are missing modes. Finally, note that the MAE in the estimation of $\mathbb{E}_{p(\mathbf{x})}\left[f(\mathbf{x})\right]$ via importance sampling with 1000 samples from $p$ is 8.6%. Also, the log-likelihood of $p$ on samples from this same distribution is $-6.85$. These are close to the values obtained by FAB with a replay buffer, meaning that the corresponding $q_\theta$ is close to $p$.

## 4.2 ALANINE DIPEPTIDE

We now consider the 22 atom molecule alanine dipeptide, shown in Figure 3a, in an implicit solvent at a temperature of $T = 300\,\mathrm{K}$ and aim to approximate its Boltzmann distribution given the 3D atomic coordinates. This is a popular benchmark when considering Boltzmann generators (Wu et al., 2020; Campbell et al., 2021; Dibak et al., 2022; Stimper et al., 2022). Previous works have used a coordinate transformation to map some but not all Cartesian coordinates to internal coordinates, which are normalized using their mean and standard deviation computed on samples generated by MD (Noé et al., 2019). Since we aim to train models without using any data, we replace the mean by the minimum energy configuration, which can be cheaply estimated through gradient descent within less than 100 steps. Similarly, we replace the standard deviations with values reflecting the typical order of magnitude of each variable. Furthermore, we represent the molecule with internal coordinates only, thereby implicitly satisfying the system's rotational and translational invariance.

We use Neural Spline Flows with 12 rational quadratic spline coupling layers (Durkan et al., 2019). Dihedral angles of those bonds that can move freely are treated as circular coordinates (Rezende et al., 2020), while the others are considered as unbound. The models trained with FAB use 8 intermediate distributions. For FAB with the replay buffer, we do $L = 8$ gradient updates per AIS forward pass. Alanine dipeptide is a chiral molecule, meaning that it can exist in two distinct forms (L-form and D-form) that are mirror images of each other, as illustrated in Figure 16. In nature, we find almost exclusively the L-form which is why only this form is considered in the literature. During training, we filter the samples generated by our flows and keep only those for the L-form, whereby the flow models learn to only generate this form. More details are given in Appendix G.1.

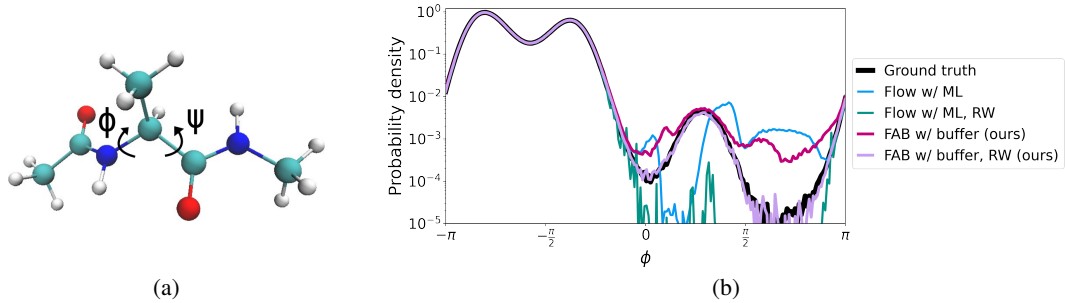

(a)                 (b)

Figure 3: (a) Visualization of alanine dipeptide and the dihedral angles $\phi$ and $\psi$ for the Ramachandran plot. (b) Marginal distribution of $\phi$ in log scale as given by the ground truth, the flow trained with ML on MD samples, and FAB with a replay buffer. RW indicates whether samples have been reweighted with importance sampling before generating the plot. Figure 17 is the same plot in normal scale, revealing how small the mode at $\phi \approx 1$ is.

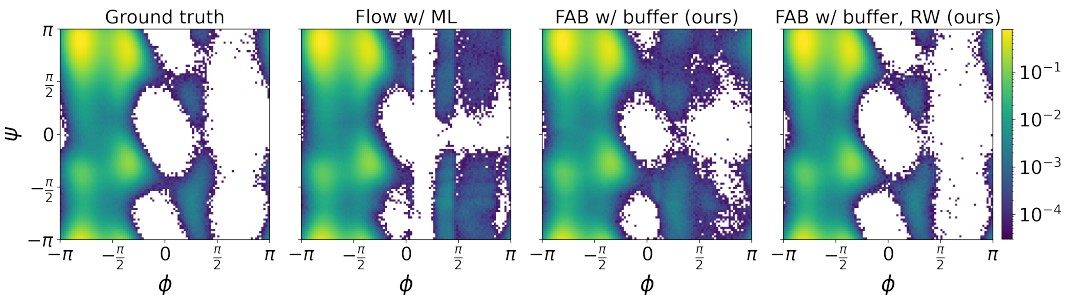

Figure 4: From left to right, Ramachandran plots of the ground truth generated by MD, a flow model trained by ML on MD samples, and by FAB using a replay buffer before and after reweighting samples to eliminate bias.

To evaluate our models, we generated samples using parallel tempering MD simulations, which serve as ground truth. They are split into training and validation sets with $10^6$ samples each and a test set with $10^7$ samples. We compare FAB to several baseline methods already mentioned in the previous section, see Table 2 for the full list. The SNF method performs 10 Metropolis-Hastings steps every two layers, meaning a total of 60 additional sampling steps. All methods are trained for $2.5 \times 10^8$ flow evaluations except for the SNF, which uses $6 \times 10^7$ as it is more expensive in terms of target evaluations. Table 8 provides an overview of the number of flow and target evaluations by each method. We compare methods via the ESS of importance sampling weights and the average log-likelihood on the test set. Moreover, we generate Ramachandran plots, which are histograms for the marginal distribution of the dihedral angles $\phi$ and $\psi$ illustrated in Figure 3a. We compute their KL divergence to the ground truth with and without reweighting using the importance weights. Our experiments are repeated over 3 random seeds and average values and standard errors are given.

Table 2 shows our results. The flow trained by minimizing $D_{\alpha=2}(p\|q_\theta)$ with samples from $q_\theta$ had convergence problems due to the high gradient variance. As a result, it performs very poorly in practice, especially in terms of ESS. The models trained by minimizing $\mathrm{KL}(q\|p)$ have no convergence problems, but they only approximate a subset of the target modes. This results in poor test log-likelihood and KLD values, and spurious values for the ESS. The flow trained by ML on MD samples obtains very good results in terms of test log-likelihood and KLD values. However, it struggles to model the dim mode at $\phi \approx 1$ correctly, as shown in Figures 3b and 4. Its ESS is also fairly low and, hence, reweighting worsens performance. The models trained with FAB have a higher ESS and test log-likelihood than the other methods. FAB with a buffer obtains lower KLD values than the flow trained by ML. When reweighting is applied to the samples generated by this version of FAB, the resulting distribution is nearly the same as the ground truth, as illustrated in Figures 3b and 4. These results show that FAB with a replay buffer outperforms the flow trained by ML on MD samples while using 100 times fewer evaluations of the target density, as shown in Table 8.

Table 2: ESS, log-likelihood on the test set, and KL divergence (KLD) of Ramachandran plots with and without reweighting (RW) for each method. Our methods are marked in *italic* and best results are emphasized in **bold**.

| | ESS (%) | $\mathrm{E}_{p(\mathbf{x})}\left[\log q(\mathbf{x})\right]$ | KLD | KLD w/ RW |
|---|---|---|---|---|
| Flow w/ ML | $2.8 \pm 0.6$ | $209.22 \pm 0.28$ | $\mathbf{(7.57 \pm 3.80) \times 10^{-3}}$ | $(2.58 \pm 0.80) \times 10^{-2}$ |
| Flow w/ $D_{\alpha=2}$ | $0.011 \pm 0.000$ | $73.5 \pm 1.3$ | $2.96 \pm 0.13$ | $17.5 \pm 0.2$ |
| Flow w/ KLD | $54 \pm 12$ | $100 \pm 32$ | $3.17 \pm 0.20$ | $3.15 \pm 0.19$ |
| RBD w/ KLD | $44 \pm 18$ | $143 \pm 22$ | $3.00 \pm 0.05$ | $3.00 \pm 0.04$ |
| SNF w/ KLD | $0.16 \pm 0.11$ | $\mathrm{N/A} \pm \mathrm{N/A}$ | $8.71 \pm 3.36$ | $9.58 \pm 2.68$ |
| *FAB w/o buffer* | $52.2 \pm 1.3$ | $211.13 \pm 0.03$ | $(6.28 \pm 0.33) \times 10^{-2}$ | $(2.66 \pm 0.90) \times 10^{-2}$ |
| *FAB w/ buffer* | $\mathbf{92.8 \pm 0.1}$ | $\mathbf{211.54 \pm 0.00}$ | $\mathbf{(3.42 \pm 0.45) \times 10^{-3}}$ | $\mathbf{(2.51 \pm 0.39) \times 10^{-3}}$ |

## 5 DISCUSSION AND RELATED WORK

SNFs combine flows with MCMC methods by introducing sampling layers between flow layers to improve model expressiveness (Wu et al., 2020; Nielsen et al., 2020). SNFs have been extended to CRAFT (Matthews et al., 2022; Arbel et al., 2021), where flows are combined with Sequential Monte Carlo (SMC). In CRAFT, flows are used to transport SMC samples between consecutive intermediate distributions, with each flow being trained by minimizing a KL divergence with respect to the next intermediate distribution. CRAFT improves the issue of mode seeking relative to SNFs, which can be seen in the Many Well problem in Appendix F where it performs well. However, CRAFT fails catastrophically on the GMM problem, as the CRAFT loss can still favour mode seeking and uses samples directly from the flow for its estimation which can provide a poor training signal. In Appendix C.2 we describe a new FAB-flavored version of CRAFT using the $D_{\alpha=2}(p\|q_\theta)$ as objective, estimating it with the MCMC samples ahead of the flow in the SMC process.

Within the MCMC/AIS literature, significant work has focused on improving transition kernels (Levy et al., 2018; Gabrié et al., 2022), intermediate distributions (Brekelmans et al., 2020), and the extended target distribution (Doucet et al., 2022a;b) of AIS. These techniques are applicable the AIS procedure used in FAB. FAB does not differentiate through AIS to obtain the gradient with respect to the flow. Combining FAB with methods that allow for differentiation through AIS (Geffner & Domke, 2021; Zhang et al., 2021; Doucet et al., 2022a) may allow for a lower variance gradient estimate. Works on differentiation through iterated systems would be relevant for this, notably (Metz et al., 2021). FAB does not use gradients of the target distribution when optimizing its loss function, although such gradients are used in the sampling process by HMC. This is in contrast with the alternative approach of training the flow by minimizing $\mathrm{KL}(q_\theta\|p)$, which does use these gradients. Such gradient information could be included in FAB through force matching (Wang et al., 2019; Köhler et al., 2021; Köhler et al., 2022) or the addition of a KL divergence loss term.

## 6 CONCLUSION

We have proposed FAB, a method for training flows to approximate complicated multimodal target distributions. FAB combines $\alpha$-divergence minimization with $\alpha = 2$ with an AIS bootstrapping mechanism for improving the samples used for the loss estimate. By focusing on this divergence, we favor mass-covering of multimodal distributions and minimize importance weight variance. Using AIS, FAB targets the ratio between the squared target density and the flow density, which provides a high-quality training signal by focusing on the regions where the flow is a poor approximation of the target. We have also proposed to use a prioritized replay buffer, which reduces the cost of FAB and improves performance. Our experiments show that FAB can produce accurate approximations of complex multimodal targets without using samples from such distributions. By contrast, other alternative approaches fail in this challenging setting. Remarkably, for the alanine dipeptide, FAB produces better results than training the flow by ML on samples generated via MD simulations while still using 100 fewer evaluations of the target than the MD simulations. In future work, we hope to scale up our approach to more challenging problems, such as the modelling of the Boltzmann distribution of large proteins.

ACKNOWLEDGMENTS

We thank Emile Mathieu, Kristopher Miltiadou, Alexandre Laterre, Clément Bonnet, and Alexander Matthews for the helpful discussions. José Miguel Hernández-Lobato acknowledges support from a Turing AI Fellowship under grant EP/V023756/1. This work was supported by the German Federal Ministry of Education and Research (BMBF): Tübingen AI Center, FKZ: 01IS18039B; and by the Machine Learning Cluster of Excellence, EXC number 2064/1 - Project number 390727645.

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

## A    EXTENDED DISCUSSION

Gabrié et al. (2022) use normalizing flows to learn the transition kernels for MCMC. These transition kernels are used to perform large MCMC steps between meta-stable states, improving the notorious issue of mixing in MCMC. In FAB, the flow has a similar function, although it is used as the base distribution for AIS rather than for a transition kernel. Namely, in FAB, the flow learns to balance mass between meta-stable states. This is done by using AIS and its importance weights to reweight inaccuracies in the mass allocated across different meta-stable states. In (Gabrié et al., 2022), *a priori* knowledge of the meta-stable states is required to obtain good performance. For example, they show that their approach fails on a 2D bimodal mixture of Gaussians problem if a mode state is missing in the model's initialization. FAB contrasts this, as the flow is able to incorporate modes discovered by AIS into the flow that were not present during initialization (see Figure 2). Notably, if a single sample from the AIS bootstrap process comes from a new mode, the flow will be updated strongly towards it immediately.

In our experiments, we have used a relatively low number of intermediate AIS distributions and minimal hyperparameter search. However, the performance of FAB could be improved by further tuning. For example, by trading off the reduction in variance from more intermediate distributions with the corresponding increase in compute cost. Furthermore, since the loss variance decreases throughout training, it may be beneficial to reduce the number of AIS distributions in an online fashion.

Wirnsberger et al. (2022) train flows to accurately approximate the Boltzmann distribution of same-atom atomic solids with up to 512 atoms just using the target distribution's density. Similarly to our approach on the alanine dipeptide molecule, they incorporated physical knowledge about the system into their base distribution and flow architecture. In their case, this corresponds to permutation and translation invariance, periodic boundary conditions and meta-stable states. FAB could help to scale their approach to larger systems with different atom species and more complex potentials. Moreover, incorporating the chiral structure of a molecule into the model architecture might simplify training and aid in applying FAB to larger proteins. In general, including prior knowledge of the system into the model is important for FAB, as it increases the effective sample size initially during training, which decreases the computational burden of AIS in reducing loss variance. Incorporating symmetries into the model has two key benefits. First, it often lets us operate in a lower dimensional space, alleviating the curse of dimensionality. Second, it often greatly reduces the number of modes in the distribution, as not incorporating symmetries causes multiple repeats of "the same" mode.

This work has focused on the application of normalizing flows. However, diffusion models have also shown great promise for learning Boltzmann generators. Jing et al. (2022) are able to train a single diffusion model to learn the Boltzmann distribution over the torsional angles of multiple molecules, while using cheminformatics methods for the bond lengths and angles. They perform energy-based training via estimation of a score matching loss using samples from the model. As with flows, this will exhibit high variance for complex target distributions, especially during initialization when the model is a poor match for the target. Thus, incorporating an AIS bootstrap process similar to FAB may improve training in these methods.

## B    DERIVATION OF THE LOSS

We consider the general case of training a parameterized probability distribution $q_\theta$ to minimize a loss function $\mathcal{L}(\theta) = \int f(\mathbf{x}, \theta) d\mathbf{x}$, where $f(\mathbf{x}, \theta) \geq 0$. Later, we will focus on the specific case of $D_{\alpha=2}(p\|q_\theta)$ minimization, where $f(\mathbf{x}, \theta) = p(\mathbf{x})^2/q_\theta(\mathbf{x})$, however, the general case is interesting as well and simplifies notation. Let us consider the gradient of $\mathcal{L}(\theta)$ written as an expectation over some distribution $g(\mathbf{x})$:

$$\nabla_\theta \mathcal{L}(\theta) = \int \nabla_\theta f(\mathbf{x}, \theta) d\mathbf{x} = \mathrm{E}_{g(\mathbf{x})} \left[ \frac{\nabla_\theta f(\mathbf{x}, \theta)}{g(\mathbf{x})} \right] . \tag{9}$$

We select $g(\mathbf{x})$ to be the minimum variance importance sampling distribution given by $g(\mathbf{x}) = f(\mathbf{x}, \theta)/Z_f$ where $Z_f = \int f(\mathbf{x}, \theta) d\mathbf{x}$ is the normalizing constant. Then, plugging in the identity $\nabla_\theta \log f(\mathbf{x}, \theta) = \nabla_\theta f(\mathbf{x}, \theta)/f(\mathbf{x}, \theta)$, i.e., applying the log-derivative trick similar to REINFORCE

(Williams, 1992), we obtain

$$\mathrm{E}_{g(\mathbf{x})} \left[ \frac{\nabla_\theta f(\mathbf{x},\theta)}{g(\mathbf{x})} \right] = Z_f \, \mathrm{E}_{g(\mathbf{x})} \left[ \frac{\nabla_\theta f(\mathbf{x},\theta)}{f(\mathbf{x},\theta)} \right] = Z_f \, \mathrm{E}_{g(\mathbf{x})} \left[ \nabla_\theta \log f(\mathbf{x},\theta) \right] . \tag{10}$$

We may not generally be able to sample directly from $g(\mathbf{x})$, but we can use AIS to estimate the right part of Equation (10). To do this, we consider running AIS with $p^2/q_\theta$ as the target and $q_\theta$ as the initial distribution. We note that, when the target density is unnormalized, the AIS weights are scaled by the target normalizing constant:

$$\begin{aligned} w_{\mathrm{AIS}}(\mathbf{x}_M) &= \frac{\tilde{p}_1\left(\mathbf{x}_1\right)}{p_0\left(\mathbf{x}_1\right)} \frac{\tilde{p}_2\left(\mathbf{x}_2\right)}{\tilde{p}_1\left(\mathbf{x}_2\right)} \cdots \frac{\tilde{p}_{M-1}\left(\mathbf{x}_{M-1}\right)}{\tilde{p}_{M-2}\left(\mathbf{x}_{M-1}\right)} \frac{\tilde{p}_M\left(\mathbf{x}_M\right)}{\tilde{p}_{M-1}\left(\mathbf{x}_M\right)} \\ &= \frac{Z_1 p_1\left(\mathbf{x}_1\right)}{p_0\left(\mathbf{x}_1\right)} \frac{Z_2 p_2\left(\mathbf{x}_2\right)}{Z_1 p_1\left(\mathbf{x}_2\right)} \cdots \frac{Z_{M-1} p_{M-1}\left(\mathbf{x}_{M-1}\right)}{Z_{M-2} p_{M-2}\left(\mathbf{x}_{M-1}\right)} \frac{Z_M p_M\left(\mathbf{x}_M\right)}{Z_{M-1} p_{M-1}\left(\mathbf{x}_M\right)} \\ &= Z_M \frac{p_1\left(\mathbf{x}_1\right)}{p_0\left(\mathbf{x}_1\right)} \frac{p_2\left(\mathbf{x}_2\right)}{p_1\left(\mathbf{x}_2\right)} \cdots \frac{p_{M-1}\left(\mathbf{x}_{M-1}\right)}{p_{M-2}\left(\mathbf{x}_{M-1}\right)} \frac{p_M\left(\mathbf{x}_M\right)}{p_{M-1}\left(\mathbf{x}_N\right)} , \end{aligned} \tag{11}$$

where $\tilde{p}_i$ denotes the unnormalized density for the $i$-th intermediate AIS distribution and $Z_i$ is the corresponding normalizing constant such that $p_i = \tilde{p}_i/Z_i$.

Given that we have set $p_M = g$, $\tilde{p}_M = f$ and $Z_M = Z_f$, expectations over the AIS forward pass hold the following relationship to expectations over $g$: $\mathrm{E}_{g(\mathbf{x})}[h(\mathbf{x})] = \mathrm{E}_{\mathrm{AIS}}\left[ \frac{w_{\mathrm{AIS}}(\mathbf{x})}{Z_f} h(\mathbf{x}) \right]$ where $h(\mathbf{x})$ is a function of interest.

Using this we can then write Equation (10) as an expectation over the AIS forward pass

$$\begin{aligned} Z_f \, \mathrm{E}_{g(\mathbf{x})}\left[ \nabla_\theta \log f(\mathbf{x},\theta) \right] &= Z_f \, \mathrm{E}_{\mathrm{AIS}}\left[ \frac{w_{\mathrm{AIS}}}{Z_f} \nabla_\theta \log f(\bar{\mathbf{x}}_{\mathrm{AIS}},\theta) \right] \\ &= \mathrm{E}_{\mathrm{AIS}}\left[ w_{\mathrm{AIS}} \nabla_\theta \log f(\bar{\mathbf{x}}_{\mathrm{AIS}},\theta) \right] , \end{aligned} \tag{12}$$

where $\bar{\mathbf{x}}_{\mathrm{AIS}}$ and $w_{\mathrm{AIS}}$ are the samples and corresponding importance weights generated by AIS when targeting $f$. We use the *bar* superscript to denote stopped gradients of the samples generated by AIS, $\bar{\mathbf{x}}_{\mathrm{AIS}}$, with respect to the parameters $\theta$. Now, returning to the case where we minimize $\mathcal{L}(\theta) = D_{\alpha=2}(p\|q_\theta)$, we set $f(x,\theta) = p(\mathbf{x})^2/q_\theta(\mathbf{x})$ to obtain

$$\begin{aligned} \nabla_\theta D_{\alpha=2}(p\|q_\theta) = Z_f \, \mathrm{E}_{g(\mathbf{x})}\left[ \nabla_\theta \log \frac{p(\mathbf{x})^2}{q_\theta(\mathbf{x})} \right] &= -Z_f \, \mathrm{E}_{g(\mathbf{x})}\left[ \nabla_\theta \log q_\theta(\mathbf{x}) \right] \\ &\approx -\frac{1}{N} \sum_i^N w_{\mathrm{AIS}}^{(i)} \nabla_\theta \log q_\theta(\bar{\mathbf{x}}_{\mathrm{AIS}}^{(i)}) , \end{aligned} \tag{13}$$

where $w_{\mathrm{AIS}}^{(i)}$ and $\mathbf{x}_{\mathrm{AIS}}^{(i)}$ are samples and weights generated by AIS with $p^2/q_\theta$ as target. Equation (13) provides an unbiased estimate of $\nabla_\theta \mathcal{L}(\theta)$. If we also stop the gradients of $w_{\mathrm{AIS}}$, we can then use the surrogate loss function

$$\mathcal{S}(\theta) = -\mathrm{E}_{\mathrm{AIS}}\left[ \bar{w}_{\mathrm{AIS}} \log q_\theta(\bar{\mathbf{x}}_{\mathrm{AIS}}) \right] , \tag{14}$$

where $\nabla_\theta \mathcal{L}(\theta) = \nabla_\theta \mathcal{S}(\theta)$. The surrogate loss function may then be estimated using Monte Carlo.

In practice, we found that replacing the unnormalized weights in Equation (14) with the normalized weights greatly improved training stability. We refer to the surrogate loss with normalized importance weights as $\mathcal{S}'(\theta)$. To normalize the weights we divide them by $\bar{Z}_f = \bar{\mathcal{L}}(\theta) = \mathrm{E}_{\mathrm{AIS}}\left[ \bar{w}_{\mathrm{AIS}} \right]$ such that $\int \bar{w}_{\mathrm{AIS}} \mathrm{d}\mathbf{x} = 1$, where the *bar* superscripts denotes stopped gradients. The relationship between $\mathcal{S}(\theta)$ and $\mathcal{S}'(\theta)$ is therefore $\mathcal{S}'(\theta) = \mathcal{S}(\theta)/\bar{\mathcal{L}}(\theta)$. The gradient of $\mathcal{S}'(\theta)$ has the same direction as the gradient of the original surrogate loss, $\mathcal{S}(\theta)$, but a has different magnitude. Using $\mathcal{S}'(\theta)$ instead of $\mathcal{S}(\theta)$ improves training stability by removing the effect of large fluctuations in the magnitude of $\mathcal{L}(\theta) = D_{\alpha=2}(p\|q_\theta)$ without changing the direction of the gradient as training proceeds.

Thus, we use the following estimate of the surrogate loss function for training:

$$\mathcal{S}'(\theta) = -\sum_i^N \frac{\bar{w}_{\mathrm{AIS}}^{(i)}}{\sum_i^N \bar{w}_{\mathrm{AIS}}^{(i)}} \log q_\theta(\bar{\mathbf{x}}_{\mathrm{AIS}}^{(i)}) , \tag{15}$$

where $\bar{w}_{\text{AIS}}^{(i)}$ and $\bar{\mathbf{x}}_{\text{AIS}}^{(i)}$ are the samples and importance weights generated by AIS but evaluated in practice using stopped gradients when computing the gradient of Equation (15). The use of self-normalization in the loss function introduces bias for finite $N$ for the estimation of $\mathcal{S}'(\theta)$. We use $q_\theta$ as the initial distribution for AIS, and a relatively small number of intermediate distributions to prevent the AIS forward pass from becoming too computationally expensive.

It is possible for $D_{\alpha=2}(p\|q_\theta)$ to have a very large value, and it can even be infinite. For example if $q$ assigns zero density to regions in $p$, then $D_{\alpha=2}(p\|q_\theta)$ is infinite. In practice we drop any points that have infinite/*NaN* density under $q$ during training. Sometimes early in training when $q$ is a poor approximation to the target, importance weights with *NaN* values arose, and thus were dropped. However, towards the end of training, when $q$ is a relatively accurate sampler, the importance weights were stable and infinite/*NaN* did not typically occur. This issue therefore did not result in any practical problems during training. Another example of when $D_{\alpha=2}(p\|q_\theta)$ can be very large, or even infinite is if target distribution is heavy tailed, and the tails of $q$ are light. Using the normalized importance weights (for FAB without the buffer), or sampling points from the buffer in proportion to their importance weights means that FAB is only effected by the direction of the gradient of $D_{\alpha=2}(p\|q_\theta)$ with respect to the parameters of the flow. This helps improve the robustness of FAB in situations where the magnitude of $D_{\alpha=2}(p\|q_\theta)$ is very large or has very high variance in its estimates. Using an architecture for $q$ that includes a defensive mixture component distribution with heavy tails would help address this issue, and this could be a way of improving the stability of FAB further (Owen, 2013).

## C    VARIATIONS OF FAB

As mentioned above, in this paper we focus on the minimization of $D_{\alpha=2}(p\|q_\theta)$ as estimated with our AIS bootstrap approach targeting $p^2/q_\theta$. However, the general approach of improving gradient estimation through the addition of the AIS bootstrap process may be applied in other settings. For example, in previous version of this work, we used a bound on $D_{\alpha=2}(p\|q_\theta)$ as objective, which we estimated with AIS targeting $p$ (Midgley et al., 2021).

In Appendix B above, the loss function in Equation (12) is written in a general manner and could therefore be used for any $f(\mathbf{x}, \theta) \geq 0$ and not only $D_{\alpha=2}(p\|q_\theta)$. We could simply plug other divergence measures of the form $\mathcal{L}(\theta) = \int f(\mathbf{x}, \theta)\mathrm{d}\mathbf{x}$ and satisfying $f(\mathbf{x}, \theta) \geq 0$ into Equation (12). In the below section we show how we can apply the FAB approach for $\alpha$-divergence minimization with other values of $\alpha$.

We can also apply the proposed approach to other types of models. For example in Appendix C.2 we show how we can obtain a FAB flavored version of Continual Repeated Annealed Flow Transport Monte Carlo (CRAFT) (Matthews et al., 2022).

### C.1    FAB FOR GENERIC $\alpha$-DIVERGENCE MINIMISATION

Below we provide a derivation of how the FAB may be generalized to $\alpha$-divergence minimization with arbitrary values of $\alpha$. We restate the definition of $\alpha$-divergence:

$$D_\alpha(p\|q) = -\frac{\int_x p(\mathbf{x})^\alpha q(\mathbf{x})^{1-\alpha}\mathrm{d}\mathbf{x}}{\alpha(1-\alpha)} .\tag{16}$$

We can write the gradient of the above expression as an expectation over an importance sampling distribution following the same approach as in Appendix B.

$$\nabla_\theta D_\alpha(p\|q_\theta) = -\frac{\nabla_\theta \int_x p(\mathbf{x})^\alpha q_\theta(\mathbf{x})^{1-\alpha}\mathrm{d}\mathbf{x}}{\alpha(1-\alpha)} = -\frac{1}{\alpha(1-\alpha)}\,\mathrm{E}_{g(\mathbf{x})}\left[\frac{\nabla_\theta\left(p(\mathbf{x})^\alpha q_\theta(\mathbf{x})^{1-\alpha}\right)}{g(\mathbf{x})}\right] .\tag{17}$$

Setting $f(\mathbf{x}, \theta) = p(\mathbf{x})^\alpha q_\theta(\mathbf{x})^{1-\alpha}$, we note that the integral in the above equation is in the form $\int f(\mathbf{x}, \theta)\mathrm{d}\mathbf{x}$ satisfying $f(\mathbf{x}, \theta) \geq 0$. Thus, we consider setting $g(\mathbf{x})$ to the minimum variance importance sampling distribution for estimating $D_\alpha(p\|q_\theta)$ given by $g(\mathbf{x}) = p(\mathbf{x})^\alpha q_\theta(\mathbf{x})^{1-\alpha}/Z_f$ where $Z_f = \int_x p(\mathbf{x})^\alpha q_\theta(\mathbf{x})^{1-\alpha}\mathrm{d}\mathbf{x}$. Using the result from Equation 12 we can then estimate the

gradient of $\nabla_\theta D_\alpha(p\|q_\theta)$ with AIS targeting $f$,

$$\nabla_\theta D_\alpha(p\|q_\theta) = -\frac{1}{\alpha(1-\alpha)} \, \mathrm{E}_{AIS} \left[ w_{\mathrm{AIS}} \nabla_\theta \log \left( p(\mathbf{x})^\alpha q_\theta(\mathbf{x})^{1-\alpha} \right) \right]$$
$$= -\frac{1}{\alpha} \, \mathrm{E}_{AIS} \left[ w_{\mathrm{AIS}} \nabla_\theta \log q_\theta(\mathbf{x}) \right] . \tag{18}$$

We see that plugging in $\alpha = 2$ to the above equation gives a gradient proportional to the original FAB gradient from Equation 13. Furthermore, if we plug in $\alpha = 1$, which is equivalent to minimizing forward KL divergence, we see that this results in maximizing the log probability of samples generated by AIS with $g = p$ as the target, multiplied by the AIS importance weight correction factor. This gradient is exactly equal to the gradient of the forward KL estimated with AIS,

$$\nabla_\theta \, \mathrm{KL}\,(p\|q_\theta) = \nabla_\theta \, \mathrm{E}_p \left[ \log p(\mathbf{x}) - \log q_\theta(\mathbf{x}) \right] = - \, \mathrm{E}_p \left[ \nabla_\theta \log q_\theta(\mathbf{x}) \right]$$
$$= - \, \mathrm{E}_{AIS} \left[ w_{\mathrm{AIS}} \nabla_\theta \log q_\theta(\mathbf{x}) \right] . \tag{19}$$

We note that this method does not hold for $D_{\alpha \to 0}(p\|q_\theta) = \mathrm{KL}\,(q\|p)$.

We can combine the generalised FAB loss from Equation 18 with the prioritised buffer training procedure from Algorithm 1 by instead simply setting the AIS target to $p(\mathbf{x})^\alpha q_\theta(\mathbf{x})^{1-\alpha}$ and $\log w_{\mathrm{correction}} = (1-\alpha)(\log q_\theta - \log q_{\theta_{\mathrm{old}}})$. We provide the pseudo code for this in Algorithm 2 below.

Using this algorithm, in Appendix E.3 and G.2 we analyse the performance of FAB with varying values of $\alpha$. We find that $\alpha = 2$ does best, which provides empirical support for this choice.

---

**Algorithm 2:** FAB for the minimization of $D_\alpha(p\|q)$ with a prioritized replay buffer

---

Set target $p$
Initialize flow $q$ parameterized by $\theta$
Initialize replay buffer to a fixed maximum size
**for** *iteration = 1 to K* **do** // Generate AIS samples and add to buffer
> Sample $\mathbf{x}_q^{(1:M)}$ from $q_\theta$ and evaluate $\log q_\theta(\mathbf{x}_q^{(1:M)})$
> Obtain $\mathbf{x}_{\mathrm{AIS}}^{(1:M)}$ and $\log w_{\mathrm{AIS}}^{(1:M)}$ using AIS with target $p^\alpha q_\theta^{1-\alpha}$ and seed $\mathbf{x}_q^{(1:M)}$ and $\log q_\theta(\mathbf{x}_q^{(1:M)})$
> Add $\mathbf{x}_{\mathrm{AIS}}^{(1:M)}$, $\log w_{\mathrm{AIS}}^{(1:M)}$ and $\log q_\theta(\mathbf{x}_{\mathrm{AIS}}^{(1:M)})$ to replay buffer
>
> **for** *iteration = 1 to L* **do** // Sample from buffer and update $q_\theta$
> > Sample $\mathbf{x}_{\mathrm{AIS}}^{(1:N)}$ and $\log q_{\theta_{\mathrm{old}}}(\mathbf{x}_{\mathrm{AIS}}^{(1:N)})$ from buffer with probability proportional to $w_{\mathrm{AIS}}^{(1:N)}$
> > Calculate $\log w_{\mathrm{correction}}^{(1:N)} = (1-\alpha)(\text{stop-grad}(\log q_\theta(\mathbf{x}_{\mathrm{AIS}}^{(1:N)})) - \log q_{\theta_{\mathrm{old}}}(\mathbf{x}_{\mathrm{AIS}}^{(1:N)}))$
> > Update $\log w_{\mathrm{AIS}}^{(1:N)}$ and $\log q_{\theta_{\mathrm{old}}}(\mathbf{x}_{\mathrm{AIS}}^{1:N})$ in buffer to $\log w_{\mathrm{AIS}}^{(1:N)} + \log w_{\mathrm{correction}}^{(1:N)}$ and $\log q_\theta(\mathbf{x}_{\mathrm{AIS}}^{1:N})$
> > Calculate loss $\mathcal{S}'(\theta) = -1/N \sum_i^N w_{\mathrm{correction}}^{(i)} \log q_\theta(\mathbf{x}_{\mathrm{AIS}}^{(i)})$
> > Perform gradient descent on $\mathcal{S}'(\theta)$ to update $\theta$

---

## C.2 FAB APPLIED TO CRAFT

Continual Repeated Annealed Flow Transport Monte Carlo (CRAFT) is an extension of SNFs (Wu et al., 2020; Nielsen et al., 2020) proposed by Matthews et al. (2022) which combines normalizing flows with sequential Monte Carlo (SMC). Specifically, flows are used to transport samples between consecutive annealing distributions in combination with SMC. CRAFT trains each of the flows by minimizing the reverse KL divergence with respect to the next annealing distribution and gradients are estimated using the samples generated by the flow transport step.

FAB could be combined with the CRAFT model. For example, the reverse KL divergence could be replaced with a mass-covering divergence such as the $\alpha$-divergence with $\alpha = 2$. Furthermore, we could improve the estimation of gradients by targeting with AIS the minimum variance distribution for importance sampling instead of just the next annealing distribution.

A FAB style version of the CRAFT algorithm is described in algorithms 3 and 4. Each flow transport step is now trained by minimizing the $\alpha$-divergence with $\alpha = 2$ and using samples generated by the

next immediate MCMC step *ahead* of the flow in the SMC process. Below, we briefly introduce CRAFT with an emphasis on the loss function used. Next, we describe how FAB may be used to improve training of the flow transport steps. We follow the notation from the CRAFT paper exactly and use their pseudo code as a basis for our proposed algorithm indicating changes clearly. We refer to (Matthews et al., 2022) for further details on the CRAFT algorithm.

### C.2.1 CONTINUAL REPEATED ANNEALED FLOW TRANSPORT MONTE CARLO

As in FAB, the aim of CRAFT is to approximate an intractable target distribution that we cannot sample from and whose density can only be evaluated up to a normalizing constant. This target distribution is denoted $\pi_K(\mathbf{x})$ (equivalent to $p(\mathbf{x})$ in our notation). In CRAFT, SMC is run with interleaved flow transport steps through a sequence of annealed distributions $(\pi_k(\mathbf{x}))_{k=0}^{K}$, each with normalization constant $Z_k$. The base distribution $\pi_0$ is a tractable distribution (e.g., a Gaussian) from which we can sample. Similarly to AIS, $(\pi_k(\mathbf{x}))_{k=1}^{K-1}$ are defined by interpolating between base and target log-densities, where the target density may be unnormalized. The SMC process in CRAFT begins by sampling from the base distribution $X_0^i \sim \pi_0$. Then, for each distribution from $k = 1$ to $k = K$, a flow $T_k$ is trained to transport samples from $\pi_{k-1}(\mathbf{x})$ to $\pi_k(\mathbf{x})$. Additionally, at each step from $k = 1$ to $k = K$, CRAFT utilises resampling and MCMC to bring the samples closer to $\pi_k(\mathbf{x})$.

Similarly to AIS, the CRAFT algorithm returns a set of points and normalized importance weights $(X_K^i, W_K^i)_{i=1}^{N}$, which may be used for approximating expectations with respect to the target. Each point $X_k^i$ has an associated normalized importance weight $W_K^i$ for importance sampling with respect to the intermediate target distribution $\pi_k(\mathbf{x})$. We refer back to the CRAFT paper, and to the pseudo code in Algorithm 3 for how these importance weights are calculated.

CRAFT minimizes the following training objective:

$$H = \sum_{k=1}^{K} \mathrm{KL}[T_k^{\#}\pi_{k-1}||\pi_k], \tag{20}$$

where $\#$ denotes the push forward between distributions. The above objective trains each flow transport step $T_k$ to minimize the KL divergence between $T_k^{\#}\pi_{k-1}$, i.e., the distribution of outputs of the flow when given as input samples from $\pi_{k-1}$, and the next intermediate distribution $\pi_k$. The gradient estimate used to train each flow transport step $T_k$ is given by

$$\nabla_{\theta_k} H \approx \sum_i W_{k-1}^i \nabla_{\theta_k} \left[ -\log \gamma_k(T_k(X_{k-1}^i)) - \log |\nabla_x T_k(X_{k-1}^i)| \right], \tag{21}$$

where $\gamma_k(\mathbf{x}) \propto \pi_k(\mathbf{x})$. The flow is trained by passing it samples $X_{k-1}$ from the previous SMC step, computing the corresponding output samples from the flow $T_k(X_{k-1})$ and using these to estimate the gradient of $\mathrm{KL}[T_k^{\#}\pi_{k-1}||\pi_k]$. The normalized importance weight in the loss $W_{k-1}^i$ account for the fact that the samples $X_{k-1}^i$ passed from the previous step in the SMC forward pass come from an approximation to $\pi_{k-1}$.

### C.2.2 FAB-CRAFT

We now propose a FAB flavored version of CRAFT. First, we re-introduce some notation from our paper: We use $q$ to denote the initial distribution used in AIS and $p$ to denote the target distribution that we wish to approximate. Recall that $q$ is trained to fit $p$. All other notation in this section follows the CRAFT paper's notation.

In our FAB-CRAFT method, we use the MCMC samples following each flow transport step in CRAFT to update the flow to minimize $D_{\alpha=2}(p||q)$, where $q = T_k^{\#}\pi_{k-1}$ and $p = \pi_k$. To do this with minimal changes to the original CRAFT algorithm, we make the observation that sampling from the initial distribution $q$ and then running MCMC targeting $p$ is equivalent to running AIS targeting $p^2/q$ with 1 intermediate distribution at $\beta = 0.5$. Thus, the samples generated by the MCMC steps following each flow transport step in CRAFT can be repurposed for an AIS bootstrap estimate of the flow training loss. For training, the only adjustment to the SMC forward pass of CRAFT is then to move the resampling step to occur after each MCMC step, where previously it

occurred after each flow transport step. At inference time the original CRAFT algorithm can be run with the flows trained with our method in its exact original form. We describe this in more detail below and provide pseudo code in Algorithm 3 and 4.

We begin by deriving the AIS importance weights when targeting $p^2/q$ with 1 intermediate distribution and setting $\beta = 0.5$. For only 1 intermediate distribution, the AIS weights are given by

$$w_{\text{AIS}}(\mathbf{x}_2) = \frac{p_1(\mathbf{x}_1)}{p_0(\mathbf{x}_1)} \frac{p_2(\mathbf{x}_2)}{p_1(\mathbf{x}_2)}. \tag{22}$$

As before, we set the intermediate distributions as interpolations between the base and the target: $\log p_i(\mathbf{x}) = \beta_i \log p_0(\mathbf{x}) + (1 - \beta_i) \log p_N(\mathbf{x})$. Now, if we set $\beta_1 = 0.5$, then plugging in $p_0 = q$, $p_1 = q^{0.5}(p^2/q)^{0.5} = p$ and $p_2 = p^2/q$, we obtain the following AIS weights:

$$w_{\text{AIS}}(\mathbf{x}_2) = \frac{p(\mathbf{x}_1)}{q(\mathbf{x}_1)} \frac{p(\mathbf{x}_2)}{q(\mathbf{x}_2)}. \tag{23}$$

Recall that in CRAFT we set $q = T_k^{\#} \pi_{k-1}$ and $p = \pi_k$. AIS is then run by first sampling $\mathbf{x}_1$ from $q$, which is done in practice by setting $Y_k^i \leftarrow T_k(X_{k-1}^i)$ where $X_{k-1}^i \sim \pi_{k-1}$ and then generating $\mathbf{x}_2$ from $\mathbf{x}_1$ by MCMC, which is done in practice by setting $X_k^i \sim \mathcal{K}_k(Y_k^i)$, where $\mathcal{K}_k$ is an MCMC transition kernel that leaves $\pi_k$ invariant. The importance weights of $X_k^i$ with respect to the AIS target $p^2/q$ are then given by

$$w_{\text{AIS},k}^i = p(Y_k^i)/q(Y_k^i) \times p(X_k^i)/q(X_k^i). \tag{24}$$

Using the normalized importance weights $W_{\text{AIS},k}^i = w_{\text{AIS},k}^i / \sum_{j=1}^{N} w_{\text{AIS},k}^j$, we can calculate the FAB gradient estimate given by

$$\hat{h}_k = -\sum_i^N W_{\text{AIS},k}^i \nabla_{\theta_k} \log q_{\theta_k}(X_k^i)$$
$$= -\sum_i^N W_{\text{AIS},k}^i \nabla_{\theta_k} \left[ \log \gamma_{k-1}(T_{\theta_k}^{-1}(X_k^i)) + \log |\nabla_x T_{\theta_k}^{-1}(X_k^i)| \right]. \tag{25}$$

This assumes that the samples passed to the flow are from the distribution $\pi_{k-1}$. However, in practice, these samples are passed from the previous SMC step which is an approximation to $\pi_{k-1}$. Similarly as in the original CRAFT loss, see Equation 21, we can correct for this by instead using

$$\hat{h}_k = -\sum_i^N W_{k-1}^i W_{\text{AIS},k}^i \nabla_{\theta_k} \left[ \log \gamma_{k-1}(T_{\theta_k}^{-1}(X_k^i)) + \log |\nabla_x T_{\theta_k}^{-1}(X_k^i)| \right] \tag{26}$$

where $W_{k-1}^i$ accounts for $X_{k-1}^i$ coming from an approximation to $\pi_{k-1}$.

Calculating the normalized AIS weights requires all the samples from the flow to be passed to the MCMC step. Because of this, we move the SMC resampling step to take place after the MCMC step instead of just after the flow transport step, see Algorithm 3. Note that the weights $W_k$ for resampling $X_k$ are equal to the weights for resampling the corresponding flow outputs $Y_k$ that generate such samples. This result is due to the MCMC kernel $\mathcal{K}_k$ leaving $\pi_k$ invariant. The resulting FAB flavor of CRAFT is shown in algorithms 3 and 4.

**Some final remarks**: our goal has been to create a FAB flavored version of CRAFT while keeping the algorithm as similar to the original version as possible. However, in practice, it would be better to make further changes. For example, using a prioritized replay buffer would significantly decrease the computational requirements of the algorithm. Furthermore, for updating each flow, it may also be beneficial to consider samples across the whole chain of intermediate distributions, instead of using only samples from the local MCMC step immediately following the flow.

## D ANALYSIS OF FAB

### D.1 GRADIENT ESTIMATION PERFORMANCE

We first analyze the quality of the noisy gradients provided by the proposed AIS bootstrap method. For this, we consider a toy problem in which $q_\theta$ and $p$ are unit variance 1D Gaussians with means 0.5

**Algorithm 3:** SMC-NF-step for FAB-CRAFT: Additions are in green and removals in red
.

1: **Input:** Approximations $(\pi_{k-1}^N, Z_{k-1}^N)$ to $(\pi_{k-1}, Z_{k-1})$, normalizing flows $T_k$, unnormalized annealed targets $\gamma_{k-1}$ and $\gamma_k$ and resampling threshold $A \in [1/N, 1)$.

2: **Output:** Gradient $\hat{h}_k$ of FAB loss w.r.t $\theta_k$, particles at iteration $k$: $\pi_k^N = (X_k^i, W_k^i)_{i=1}^N$, approximation $Z_k^N$ to $Z_k$.

3: Transport particles: $Y_k^i = T_k(X_{k-1}^i)$.

4: Compute IS weights:
$w_k^i \leftarrow W_{k-1}^i G_k(X_{k-1}^i)$ // unnormalized
$W_k^i \leftarrow w_k^i / \sum_{j=1}^N w_k^j$ // normalized

5: Estimate normalizing constant $Z_k$:
$Z_k^N \leftarrow Z_{k-1}^N \left( \sum_{i=1}^N w_k^i \right)$.

6: Compute effective sample size $\text{ESS}_k^N$.

7: **if** $\text{ESS}_k^N \leq NA$ **then**

8:     Resample $N$ particles denoted abusively also $Y_k^i$ according to the weights $W_k^i$, then set $W_k^i = \frac{1}{N}$.

9: **end if**

10: Generate samples and IS weights via AIS targetting $p^2/q_{\theta_k}$ with 1 intermediate distribution, where $p = \pi_k$ and $q = T_k^\# \pi_{k-1}$. By setting $\beta = 0.5$ we simply run the original CRAFT MCMC transition kernel with $p$ as a target.
Sample $X_k^i \sim \mathcal{K}_k(Y_k^i, \cdot)$. // MCMC
$w_{\text{AIS},k}^i \leftarrow p(Y_k^i)/q(Y_k^i) \times p(X_k^i)/q(X_k^i)$
$W_{\text{AIS},k}^i \leftarrow w_{\text{AIS},k}^i / \sum_{j=1}^N w_{\text{AIS},k}^j$

11: Estimate gradient of FAB objective
$\hat{h}_k = -\sum_i^N W_{k-1}^i W_{\text{AIS},k}^i \nabla_{\theta_k} \left[ \log \gamma_{k-1}(T_{\theta_k}^{-1}(X_k^i)) + \log |\nabla_x T_{\theta_k}^{-1}(X_k^i)| \right]$

12: **if** $\text{ESS}_k^N \leq NA$ **then**

13:     Resample $N$ particles denoted abusively also $X_k^i$ according to the weights $W_k^i$, then set $W_k^i = \frac{1}{N}$.

14: **end if**

15: Return $\left( \pi_k^N, Z_k^N, \hat{h}_k \right)$.

---

**Algorithm 4:** CRAFT-training: Additions are shown in green and removals in red

1: **Input:** Initial NFs $\{T_k\}_{1:N}$, number of particles $N$, unnormalized annealed targets $\{\gamma_k\}_{k=0}^K$ with $\gamma_0 = \pi_0$ and $\gamma_K = \gamma$, resampling threshold $A \in [1/N, 1)$.

2: **Output:** Learned flows $T_k$ and length $J$ sequence of approximations $(\pi_K^N, Z_K^N)$ to $(\pi_K, Z_K)$.

3: **for** $j = 1, \ldots, J$ **do**

4:     Sample $X_0^i \sim \pi_0$ and set $W_0^i = \frac{1}{N}$ and $Z_0^N = 1$.

5:     **for** $k = 1, \ldots, K$ **do**

6:         $\hat{h}_k \leftarrow \texttt{flow-grad}\left(T_k, \pi_{k-1}^N\right)$ using eqn (21).

7:         $\left(\pi_k^N, Z_k^N, \hat{h}_k\right) \leftarrow \texttt{SMC-NF-step}\left(\pi_{k-1}^N, Z_{k-1}^N, T_k\right)$

8:         Update the flow $T_k$ using gradient $\hat{h}_k$.

9:     **end for**

10:     Yield $(\pi_K^N, Z_K^N)$ and continue for loop.

11: **end for**

12: Return learned flows $\{T_k\}_{k=1}^K$.

---

and $-0.5$, respectively, as shown in Figure 5a. We estimate the gradient of $D_{\alpha=2}(p \| q_\theta)$ with respect to the mean of $q_\theta$ and compare different methods: first, importance sampling (IS) with samples from $q_\theta$; second, IS with samples from $p$; third, AIS with $p$ as target and $q_\theta$ as initial distribution; and fourth, our proposed method using AIS with $p^2/q_\theta$ as target and $q_\theta$ as initial distribution. For AIS

we use 3 intermediate distributions and, as transition operator, HMC with 5 leapfrog steps and resampling of momentum variables once per intermediate AIS distribution.

Figure 5b shows the Signal-to-noise ratio (SNR) for the different gradient estimators as a function of the number of samples used. AIS bootstrap is clearly the best method. IS with $q_\theta$ performs very poorly and it is outperformed by both IS with $p$ and AIS targeting $p$, with these two latter techniques performing similarly but way worse than AIS bootstrap. Figure 5c shows that the quality of the proposed method increases fast as the number of intermediate AIS distributions grows, with IS with samples from $p$ being outperformed quite early in the plot while still using a rather small number of distributions. It is important to note, however, that in more challenging problems, it is unlikely that our AIS bootstrap method will outperform IS with samples from $p$, especially early in training when $q_\theta$ is a poor approximation to $p$.

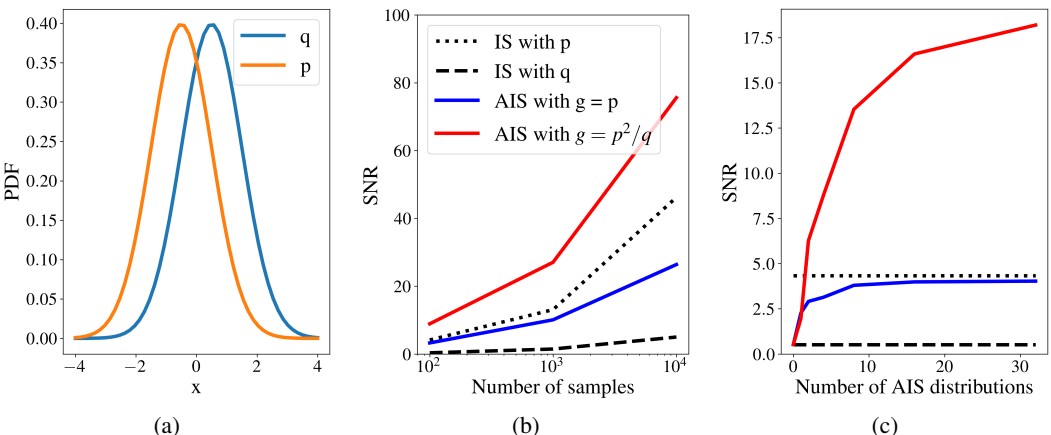

(a)                              (b)                              (c)

Figure 5: (a) Target and approximating densities $p$ and $q$, respectively. (b) Signal-to-noise ratio (SNR) of various gradient estimators as a function of the number of samples used. (c) SNR as a function of the number of intermediate AIS distributions when using 100 samples. Legend as in (b).

### D.2    Scaling FAB to higher dimensions

Now we consider how the performance of FAB is affected by an increasing problem dimensionality. To investigate this, we analyse a simple scenario where we assume $p$ and $q$ to be factorised, with each marginal of $q$ having its own separate parameters (no parameter sharing between dimensions). We acknowledge that these are strong simplifying assumptions and leave a more general analysis to future work. Given these assumptions, we show the following: 1) The variance in the estimates of the gradient of $D_{\alpha=2}$ by importance sampling with samples from $q$ and $p$ increases exponentially with respect to the dimensionality of the problem. 2) The variance in the corresponding estimates obtained with FAB can remain constant if the number of AIS intermediate distributions increases linearly with the dimensionality of the problem.

#### D.2.1    Theoretical analysis on factorized $p$ and $q$

We consider the problem of estimating the gradient of $D_{\alpha=2}$ where both $p$ and $q$ are factorized distributions: they are equal to the product of their marginals. To further simplify the analysis, we consider the gradient with respect to the parameters of the $j$-th dimension of $q$ and assume that there is no parameter sharing across dimensions of $q$. In this case, $D_{\alpha=2}$ is given by

$$\mu = \int \frac{p(\mathbf{x})^2}{q(\mathbf{x})}\,\mathrm{d}\mathbf{x} = \prod_d^D \int \frac{p_d(x_d)^2}{q_d(x_d)}\,\mathrm{d}x_d\,, \tag{27}$$

where $p_d$ and $q_d$ are the marginal distributions for dimension $d$. The gradient of this quantity with respect to the parameters $\theta_j$ of the $j$-th marginal of $q$ is given by,

$$\nabla_{\theta_j}\mu = \nabla_{\theta_j} \int \frac{p_j(x_j)^2}{q_j(x_j)}\,\mathrm{d}x_j \times \prod_{d \neq j}^{D} \int \frac{p_d(x_d)^2}{q_d(x_d)}\,\mathrm{d}x_d\,. \tag{28}$$

We are interested in studying how the variance in our estimate of $\nabla_{\theta_j}\mu$ scales with $D$. Equation (28) shows that each additional dimension $d \neq j$ adds an extra factor $\mu_d = \int p_d(x_d)^2/q_d(x_d)\,\mathrm{d}x_d$ in the gradient expression. To eliminate the effect of this change in the gradient as $D$ increases, we focus our analysis on $\mathrm{Var}[\nabla_{\theta_j}\bar{\mu}/\prod_{d\neq j}^{D}\mu_d]$, where $\bar{\mu}$ is an estimate of $\mu$ and $\prod_{d\neq j}^{D}\mu_d$ is a normalization factor that cancels the effect of the additional dimensions.

**Importance Sampling with $q$**: We consider first how increasing $D$ affects the variance in the estimation of the gradient of $D_{\alpha=2}$ by importance sampling with samples from $q$. The importance sampling estimate of $D_{\alpha=2}$ with $N$ samples from $q$ is given by

$$\bar{\mu}_q = \frac{1}{N}\sum_{n=1}^{N} w(\mathbf{x}_n)^2\,, \tag{29}$$

where $w(\mathbf{x}_n) = p(\mathbf{x}_n)/q(\mathbf{x}_n)$ and $\mathbf{x}_n \sim q$. Now, let us define $w_d(x_{n,d}) = p_d(x_{n,d})/q_d(x_{n,d})$, where $x_{n,d}$ is the $d$-th entry in $\mathbf{x}_n$. We then obtain

$$\nabla_{\theta_j}\bar{\mu}_q = \frac{1}{N}\nabla_{\theta_j}\sum_n \prod_d^{D} w_d(x_{n,d})^2 = \frac{1}{N}\sum_n \nabla_{\theta_j}w_j(x_{n,j})^2 \prod_{d\neq j} w_d(x_{n,d})^2\,. \tag{30}$$

The variance of this estimate, after dividing by the aforementioned normalization factor, is given by

$$N\,\mathrm{Var}\left[\frac{\nabla_{\theta_j}\bar{\mu}_q}{\prod_{d\neq j}^{D}\mu_d}\right] = \mathrm{E}_{q_j}\left[\left\{\nabla_{\theta_j}w_j(x_{n,j})^2\right\}^2\right] \times \prod_{d\neq j}^{D}\frac{\mathrm{E}_{q_d}\left[w_d(x_{d,n})^4\right]}{\mu_d^2} - (\nabla_{\theta_j}\mu_j)^2\,. \tag{31}$$

Since $\mathrm{Var}_{q_d}\left[w_d(x_{d,n})^2\right] = \mathrm{E}_{q_d}\left[w_d(x_{d,n})^4\right] - \mu_d^2 > 0$, we have that $\mathrm{E}_{q_d}\left[w_d(x_{d,n})^4\right]/\mu_d^2 \geq 1$. Thus, the first factor in Equation (31) is multiplied in this equation by $D-1$ factors all larger than 1, which implies that the variance of this estimator increases exponentially as a function of $D$. This is a well-known problem of importance sampling.

**Importance Sampling with $p$**: Interestingly, we get a similar result when estimating $D_{\alpha=2}$ by importance sampling with samples from $p$. The estimate for the gradient of $D_{\alpha=2}$ with respect to the $j$-th dimension of $q$ is now

$$\nabla_{\theta_j}\bar{\mu}_p = \frac{1}{N}\sum_n \nabla_{\theta_j}w_j(x_{n,j})\prod_{d\neq j}w_d(x_{n,d})\,, \tag{32}$$

where $x_{n,d} \sim p_d$. The variance of this gradient estimate, after dividing by the normalization factor, is given by

$$N\,\mathrm{Var}\left[\frac{\nabla_{\theta_j}\bar{\mu}_p}{\prod_{d\neq j}^{D}\mu_d}\right] = \mathrm{E}_{p_j}\left[\left\{\nabla_{\theta_j}w_j(x_{n,j})\right\}^2\right] \times \prod_{d\neq j}^{D}\frac{\mathrm{E}_{p_d}\left[w_d(x_{n,d})^2\right]}{\mu_d^2} - (\nabla_{\theta_j}\mu_j)^2\,. \tag{33}$$

Since $\mathrm{Var}_{p_d}\left[w_d(x_{d,n})\right] = \mathrm{E}_{p_d}\left[w_d(x_{d,n})^2\right] - \mu_d^2 > 0$, we have that $\mathrm{E}_{p_d}\left[w_d(x_{d,n})^2\right]/\mu_d^2 \geq 1$. Thus, the first factor in Equation (33) is again multiplied by $D-1$ factors all larger than 1, which implies that the variance of this estimator increases exponentially as a function of $D$, albeit at a lower rate than in the case of importance sampling with $q$. This implies that, even with access to ground truth samples from $p$, the number $N$ of samples required to keep the variance of the gradient estimates of $D_{\alpha=2}$ constant grows exponentially as a function of the problem dimensionality $D$.

**FAB**: We now apply the same type of analysis to the estimates of the gradient given by FAB. We consider the FAB gradient estimate from Equation (13), which is equal in expectation to $\nabla_\theta\mu$. This estimate relies on the raw importance weights from AIS rather than the self-normalized importance

weights, which makes it easier to analyze. The FAB estimate of the gradient of $D_{\alpha=2}$ with respect to the parameters of the $j$-th marginal of $q$ is given by

$$
\begin{aligned}
\nabla_{\theta_j} \bar{\mu}_{\text{FAB}} &= \frac{1}{N} \nabla_{\theta_j} \sum_n \log q(\mathbf{x}_n) \bar{w}(\mathbf{x}_n) \\
&= \frac{1}{N} \sum_n \nabla_{\theta_j} \log q_j(x_{n,j}) \bar{w}_j(x_{n,j}) \times \prod_{d \neq j} \bar{w}_d(x_{n,d}),
\end{aligned}
\tag{34}
$$

where $\bar{w}$ are the importance weights from AIS with stopped gradients and we have decomposed the AIS weights into the contributions from each dimension: $\bar{w}(\mathbf{x}_n) = \prod_d \bar{w}_d(x_{n,d})$. Note that, since $p$ and $q$ are factorized, we have that all the intermediate AIS distributions are factorized as well. If we assume that the MCMC transition kernels in AIS produce independent samples from the ground truth intermediate target distributions, we have that $x_{n,1}, \ldots, x_{n,D}$ are independent random variables. The variance of $\nabla_{\theta_j} \bar{\mu}_{\text{FAB}}$ after dividing by the normalization factor is then given by

$$
N \operatorname{Var} \left[ \frac{\nabla_{\theta_j} \bar{\mu}_{\text{FAB}}}{\prod_{d \neq j}^D \mu_d} \right] = \mathrm{E}_{\text{AIS}_j} \left[ \left\{ \nabla_{\theta_j} \log q_j(x_{j,n}) \bar{w}_j(x_{n,j}) \right\}^2 \right] \times \prod_{d \neq j}^D \frac{\mathrm{E}_{\text{AIS}_d} \left[ \bar{w}_d(x_{n,d})^2 \right]}{\mu_d^2} - (\nabla_{\theta_j} \mu_j)^2, \tag{35}
$$

where $\mathrm{E}_{\text{AIS}}$ denotes the expectation with respect to the AIS forward pass. The first expectation in the equation above is constant as $D$ increases. Therefore, we focus on the contributions of the other expectations for the importance weights of dimensions $d \neq j$. As in the previous cases where we used importance sampling with samples from $q$ and $p$, the variance in Equation (35) will again increase exponentially with $D$. However, under the assumption that the MCMC transitions produce independent samples from the intermediate AIS distributions, Neal (2001) shows that the variance in the log importance weights of AIS is proportional to $D/K$ where $K$ is the number of intermediate AIS distributions. This implies that $\prod_{d \neq j}^D \mathrm{E}_{\text{AIS}_d}[\bar{w}_d(x_{n,d})^2]/\mu_d^2$ will remain roughly constant if we increase the number $K$ of AIS distributions by the same factor as the dimensionality of the problem. In this case, the variance of $\nabla_{\theta_j} \bar{\mu}_{\text{FAB}}$ will remain constant as we increase $D$ and the cost of the FAB gradient estimator will only increase linearly as $D$ increases.

### D.2.2 EMPIRICAL ANALYSIS ON TOY PROBLEM

We now run an empirical analysis to assess the performance of the FAB gradient estimator as $D$ increases. We consider the case where $q$ and $p$ are both factorized Gaussians with unit marginal variances and with mean vectors equal to $0.5 \times \mathbf{1}^D$ and $-0.5 \times \mathbf{1}^D$, where $\mathbf{1}^D$ is a vector of dimension $D$ with all of its entries equal to one. For the FAB gradient estimate, we increase the number of AIS distributions by the same factor as the dimensionality of the problem. We analyze the SNR in the estimates of the gradient of the mean for the first marginal of $q$ as the dimensionality of the problem increases. The AIS transition operators are performed by running a single iteration of HMC, with 5 leapfrog steps, with a step size of 0.5. We found our results to be sensitive to the choice of step size. In practice, we selected this value by trial and error, assessing the quality of the AIS samples by looking at their empirical histogram, as shown in Figure 7.

Figure 6a shows that, when using importance sampling with samples from $p$, the log-weight variance increases linearly as the number of dimensions increases. By contrast, this variance remains constant with FAB. This is achieved by fixing the number of AIS intermediate distributions to be equal to the number of dimensions. This result is consistent with the analysis from the previous section and with the results of Neal (2001). Furthermore, in Figure 6b we see that the SNR remains roughly flat for FAB (stays within a single order of magnitude), while it quickly decreases for importance sampling with samples from $p$. If the same results were to hold in more complex problems, this would imply that we could safely apply FAB in those settings by linearly increasing the number of intermediate AIS distributions as the dimensionality of the problem grows.

## E  MIXTURE OF GAUSSIANS EXPERIMENTS

### E.1  TRAINING SETUP

All flow models have 15 RealNVP layers (Dinh et al., 2017), with a 2 layer (80 unit layer width) MLP for the conditioner. The flow is initialized to the identity transformation, so $q_\theta$ is initially a

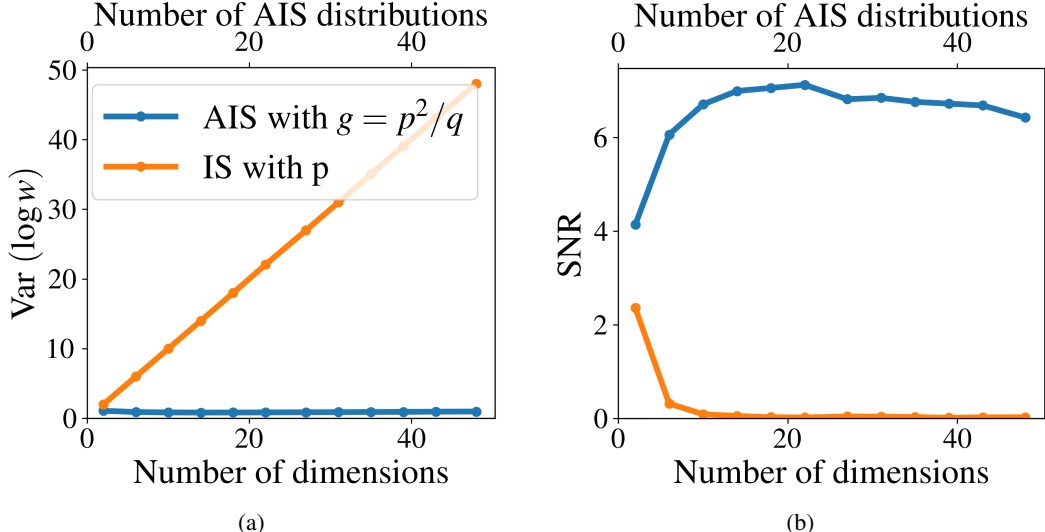

Figure 6: Analysis of efficiency of AIS bootstrap with increasing dimensionality. For AIS, we set the number of AIS distributions to be equal to the number of dimensions of the problem—which results in a linearly increasing compute cost as the dimension scales. (a) Variance in the log importance weights, for the AIS bootstrap vs importance sampling with $p$. (b) Signal-to-noise ratio (SNR) for gradient estimation with the AIS bootstrap vs importance sampling with $p$. Legend as in (a).

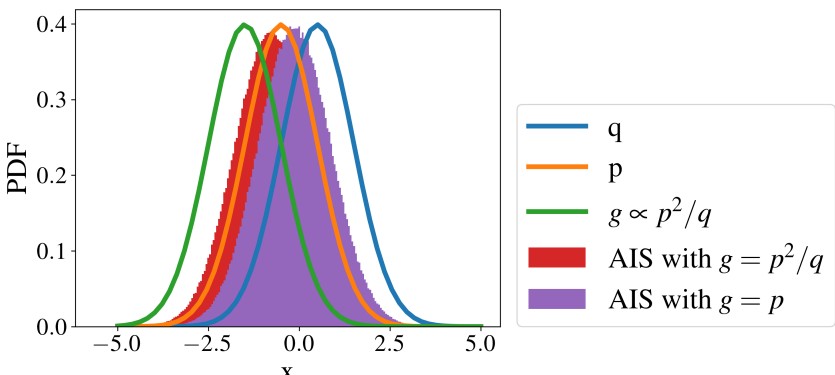

Figure 7: Histogram of samples from AIS targeting $p$ and $p^2/q$ compared to the PDF of $p$, $q$ and (normalized) $p^2/q$. The AIS samples are generated with the tuned HMC step size of 0.5. This tuning was performed by trial and error using the displayed histogram. AIS is run with 4 intermediate distributions, and the transition to each intermediate distribution is performed by running a single iteration of HMC with 5 leapfrog steps.

standard Gaussian distribution. Training is performed with a batch size of 128, using the Adam optimizer (Kingma & Ba, 2015) with a learning rate of $1 \times 10^{-4}$ and we clip the gradient norm to a maximum value of 100. For the model that uses a RBD, we use an acceptance function composed of a residual network with three blocks containing 512 hidden units per layer. The truncation parameter is set to the common value $T = 100$. For the SNF and CRAFT methods, we do 1 Metropolis-Hastings step every three flow layers. We used a fixed step size of $\sigma = 5.0$ for the Gaussian perturbation of the Metropolis-Hastings step, which is the same as what is used within AIS for FAB. This means that the SNF and CRAFT models has 5 stochastic Metropolis-Hastings steps in total. The CRAFT model uses 6 annealing temperatures with geometric spacing, and a resampling threshold of 0.3. We use the code provided by (Matthews et al., 2022) at `https://github.com/deepmind/annealed_flow_transport` for training the CRAFT model. We train all

models for $2 \times 10^7$ flow evaluations. For each method, we train 3 models, each with a different random seed, and results are reported as averages over these seeds.

**FAB specific details**: The batch size for both the AIS forward pass $(M)$ and sampling from the buffer $(N)$ is equal to 128. We run AIS with a single intermediate distribution $(\beta = 0.5)$ and MCMC transitions are given by a single Metropolis-Hastings step: a Gaussian perturbation and then an accept-reject step. We used a fixed step size of $\sigma = 5.0$ for the Gaussian perturbation. We initialize the buffer with 1280 samples from the initialized flow-AIS combination and use a maximum buffer length of 12800. We do not use any clipping when computing $w_{\text{correction}}$. The log density of the flow occasionally gave *NaN* values to points sampled from the buffer, resulting in *NaN* values for $w_{\text{correction}}$. As this resulted in *NaN* loss values, the parameter update was skipped in iterations where this occurred. Furthermore, since the $w_{\text{correction}}$ adjustment for these points in the buffer is invalid, the weights and $q_{\theta_{\text{old}}}(\mathbf{x})$ values in the buffer were left as their previous values.

Table 3: Number of flow and target evaluations during training for each method on the mixture of Gaussians problem. For SNF and CRAFT we report the number of model forward passes - each of which contains multiple flow transport and MCMC steps.

|  | Number of flow/model evaluations | Number of target evaluations |
|---|---|---|
| Flow w/ ML | $2 \cdot 10^7$ | $2 \cdot 10^7$ |
| Flow w/ $D_{\alpha=2}$ | $2 \cdot 10^7$ | $2 \cdot 10^7$ |
| Flow w/ KLD | $2 \cdot 10^7$ | $2 \cdot 10^7$ |
| RBD w/ KLD | $2 \cdot 10^7$ | $2 \cdot 10^7$ |
| SNF w/ KLD | $2 \cdot 10^7$ | $10^8$ |
| CRAFT | $2 \cdot 10^7$ | $10^8$ |
| *FAB w/o buffer* | $2 \cdot 10^7$ | $2 \cdot 10^7$ |
| *FAB w/ buffer* | $2 \cdot 10^7$ | $6.6 \cdot \times 10^6$ |

## E.2 Evaluation Setup

For each method, we compute after training the effective sample size (ESS) obtained when doing importance sampling with $q_\theta$; the average log-likelihood of $q_\theta$ on samples from $p$; the forward KL divergence with respect to the target; and the mean absolute error (MAE) in the estimation of $\mathbb{E}_{p(\mathbf{x})}[f(\mathbf{x})]$ by importance sampling with 1000 samples from $q_\theta$, where $f(\mathbf{x}) = \mathbf{a}^{\mathrm{T}}(\mathbf{x} - 2\mathbf{b}) + 2(\mathbf{x} - 2\mathbf{b})^{\mathrm{T}}\mathbf{C}(\mathbf{x} - 2\mathbf{b})$, with the entries in vectors $\mathbf{a}$ and $\mathbf{b}$ and matrix $\mathbf{C}$ randomly initialized by sampling from a standard Gaussian then kept fixed to such values during all the experiments. We express the MAE as a percentage of the true expectation to make it easier to interpret. We also report the MAE that is obtained when we do not reweight samples according to the importance weights. The ESS is calculated using $5 \times 10^4$ samples from $q_\theta$. The MAE is calculated by averaging over 100 repetitions.

## E.3 Further Results

Figure 8 shows a plot of samples from each trained model on the mixture of Gaussians problem, with the target contours in the background. We see that the FAB based methods and the flow trained with ML cover all the modes in the target distribution. All the other methods fit a subset of the modes. The flow trained with $D_{\alpha=2}$ minimization exhibited highly unstable behavior during training and, thus, is the worst performing model.

**FAB with varying values of $\alpha$** In Appendix C.1, we derived a variant of FAB that works with an arbitrary value for the $\alpha$ parameter of the $\alpha$-divergence. Here, we want to investigate how the performance of models trained with FAB changes as we vary $\alpha$. Therefore, we leave the setup the same as used in Section 4.1 and only changed the $\alpha$ parameter of FAB. The results when using FAB without the replay buffer are given in Table 4 and Figure 9. The results when using FAB with the replay buffer are reported in Table 5 and Figure 10. For FAB without the replay buffer, $\alpha = 2$ is slightly superior in performance to FAB with other values of $\alpha$. For FAB with the replay buffer, we see that all of the methods with $\alpha \geq 1$ are able to obtain a good fit for the target, with $\alpha = 2$ and

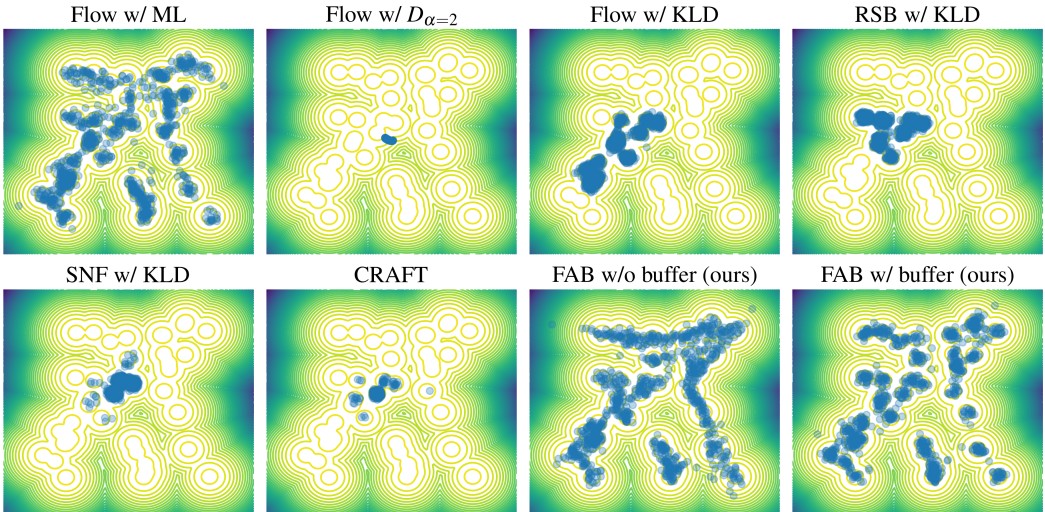

Figure 8: Contour lines for the target distribution $p$ and samples (blue discs) drawn from the approximation $q_\theta$ obtained by different methods on the mixture of Gaussians problem.

$\alpha = 3$ achieving the best performance. For these runs, the limits of the expressiveness of the flow is most likely the limiting factor to improving performance even further. The same style of analysis for FAB with varying values of $\alpha$ is performed with the Alanine Dipeptide problem in Appendix G.2, which finds $\alpha = 2$ is best, with a larger differences in performance between different values of $\alpha$.

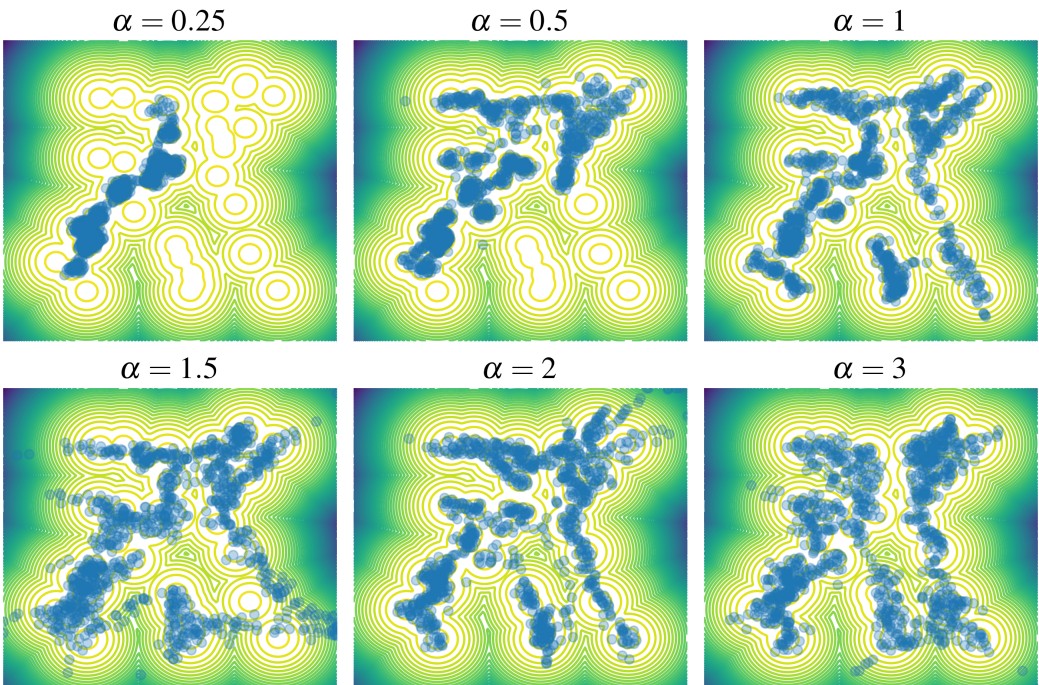

Figure 9: Contour lines for the target distribution $p$ and samples (blue discs) drawn from the approximation $q_\theta$ obtained by FAB **without** the replay buffer for varying values of $\alpha$ on the mixture of Gaussians problem.

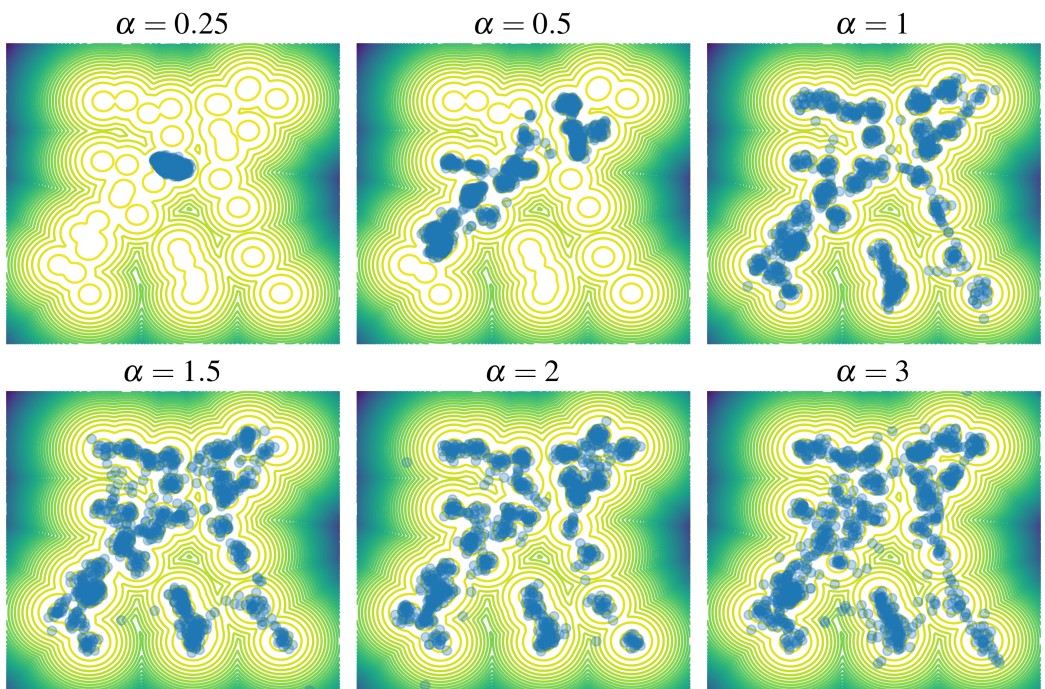

Figure 10: Contour lines for the target distribution $p$ and samples (blue discs) drawn from the approximation $q_\theta$ obtained by FAB **with** the replay buffer for varying values of $\alpha$ on the mixture of Gaussians problem.

Table 4: Results for the mixture of Gaussians problem for FAB **without** the replay buffer for varying values of $\alpha$. Log-likelihood values for the first two methods are NaN because they assign zero density to samples from missing modes. Best results are emphazised in **bold**.

| $\alpha$ | ESS (%) | $\mathrm{E}_{p(\mathbf{x})}\left[\log q(\mathbf{x})\right]$ | KL$(p\|q)$ | MAE (%) | MAE w/o RW (%) |
|---|---|---|---|---|---|
| 0.25 | $52.4 \pm 10.0$ | NaN $\pm$ NaN | NaN $\pm$ NaN | $44.2 \pm 19.9$ | $48.8 \pm 18.2$ |
| 0.5 | $14.7 \pm 5.7$ | NaN $\pm$ NaN | NaN $\pm$ NaN | $14.6 \pm 3.6$ | $15.9 \pm 6.0$ |
| 1.0 | $30.8 \pm 2.2$ | $\mathbf{-7.59 \pm 0.08}$ | $\mathbf{0.73 \pm 0.08}$ | $4.8 \pm 0.4$ | $\mathbf{4.3 \pm 1.3}$ |
| 1.5 | $19.0 \pm 10.1$ | $-7.97 \pm 0.22$ | $1.10 \pm 0.22$ | $6.2 \pm 1.2$ | $13.4 \pm 7.5$ |
| 2.0 | $\mathbf{38.4 \pm 3.2}$ | $\mathbf{-7.59 \pm 0.06}$ | $\mathbf{0.73 \pm 0.06}$ | $\mathbf{3.7 \pm 0.2}$ | $7.6 \pm 1.3$ |
| 3.0 | $21.6 \pm 7.9$ | $-8.09 \pm 0.23$ | $1.23 \pm 0.23$ | $5.8 \pm 0.9$ | $19.3 \pm 11.4$ |

## F  MANY WELL EXPERIMENTS

### F.1  DESCRIPTION AND RESULTS

We consider another synthetic problem that is significantly more difficult than the GMM problem: approximating the 32-dimensional *"Many Well"* distribution given by the product of 16 copies of the 2-dimensional Double Well distribution[6] from Wu et al. (2020); Noé et al. (2019):

$$\log p(x_1, x_2) = -x_1^4 + 6x_1^2 + 1/2x_1 - 1/2x_2^2 + \text{constant}, \qquad (36)$$

where each copy of the Double Well is evaluated on a different pair of the 32 inputs to the Many Well. The original Double Well has two modes as shown in the top-right contour plot in Figure 11. Therefore, our 32-dimensional Many Well has $2^{16} = 65536$ modes, one for each possible choice of

---

[6]We use the Double Well distribution from the code provided in Wu et al. (2020), which has different coefficients to the Noé et al. (2019).

Table 5: Results for the mixture of Gaussians problem for FAB **with** the replay buffer for varying values of $\alpha$. Log-likelihood values for the first two methods are NaN because they assign zero density to samples from missing modes. Best results are emphasized in **bold**.

| $\alpha$ | ESS (%) | $\mathrm{E}_{p(\mathbf{x})}\left[\log q(\mathbf{x})\right]$ | KL$(p\|\|q)$ | MAE (%) | MAE w/o RW (%) |
|---|---|---|---|---|---|
| 0.25 | $44.4 \pm 18.2$ | $\mathrm{NaN} \pm \mathrm{NaN}$ | $\mathrm{NaN} \pm \mathrm{NaN}$ | $91.7 \pm 3.4$ | $91.5 \pm 3.5$ |
| 0.5 | $23.6 \pm 4.0$ | $\mathrm{NaN} \pm \mathrm{NaN}$ | $\mathrm{NaN} \pm \mathrm{NaN}$ | $35.8 \pm 6.8$ | $37.1 \pm 7.6$ |
| 1.0 | $59.1 \pm 3.4$ | $-7.19 \pm 0.00$ | $0.33 \pm 0.00$ | $10.1 \pm 2.7$ | $11.4 \pm 3.7$ |
| 1.5 | $48.2 \pm 10.8$ | $\mathbf{-7.16 \pm 0.02}$ | $0.30 \pm 0.02$ | $3.6 \pm 0.1$ | $\mathbf{2.8 \pm 0.2}$ |
| 2.0 | $\mathbf{63.1 \pm 3.4}$ | $-7.14 \pm 0.02$ | $0.28 \pm 0.01$ | $\mathbf{3.0 \pm 0.1}$ | $3.0 \pm 0.5$ |
| 3.0 | $\mathbf{65.6 \pm 1.7}$ | $-7.15 \pm 0.03$ | $0.30 \pm 0.03$ | $3.2 \pm 0.1$ | $\mathbf{2.8 \pm 0.1}$ |

mode in each of the 16 copies of the Double Well. We obtain exact samples from the Many Well by sampling from each independent copy of the Double Well. Exact samples from the Double Well are obtained by sampling from each independent marginal distribution. The first marginal $p(x_1)$ can be sampled from exactly using rejection sampling (see Appendix F.2), while the second marginal distribution $p(x_2)$ can be sampled from directly as it is a (unnormalized) standard Gaussian. These samples are cheap to produce. We use them for training a flow by maximum likelihood as well as for the evaluation of the different methods. Additionally, we created an artificial test set for evaluation purposes by manually placing a point on each of the $2^{16}$ modes. By computing log-likelihoods on this test set, we can then check if a method is covering the entire target distribution, as any missing mode will result in very low log-likelihood values. We can calculate the normalizing constant for each marginal of the Double Well problem via numerical integration (for $p(x_1)$) and analytical integration (for $p(x_2)$), and use this to obtain the normalizing constant of the Many Well distribution (see Appendix F.2). This may then be used to obtain the normalized probability density function of the Many Well distribution, which is useful for model evaluation. We can also compare models on how accurately they estimate the normalizing constant as the average unnormalized importance weights. For each model, we report the MAE in the estimation of the Many Well's normalizing constant using 1000 samples, averaged over 50 runs. We express this as a percentage of the true value of the normalizing constant.

Table 6: Results on the 32 dimensional Many Well Problem. Our methods are marked in *italic*. Best results are emphasized in **bold**. CRAFT (config 1) refers to the CRAFT model trained with a similar flow and MCMC config to FAB. CRAFT (config 2) uses the configuration provided in the CRAFT repository, that uses more expressive neural spline flows, and a larger number of intermediate distributions within SMC.

| | ESS (%) | $\mathbb{E}_{p(\mathbf{x})}\left[\log q(\mathbf{x})\right]$ | Mean $\log q(\mathbf{x}_{\mathrm{modes}})$ | KL$[p\|\|q]$ | MAE (%) |
|---|---|---|---|---|---|
| Flow w/ ML | $\mathbf{80.6 \pm 2.1}$ | $\mathbf{-27.6 \pm 0.01}$ | $\mathbf{-21.3 \pm 0.0}$ | $\mathbf{0.1 \pm 0.0}$ | $\mathbf{1.2 \pm 0.0}$ |
| Flow w/ $D_{\alpha=2}$ | $0.0 \pm 0.0$ | $\mathrm{NaN} \pm \mathrm{NaN}$ | $\mathrm{NaN} \pm \mathrm{NaN}$ | $\mathrm{NaN} \pm \mathrm{NaN}$ | $\mathrm{NaN} \pm \mathrm{NaN}$ |
| Flow w/ KLD | $27.7 \pm 6.0$ | $-176.9 \pm 18.08$ | $-536.6 \pm 41.2$ | $149.4 \pm 18.1$ | $89.2 \pm 1.2$ |
| RBD w/ KLD | $51.9 \pm 16.8$ | $-183.8 \pm 21.8$ | $-533.7 \pm 73.2$ | $156.2 \pm 21.8$ | $88.9 \pm 2.2$ |
| SNF w/ KLD | $15.1 \pm 9.7$ | $\mathrm{N/A} \pm \mathrm{N/A}$ | $\mathrm{N/A} \pm \mathrm{N/A}$ | $167.9 \pm 12.8$ | $88.9 \pm 0.0$ |
| CRAFT (config 1) | $\mathrm{N/A} \pm \mathrm{N/A}$ | $\mathrm{N/A} \pm \mathrm{N/A}$ | $\mathrm{N/A} \pm \mathrm{N/A}$ | $\mathrm{N/A} \pm \mathrm{N/A}$ | $116.3 \pm 1.8$ |
| CRAFT (config 2) | $\mathrm{N/A} \pm \mathrm{N/A}$ | $\mathrm{N/A} \pm \mathrm{N/A}$ | $\mathrm{N/A} \pm \mathrm{N/A}$ | $\mathrm{N/A} \pm \mathrm{N/A}$ | $3.0 \pm 1.7$ |
| *FAB w/o buffer* | $4.7 \pm 0.9$ | $-29.8 \pm 0.23$ | $-28.0 \pm 43.0$ | $2.3 \pm 0.2$ | $12.7 \pm 1.9$ |
| *FAB w/ buffer* | $\mathbf{78.9 \pm 1.6}$ | $\mathbf{-27.6 \pm 0.01}$ | $\mathbf{-21.3 \pm 0.0}$ | $\mathbf{0.1 \pm 0.0}$ | $1.4 \pm 0.1$ |

We compare FAB to the same alternative approaches as in the mixture of Gaussians problem and use also the Real NVP flow architecture but with 10 layers. The MLP used for the conditioner is composed of 2 layers each with 320 units. For the the model that uses a RBD, we use the same architecture as before, with an acceptance function composed of a residual network with three blocks containing 512 hidden units per layer. The truncation parameter is set to the common value $T = 100$. For FAB based methods, we use AIS with 4 intermediate distributions (linearly spaced) and with

a HMC transition operator containing a single iteration with 5 leapfrog steps. For FAB with prioritized buffer we use $L = 8$. The SNF model uses 1 step of HMC with 5 inner leapfrog steps every 2 layers. As before, when training the flow by maximum likelihood, we draw new samples from the target for each loss estimation. All models except CRAFT are trained for $10^{10}$ flow evaluations with the number of target evaluations by each method being reported in Table 7. For CRAFT we run our experiments with two different setups, which we simply refer to as CRAFT (config 1) and CRAFT (config 2). The first uses a configuration that is similar to the that of FAB in terms of the flow architecture, and MCMC. This CRAFT model uses 5 temperatures, and uses an autoregressive affine flow. The second setup uses the configuration provided in the CRAFT implementation at `https://github.com/deepmind/annealed_flow_transport`, which uses neural spline flows, with 11 temperatures (10 flow/MCMC steps) and HMC containing 10 leapfrog steps. This model is significantly more expensive both in terms of the flow, and in terms of the MCMC performed in each forward pass. For CRAFT we train for $10^{10}$ target evaluation budget - this is the same number used in FAB without the buffer, and slightly more than FAB with the buffer. Further details on the hyper-parameters and architectures used by each algorithm are provided in Appendix F.3.

Figure 11 shows contour plots for several two-dimensional marginals of the Many Well target. Each plot is obtained by scanning two variables that are inputs to different Double Well factors in the Many Well distribution while the other variables are kept fixed to zero. We also show in this figure samples generated by FAB with a replay buffer (left) and by the method that tunes $q_\theta$ by minimizing $KL(q\|p)$ (right). We see that FAB generates samples on each of the contour modes while this is not the case for the alternative baseline, which misses several modes. Figure 12 shows the same contour plot as above for the two CRAFT models, both of which successfully sample from all the modes. Additional plots for all other methods can be found in Appendix F.4.

Table 6 shows for each method 1) the ESS when doing importance sampling with $q_\theta$; the average log-likelihoods for $q_\theta$ 2) on samples from the target and 3) on test points placed on the modes of the Many Well distribution; 4) the forward KL divergence with respect to the target; and 5) the MAE in the estimation of the May Well normalizing constant. Average log-likelihoods and ESS are calculated with $5 \times 10^4$ samples. All the results in the table are averages across 3 random seeds. We see similar results as in the previous experiment: FAB with a buffer performs similarly to the benchmark of training the flow by maximum likelihood. These are the two best performing methods, obtaining the highest ESS and average log-likelihoods and the lowest forward KL divergence and MAE values. The next best method is the CRAFT (config 2) model, and then FAB without buffer, while the other methods perform very poorly. The method that minimizes $D_{\alpha=2}(p\|q_\theta)$ as estimated by sampling from $q_\theta$ diverged early in training and always returned NaN values. The ESS for the flow trained by minimizing $KL(q_\theta\|p)$, RBD and the SNF are spurious, as they are missing modes (see Figure 11 and 14). After training, we may combine the trained flows with AIS to further improve the ESS. If we run AIS as during training but targeting $p$ instead of $p^2/q$, the ESS is 89.9% and 13.6% for the FAB flows trained with and without a buffer, respectively. The log-likelihood of $p$ on samples from $p$ and on the test set with points at the modes are -27.4 and -20.9, respectively. These values are very close to the ones obtained by FAB with buffer, showing that this method produces highly accurate approximations to the target distribution.

Table 6 shows that the CRAFT (config 2) model, which uses the more expressive spline flow architecture, and a larger number of intermediate distributions in SMC, performs well and is able to provide accurate estimates of the normalizing constant for the Many Well. However, the flow trained with FAB provides more accurate estimates, even though the CRAFT model runs a large amount of HMC at evaluation time. This comparison could be made more fair by taking the flow trained with FAB, and then at inference time combining it with AIS targetting $p$.

## F.2 OBTAINING THE NORMALIZING CONSTANT AND EXACT SAMPLES

In this section, we describe how to obtain the exact normalizing constant and exact samples from the Double Well distribution. These allow us to obtain the normalizing constant and exact samples from the Many Well distribution.

The Double Well log-density is given by

$$\log p(x_1, x_2) = -x_1^4 + 6x_1^2 + 1/2x_1 - 1/2x_2^2 + \text{constant}. \tag{37}$$

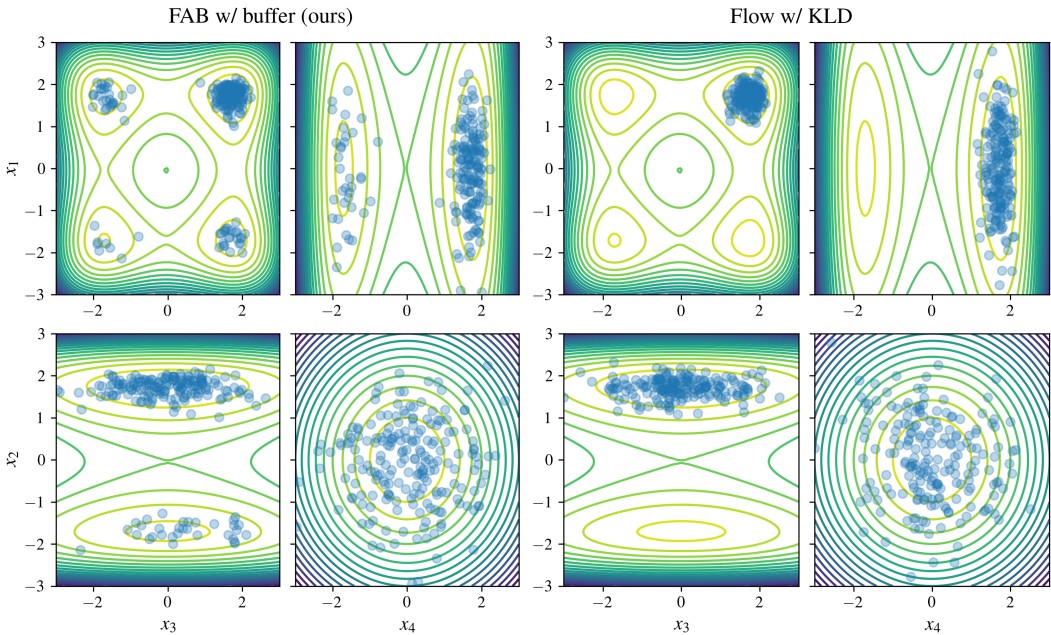

Figure 11: Samples from $q_\theta$ and target contours for marginal distributions over the first four elements of $\mathbf{x}$ in the 32 dimensional Many Well Problem. The flow trained with FAB with buffer (left) captures the target far better than the flow trained by minimizing $\text{KL}(q\|p)$ (right), which misses several modes.

By noting that $x_1$ and $x_2$ are independent, we see that their distribution factorises as $p(x_1, x_2) = p(x_1)p(x_2)$. Thus, the normalizing constant of $p(x_1, x_2)$ is given by the product of the normalizing constants of each marginal distribution. Furthermore, samples from $p(x_1, x_2)$ may be obtained by sampling independently from each marginal. By inspection, we see that the marginal $p(x_2)$ is standard Gaussian. Thus, its normalizing constant is given by $\sqrt{2\pi}$, and samples from this marginal may be obtained trivially. The normalizing constant of the second marginal distribution may be calculated via numerical integration $Z_1 = 11784.51$. We obtain exact samples from $p(x_1)$ by using rejection sampling, a visual summary of this is provided in Figure 13. For the rejection sampling proposal distribution, denoted $q$, we use a two-component Gaussian mixture distribution with mixture weights $(0.2, 0.8)$, means $(-1.7, 1.7)$ and standard deviations equal to 0.5. For the comparison function $kq(x_1)$, we set $k = 3Z_1$ to ensure that $kq(x_1) > p(x_1)$.

### F.3 SETUP

All flow models besides CRAFT have 10 RealNVP layers (Dinh et al., 2017), with a 2 layer (320 unit layer width) MLP for the conditioner. The flow is initialized to the identity transformation and, consequently, $q_\theta$ is initially a standard Gaussian distribution. Training is performed with a batch size of 2048 and using the Adam optimizer (Kingma & Ba, 2015) with a learning rate of $3 \times 10^{-4}$. We clip the gradient norm to a maximum value of 100. For the SNF method, we do 10 Metropolis-Hastings steps every two flow layers. We train all models for $10^{10}$ flow evaluations. For each method, we train 3 models using different random seeds. Results are reported as averages over these three models.

**FAB specific details**: In FAB, the batch sizes for the AIS forward pass ($M$) and sampling from the buffer ($N$) are both equal to 2048. We run AIS using four intermediate distributions with linear spacing. Each MCMC transition is given by a single Hamiltonian Monte Carlo step consisting of 5 leapfrog steps. The momentum variable for HMC is sampled from a standard Gaussian and it is not tuned throughout training. However, an important parameter to tune in HMC is the step size parameter for the leapfrog integrator. We do tune this parameter for each intermediate distribution. This is done by using a parametrization of step sizes that includes coefficients that are specific and

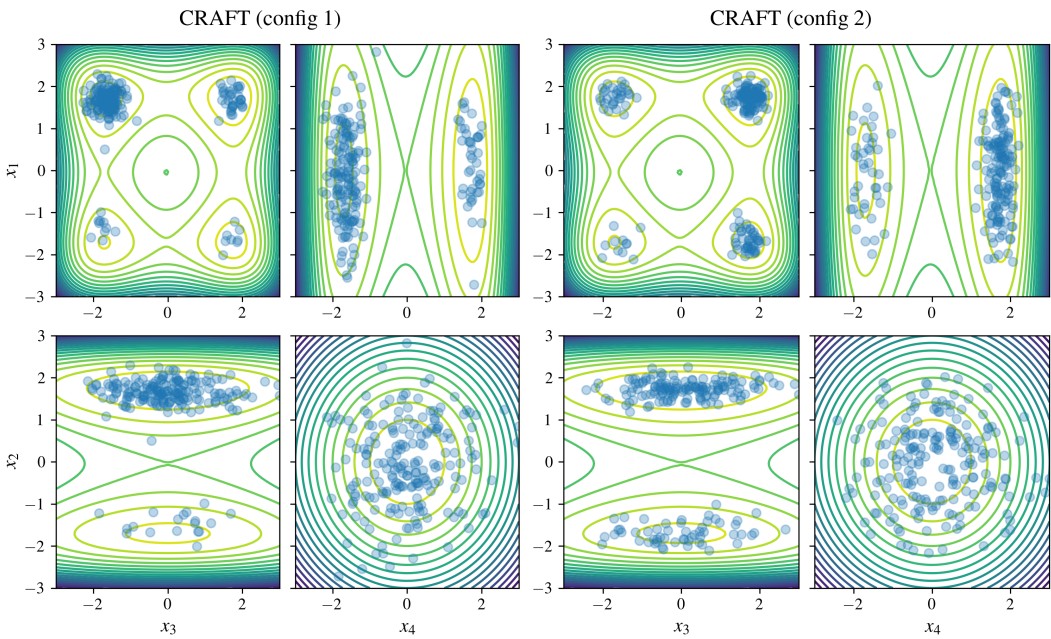

Figure 12: Target contours and model samples for 2D marginals over the first four elements of **x** in the 32 dimensional Many Well Problem for the CRAFT models. CRAFT (config 1) refers to the CRAFT model trained with a similar flow and MCMC config to FAB. CRAFT (config 2) uses the configuration provided in the CRAFT repository, that uses more expressive neural spline flows, and a larger number of intermediate distributions within SMC. Both CRAFT models sample well from the modes in the pairwise marginal distributions.

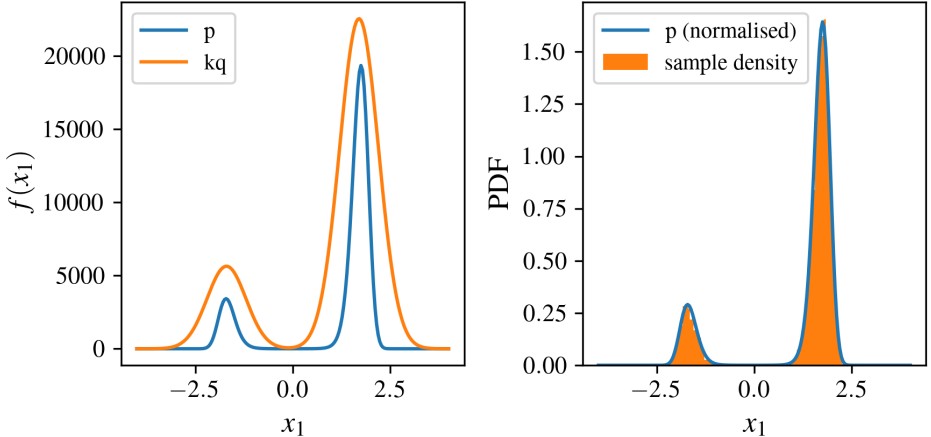

Figure 13: Using rejection sampling to obtain exact samples from the first marginal of the Double Well distribution. (LHS) We see that $kq(x_1) > p(x_1)$. (RHS) Sample density (normalized histogram using 10000 samples) vs. the normalized probability density of $p$. We see that rejection sampling provides exact samples from $p(x_1)$.

shared across intermediate distributions. In more detail, we define the HMC step size for the $n$-th intermediate AIS distribution as $\epsilon_n = \epsilon_{\text{shared}} + \hat{\epsilon}_n$, where we define $\epsilon_{\text{shared}}$ and $\hat{\epsilon}_n$ as follows: $\epsilon_{\text{shared}}$ is a shared parameter across all AIS transitions, and is updated at the transition for every intermediate distribution. This allows for faster adaption of the step size if the step sizes for all transition kernels are "too big" or "too small", which is common at the start of training. $\hat{\epsilon}_n$ is a

parameter specific to each $n$-th intermediate distribution, and is only updated during its specific transition. This allows for $\epsilon_n$ to be tailored to the specific $n$-th intermediate distribution transition. We found this parameter sharing to improve performance in practice. The HMC transition kernel for each intermediate distribution is initialized with a step size of 1.0, where we set $\epsilon_{\text{shared}} = 0.1$ and $\hat{\epsilon}_{1:N-1} = 0.9$. The step size is then tuned to target a Metropolis acceptance probability of 0.65. For the transition corresponding to each intermediate distribution, if the average acceptance probability across the batch is greater than 0.65, we set $\hat{\epsilon}_n = 1.05\hat{\epsilon}_n$, and $\epsilon_{\text{shared}} = 1.02\epsilon_{\text{shared}}$. If the average acceptance probability across a batch is lower than 0.65, we set $\hat{\epsilon}_n = \hat{\epsilon}_n/1.05$, and $\epsilon_{\text{shared}} = \epsilon_{\text{shared}}/1.02$. Adapting shared parameters across the AIS forward pass violates Markov property, as the transitions late in the MCMC chain will have a weak dependency on the earlier transitions. However, the effect of this is minor as the step size changes are relatively small for each run. For evaluation the AIS parameters are frozen, so it respects the Markov property.

In FAB with prioritized buffer, we use a total of $L = 8$ gradient update steps per AIS sampling step. We initialise the buffer with $65,536$ samples from the initialized flow-AIS combination and use a maximum buffer length of $512,000$. We do not use any clipping for $w_{\text{correction}}$.

**CRAFT specific details**: For CRAFT we run our experiments with two different setups, which we simply refer to as CRAFT (config 1) and CRAFT (config 2). The first uses a configuration that is similar to the that of FAB in terms of the flow architecture, and MCMC. This CRAFT model uses 5 temperatures, and uses an auto-regressive affine flow. The second setup uses the configuration provided in the CRAFT implementation at `https://github.com/deepmind/annealed_flow_transport`, which uses neural spline flows, with 11 temperatures (10 flow/MCMC steps) and HMC containing 10 leap-frog steps. We use the default HMC implementation and configuration for CRAFT, which uses a fixed step sizes of 0.3 for the first half of the intermediate distributions and 0.2 for the rest. Both models use 3 flow layers per temperature.

Table 7: Number of flow and target evaluations during training for each method on the Many Well Problem. For SNF and CRAFT we report the number of model forward passes - each of which contains multiple flow transport and MCMC steps.

|  | Number of flow/model evaluations | Number of target evaluations |
|---|---|---|
| Flow w/ ML | $10^{10}$ | $10^7$ |
| Flow w/ $D_{\alpha=2}$ | $10^{10}$ | $10^{10}$ |
| Flow w/ KLD | $10^{10}$ | $10^{10}$ |
| Flow w/ RBD | $10^{10}$ | $10^{10}$ |
| SNF w/ KLD | $10^{10}$ | $2.5 \times 10^{11}$ |
| CRAFT (config 1) | $2.5 \times 10^9$ | $10^{10}$ |
| CRAFT (config 2) | $10^9$ | $10^{10}$ |
| *FAB w/o buffer* | $10^{10}$ | $10^{10}$ |
| *FAB w/ buffer* | $10^{10}$ | $7.2 \times 10^9$ |

### F.4 FURTHER RESULTS

Figure 14 and Figure 15 show contour plots of 2D marginals from the 32 dimensional Many Well target. The contours are for pairs of variables in the first four elements of $\mathbf{x}$ belonging to different copies of the Double Well distribution. Each plot is obtained by scanning two variables while the other ones are kept fixed to zero. These figures also shows samples from each analyzed method. FAB based methods and the flow trained by maximum likelihood place samples at each of the modes in the contour plots, while the other methods fail to do so.

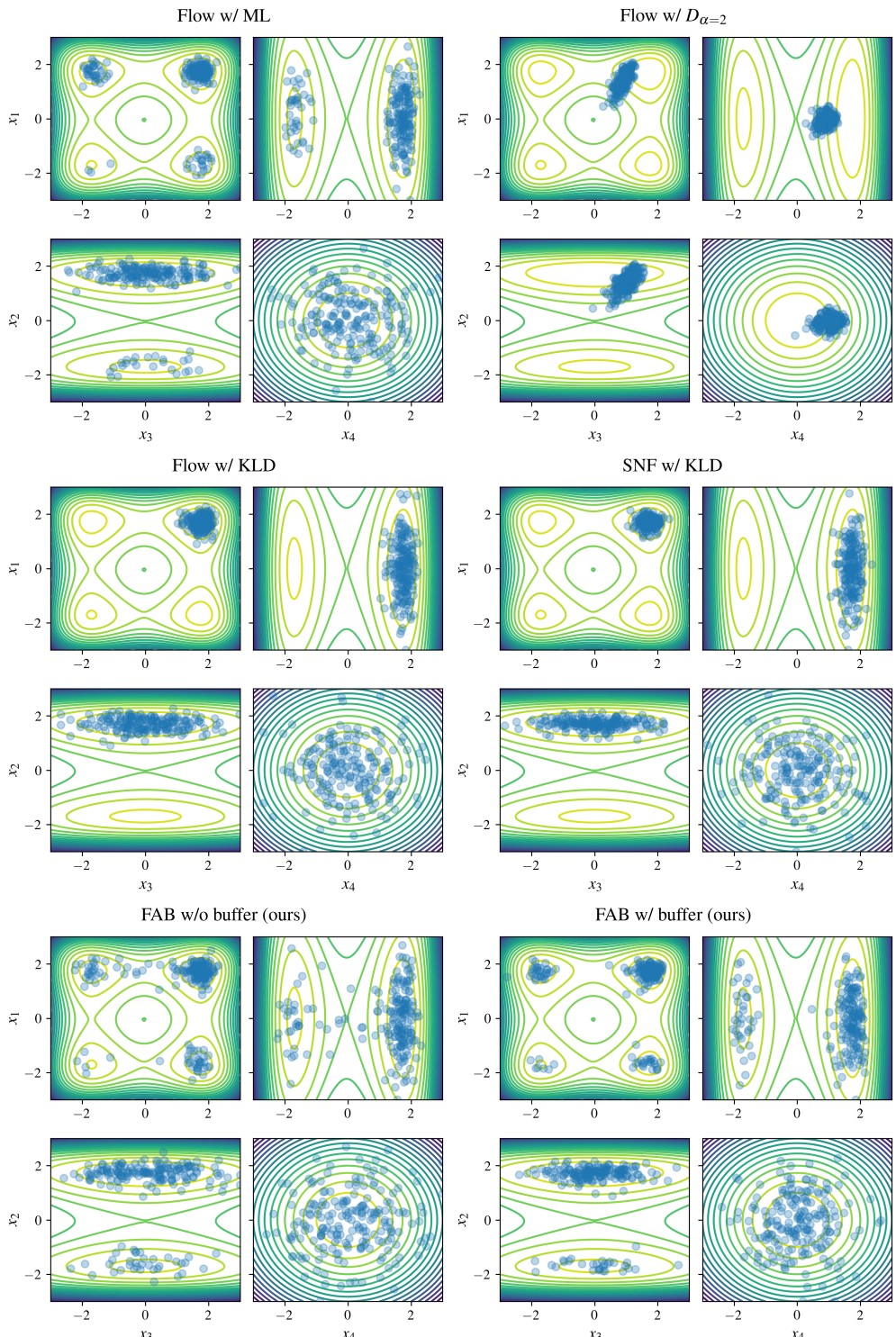

Figure 14: Target contours and model samples for 2D marginals over the first four elements of $\mathbf{x}$ in the 32 dimensional Many Well Problem. The plots are for pairs of variables belonging to different copies of the Double Well distribution. For $D_{\alpha=2}(p\|q_\theta)$ minimization with samples from the flow, we plot results at iteration 56 of training as the final model samples were outside of the plotting regions due to training instabilities.

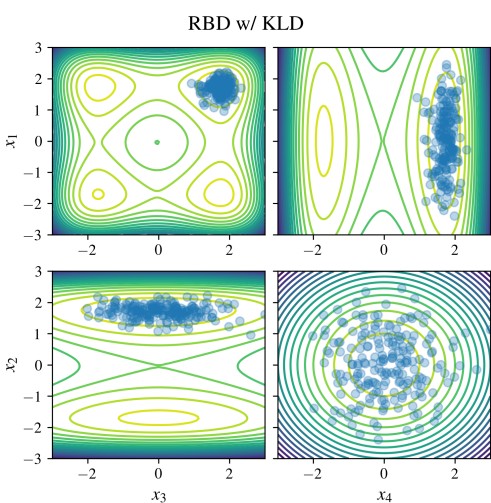

Figure 15: Target contours and model samples for 2D marginals over the first four elements of **x** in the 32 dimensional Many Well Problem for the Resampling Base Distribution (RBD) flow model trained with the KL divergence.

# G  ALANINE DIPEPTIDE EXPERIMENTS

## G.1  SETUP

**Coordinate transformation**     Boltzmann generators usually do not operate on Cartesian coordinates. In particular, Noé et al. (2019) introduced a coordinate transformation whereby a subset of the coordinates are mapped to internal coordinates, i.e., bond lengths, bond angles, and dihedral angles, see also Appendix G.1 and Figure 11 in (Stimper et al., 2022). The internal coordinates are normalized and the respective means and standard deviations for these coordinates are computed on samples from the target distribution generated with MD. For the remaining Cartesian coordinates, principal component analysis is applied to the samples and the six coordinates with the lowest variance are eliminated. The rationale behind this is that the Boltzmann distribution is invariant in six degrees of freedom, i.e., three of translation and three of rotation, and consequently, the corresponding unnecessary coordinates should be removed. However, the mapping of vectors onto a fixed set of principal components is generally neither invariant to translations nor to rotations, and, therefore, the transformed coordinates do not satisfy these invariances. When training Boltzmann generators with samples, this is not a problem since the flow will learn to generate molecular configurations for a specific rotation or translation, but when only using the target distribution to train the flow, the model will spend some of its capacity to sample different translational and rotational states, which is unnecessary since they can easily be sampled independently. This will harm performance.

Instead, we transform all Cartesian coordinates to internal coordinates, which is a representation invariant to translations and rotations. Since we do not want to use MD samples for our model, we use the position with the minimum energy instead as shift and fix values for the scale when normalizing the coordinates. The former can be easily estimated with gradient descent using less than 100 steps. As scale parameters, we used $0.005\,\mathrm{nm}$ for the bond lengths, $0.15\,\mathrm{rad}$ for the bond angles, and $0.2\,\mathrm{rad}$ for the dihedral angles. Coordinates which are treated as circular are not scaled.

**Model architecture**     We use Neural Spline Flows with rational quadratic splines having 8 bins each. The parameter mapping is done through coupling (Durkan et al., 2019). Dihedral angles which can freely rotate, e.g., because it is not a double bond, are treated as periodic coordinates (Rezende et al., 2020). For these coordinates, we use a uniform base distribution, while we pick a Gaussian for the other ones. The flow has 12 layers and the parameter maps are residual networks with one residual block, while the two linear layers in each block have 256 hidden units. The flow layers were initialized in a way that they correspond to the identity map.

One model uses a RBD, which has a residual network with two blocks having 512 hidden units per layer as acceptance function. The truncation parameter is set to the common value $T = 100$.

The models trained with FAB do AIS with 8 intermediate distributions, which are linear interpolations between the flow and target log-densities (Neal, 2001), where the latter one is unnormalized. We use HMC with 4 Leapfrog steps as the MCMC operator in AIS. The same procedure is used when we use AIS in the other baseline models, see Table 10, Figure 20, Figure 21, and Figure 24. The HMC parameters are initialized and tuned using the same procedure as the Many Well problem, see Appendix F.3. The SNF model does additionally 10 Metropolis-Hastings steps every two flow layers. Since this renders sampling from this model already expensive, we do not do AIS with this model.

**Dataset**     Since the energy surface of alanine dipeptide in an implicit solvent has several modes of different sizes with large energy barriers between them, see Figure 19, a very long MD simulation would be required to obtain samples that represent the target distribution well. To get around this problem, which is well known in computational physics and chemistry, we do a replica exchange MD simulation (Mori & Okamoto, 2010), which is a parallel tempering technique (Earl & Deem, 2005). We use 21 replicas starting at a temperature of $300\,\mathrm{K}$ and increasing the temperature by an increment of $50\,\mathrm{K}$. The replicas are exchanged every 200 iterations and use the state at each multiple of 1000 time steps as samples. To reduce the time it takes to generate the data, we run many of these simulations in parallel with different seeds. Since the initial condition is always the same, i.e., the position with minimum energy as it is usually done, we let the system equilibrate for $2 \times 10^5$ iterations and run the simulation subsequently for $2 \times 10^6$ iterations.

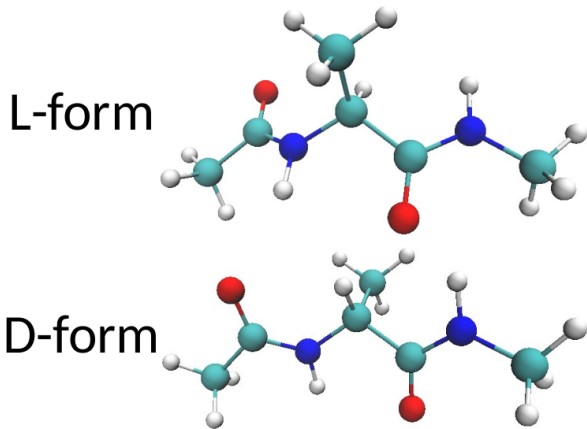

Figure 16: Visualization of alanine dipeptide in its two chiral forms. In nature, we see almost exclusively the L-form and, likewise, we aim to only generate samples of this form.

We split the data into 1) a training set, which consists of $10^6$ samples and is only used to train the baseline flow model with ML; 2) a validation set consisting of $10^6$ samples as well, which is used to find a suitable set of hyperparameters for our experiments; and 3) a test set with $10^7$ samples, which is used to evaluate all models.

To generate the training data alone we had to evaluate the target distribution and its gradients $2.3 \times 10^{10}$ times, which is what we report as cost in terms of target evaluations in Table 8.

**Filtering chiral forms** As mentioned in Section 4.2, alanine dipeptide is a chiral molecule, i.e., it can occur in two different forms that are mirror images of each other, see Figure 16. They cannot be easily converted into each other as this would involve breaking existing and forming new bonds. In nature, we find almost exclusively, while the D-form typically only exists in synthetically created compounds. Hence, whenever alanine dipeptide is considered in the literature, it is almost always as the L-form (Wu et al., 2020; Campbell et al., 2021; Stimper et al., 2022; Dibak et al., 2022; Köhler et al., 2022). Therefore, we aim to train our model on only this form as well. However, since the energy of the molecule does not change when creating a mirror image of it, models trained to approximate its Boltzmann distribution will a priori learn to generate both forms.

The two forms can be separated by using the following procedure. The two chiral forms differ by the positioning of the neighboring atoms at a chiral center, i.e., the center carbon atom. Hence, the difference of the dihedral angles of those atoms with respect to a fixed reference will differ relative to each other, i.e., their difference will change. Hence, we compute this difference and check whether it is close to a reference configuration for which we know that it corresponds to the L-form. As reference configuration, we use the position with minimum energy, which we already determined for the coordinate transformation.

We use this procedure to filter the configurations generated by the flow model during training and included only the samples that correspond to the L-form when computing the loss. Thereby, the model learns to only generate this chiral form.

In Appendix G.2 we will investigate a model trained on both chiral forms and compare it to one that was only trained on the L-form.

**Training** All models were trained using the Adam optimizer (Kingma & Ba, 2015) with a batch size of 1024. A learning rate of $5 \times 10^{-4}$ is initially linearly warmed up over 1000 iterations and decayed with a cosine annealing schedule over the course of training. We use a weight decay of $10^{-5}$ and clip gradients at a value of $10^3$. When training the models with the prioritized replay buffer, we ensured a minimum buffer length of 64 batches and started replacing the oldest samples once its length reached 512 batches.

**Evaluation** To evaluate the models, we draw $10^7$ samples from the models with and without the use of AIS. Since there were some outliers of the importance weights due to flow numerics, we took the $10^3$ highest weights and clipped them to the lowest value in this set to compute the ESS

(Koblents & Míguez, 2015; Dibak et al., 2022). This corresponds to a fraction of $10^{-4}$, or $0.01\%$, of the weights. For the flow trained with the $\alpha$-divergence with $\alpha = 2$, the resulting ESS is close to $10^{-4}$, or $0.01\%$, indicating that the true ESS is even lower. We estimated the Ramachandran plots, i.e., made a histogram of the dihedral angles $\phi$ and $\psi$, see Figure 3a, with $100 \times 100$ bins, and used them to compute the KL divergence between the test samples and the samples from the model. We repeated this with the reweighted samples, whereby we also used the clipped weights. We computed the log-likelihood on the test set with the models, but we did not do so for the SNF, as it only computes importance weights and does not directly estimate the density.

**Computational cost**    The two main contributors to the computational expenses necessary to train flows approximating Boltzmann distributions are the number of evaluations of the flow and the target. Typically, we need both the value and the gradient and, hence, we regard obtaining them as one operation. Moreover, the flows that we use take the same time for sampling and for pure likelihood computation, which is why we regard them as the same operation as well. The flow with RBD is an exception, as sampling from it is more expensive due to learned rejection sampling being used in the base distribution. The number of flow and target evaluations for each model and training procedure are listed in Table 8.

In general, we trained all models using an equal number of flow evaluations, with the exception of SNF. SNF requires a large number of target evaluations due to the sampling layers. Because of this, we reduced the number of flow evaluations done in total by this method.

Table 8: Number of flow and target evaluations needed to train the models. For the flow being trained with ML on MD samples, we report the number of target evaluations that are needed to generate the training dataset with MD. Our methods are marked in *italic*.

|  | Number of flow evaluations | Number of target evaluations |
| --- | --- | --- |
| Flow w/ ML | $2.5 \times 10^8$ | $2.3 \times 10^{10}$ |
| Flow w/ $D_{\alpha=2}$ | $2.5 \times 10^8$ | $2.5 \times 10^8$ |
| Flow w/ KLD | $2.5 \times 10^8$ | $2.5 \times 10^8$ |
| RBD w/ KLD | $2.5 \times 10^8$ | $2.5 \times 10^8$ |
| SNF w/ KLD | $6.0 \times 10^7$ | $3.6 \times 10^9$ |
| *FAB w/o buffer* | $2.5 \times 10^8$ | $2.5 \times 10^8$ |
| *FAB w/ buffer* | $2.5 \times 10^8$ | $2.0 \times 10^8$ |

**Computational resources and runtime**    To generate the MD dataset, we ran the replica exchange MD simulations on servers with an Intel Xeon IceLake-SP 8360Y processors having 72 cores and 256 GB RAM. We used a total of 100 nodes which needed roughly 15.7h each adding up to about 113 kCPUh.

The flow models were trained on servers with an NVIDIA A100 GPU and an Intel Xeon IceLake-SP 8360Y processor with 18 cores and 128 GB RAM. The training time for each model is listed in Table 9. In total, we invested around 1.02 kGPUh in the experiments. Although training the flow with ML is faster than training with FAB, note that generating the data for ML training with MD requires an additional 9.4 kCPUh, which would take 131h when executed on one server.

Table 9: Runtime for training the models on the same server type as specified in the text. Our methods are marked in *italic*.

|  | Runtime |
| --- | --- |
| Flow w/ ML | 13.8h |
| Flow w/ $D_{\alpha=2}$ | 20.0h |
| Flow w/ KLD | 20.0h |
| RBD w/ KLD | 82.5h |
| SNF w/ KLD | 170h |
| *FAB w/o buffer* | 18.8h |
| *FAB w/ buffer* | 15.7h |

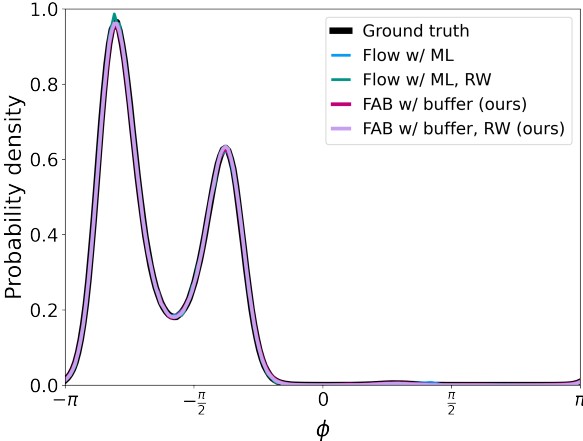

Figure 17: Marginal distribution of the dihedral angle $\phi$ for selected models. The visualized data is the same as in Figure 3b, but here a normal scale instead of a log scale is used for the density.

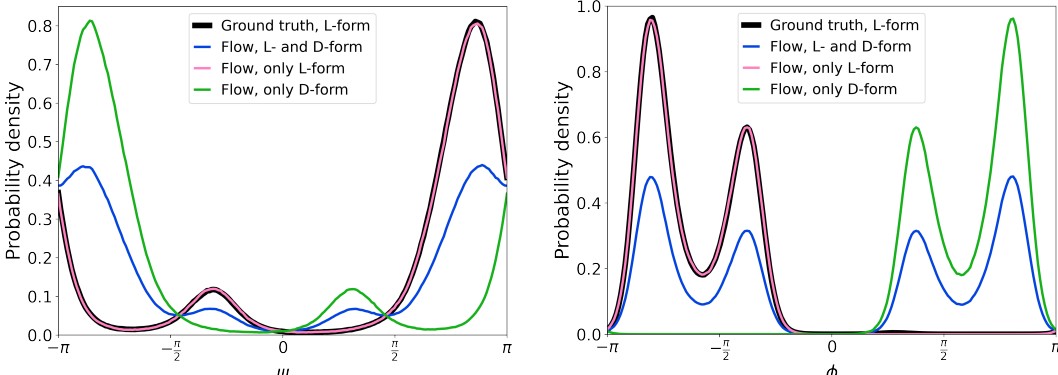

Figure 18: Marginal distribution of the dihedral angles $\psi$ and $\phi$ of a model which has been trained on both chiral forms of the alanine dipeptide, i.e., the L- and D-form. We plot the density obtained with all samples from the model in blue and separate the samples in the two forms, yielding the pink and green curves. The distributions of the two forms are mirror images of each other.

## G.2 FURTHER RESULTS

**Model trained on both chiral forms** To demonstrate the importance of filtering for the L-form during training, we trained a model on both chiral forms using FAB with a replay buffer with the same setting as in the other experiments. We drew $10^7$ samples from the model and found exactly 50% of them correspond to the L- and 50% to the D-form. As can be seen in Figure 18, the marginal distributions of the dihedral angles $\psi$ and $\phi$ for the two forms are mirror images of each other, while the flow model, generating both forms, is a mixture of the two.

The log-likelihood on the test set is 210.80, which is 0.74 less than the corresponding model only trained on the L-form. This is close to $\log(2) \approx 0.69$, i.e. the density is roughly by a factor of two lower, confirming once more that the flow density is a mixture of the density of the two forms.

**Model performance with AIS** As mentioned in the previous section, we do AIS with all our trained models except the SNF, which already involves sampling layers. We adopt the same AIS setting used for training the flow models with FAB, i.e., we use 8 intermediate distributions given by linear interpolations between the flow and the target distributions and sample from them with HMC performing 4 Leapfrog steps. For comparison, we provide the performance when using the untrained base distribution of our flow as proposal for AIS. The results are shown in Table 10. When comparing tables 2 and 10, we observe that AIS improves performance for those models that

approximate the target distribution at least fairly well. Again, the flow trained with FAB with a replay buffer outperforms the baselines.

Table 10: ESS, and the KL divergence of the Ramachandran plots with and without reweighting (RW) for flow models when sampling from them with AIS. The results are averages over 3 runs and the standard error is given as uncertainty. Our methods are marked in *italic* and the highest ESS or log-likelihood or lowest KL divergence values are emphasized in **bold**.

|  | ESS (%) | KLD | KLD w/ RW |
|---|---|---|---|
| Base untrained | $0.013 \pm 0.000$ | $1.96 \pm 0.00$ | $8.5 \pm 0.0$ |
| Flow w/ ML | $11.5 \pm 0.5$ | $\mathbf{(4.92 \pm 2.13) \times 10^{-3}}$ | $(1.88 \pm 0.75) \times 10^{-2}$ |
| Flow w/ $D_{\alpha=2}$ | $0.012 \pm 0.000$ | $2.52 \pm 0.06$ | $11.3 \pm 0.2$ |
| Flow w/ KLD | $80 \pm 8$ | $2.99 \pm 0.19$ | $2.95 \pm 0.19$ |
| RBD w/ KLD | $61 \pm 23$ | $2.84 \pm 0.05$ | $2.81 \pm 0.04$ |
| *FAB w/o buffer* | $70.6 \pm 0.7$ | $(2.65 \pm 0.12) \times 10^{-2}$ | $(2.45 \pm 0.91) \times 10^{-2}$ |
| *FAB w/ buffer* | $\mathbf{96.7 \pm 0.2}$ | $\mathbf{(2.53 \pm 0.41) \times 10^{-3}}$ | $\mathbf{(2.17 \pm 0.15) \times 10^{-3}}$ |

**FAB with varying values of $\alpha$**  In Appendix C.1, we derived a variant of FAB having an arbitrary value for the $\alpha$ parameter of the $\alpha$-divergence. Here, we want to investigate how the performance of models trained with FAB changes as we vary $\alpha$. Therefore, we leave the setup the same as used in Section 4.2 and only changed the $\alpha$ parameter of FAB. The results when using a replay buffer are given in Table 11 and without it they are reported in Table 12. We see that $\alpha = 2$ outperforms the other values in almost all performance metrics, no matter whether a replay buffer is used or not. This justifies our theoretical arguments for picking $\alpha = 2$ empirically.

Table 11: ESS, log-likelihood on the test set, and KL divergence (KLD) of Ramachandran plots with and without reweighting (RW) for each method. Here, all models are trained with FAB using a replay buffer, but we varied the value of $\alpha$ being used, see Appendix C.1. The results are averaged over 3 runs and the standard error is given as uncertainty. Best results are emphasized in **bold**.

| $\alpha$ | ESS (%) | $\mathrm{E}_{p(\mathbf{x})}\left[\log q(\mathbf{x})\right]$ | KLD | KLD w/ RW |
|---|---|---|---|---|
| 0.25 | $0.021 \pm 0.000$ | $-546 \pm 14$ | $5.36 \pm 0.72$ | $10.9 \pm 1.4$ |
| 0.5 | $0.023 \pm 0.001$ | $-606 \pm 20$ | $4.84 \pm 0.95$ | $10.4 \pm 1.9$ |
| 1 | $0.027 \pm 0.000$ | $60.1 \pm 9.0$ | $2.79 \pm 0.81$ | $6.73 \pm 1.76$ |
| 1.5 | $89.4 \pm 0.3$ | $211.49 \pm 0.02$ | $(1.44 \pm 0.90) \times 10^{-2}$ | $(1.52 \pm 0.93) \times 10^{-2}$ |
| 2 | $\mathbf{92.8 \pm 0.1}$ | $\mathbf{211.54 \pm 0.00}$ | $\mathbf{(3.42 \pm 0.45) \times 10^{-3}}$ | $\mathbf{(2.51 \pm 0.39) \times 10^{-3}}$ |
| 3 | $\mathbf{93.9 \pm 1.1}$ | $211.49 \pm 0.03$ | $(2.58 \pm 0.96) \times 10^{-2}$ | $(2.19 \pm 0.82) \times 10^{-2}$ |

Table 12: ESS, log-likelihood on the test set, and KL divergence (KLD) of Ramachandran plots with and without reweighting (RW) for each method. Here, all models are trained with FAB without a replay buffer, but we varied the value of $\alpha$ being used, see Appendix C.1. The results are averaged over 3 runs and the standard error is given as uncertainty. Best results are emphasized in **bold**.

| $\alpha$ | ESS (%) | $\mathrm{E}_{p(\mathbf{x})}\left[\log q(\mathbf{x})\right]$ | KLD | KLD w/ RW |
|---|---|---|---|---|
| 0.25 | $1.7 \pm 0.3$ | $198.71 \pm 0.09$ | $3.94 \pm 0.40$ | $8.45 \pm 0.20$ |
| 0.5 | $9.4 \pm 4.4$ | $-192 \pm 326$ | $7.70 \pm 5.59$ | $7.99 \pm 5.49$ |
| 1 | $15.8 \pm 4.9$ | $210.16 \pm 0.40$ | $(1.09 \pm 0.19) \times 10^{-1}$ | $(5.60 \pm 0.75) \times 10^{-2}$ |
| 1.5 | $34.8 \pm 3.6$ | $210.87 \pm 0.07$ | $(7.72 \pm 0.72) \times 10^{-2}$ | $(4.04 \pm 0.04) \times 10^{-2}$ |
| 2 | $\mathbf{52.2 \pm 1.3}$ | $\mathbf{211.13 \pm 0.03}$ | $\mathbf{(6.28 \pm 0.33) \times 10^{-2}}$ | $\mathbf{(2.66 \pm 0.90) \times 10^{-2}}$ |
| 3 | $24.8 \pm 4.6$ | $210.48 \pm 0.14$ | $(1.31 \pm 0.05) \times 10^{-1}$ | $(3.88 \pm 0.03) \times 10^{-2}$ |

**Ramachandran plots**  Figure 19 shows the Ramachandran plot of the test set and Figure 20, Figure 21, Figure 22, Figure 23, Figure 24, Figure 25, and Figure 26 show the Ramachandran plots of all the models we trained for the first run, including the samples drawn from them via AIS.

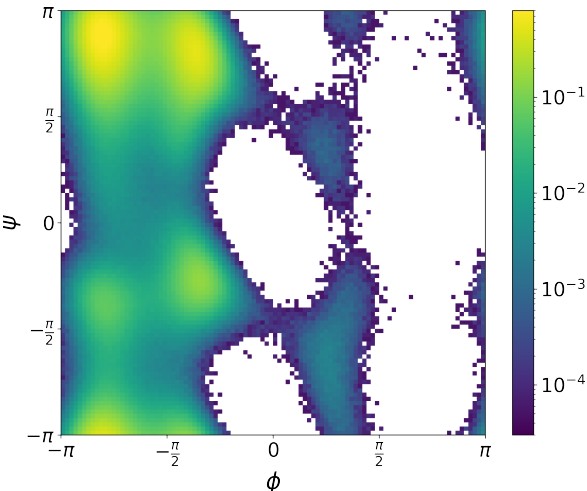

Figure 19: Ramachandran plot of the test data.

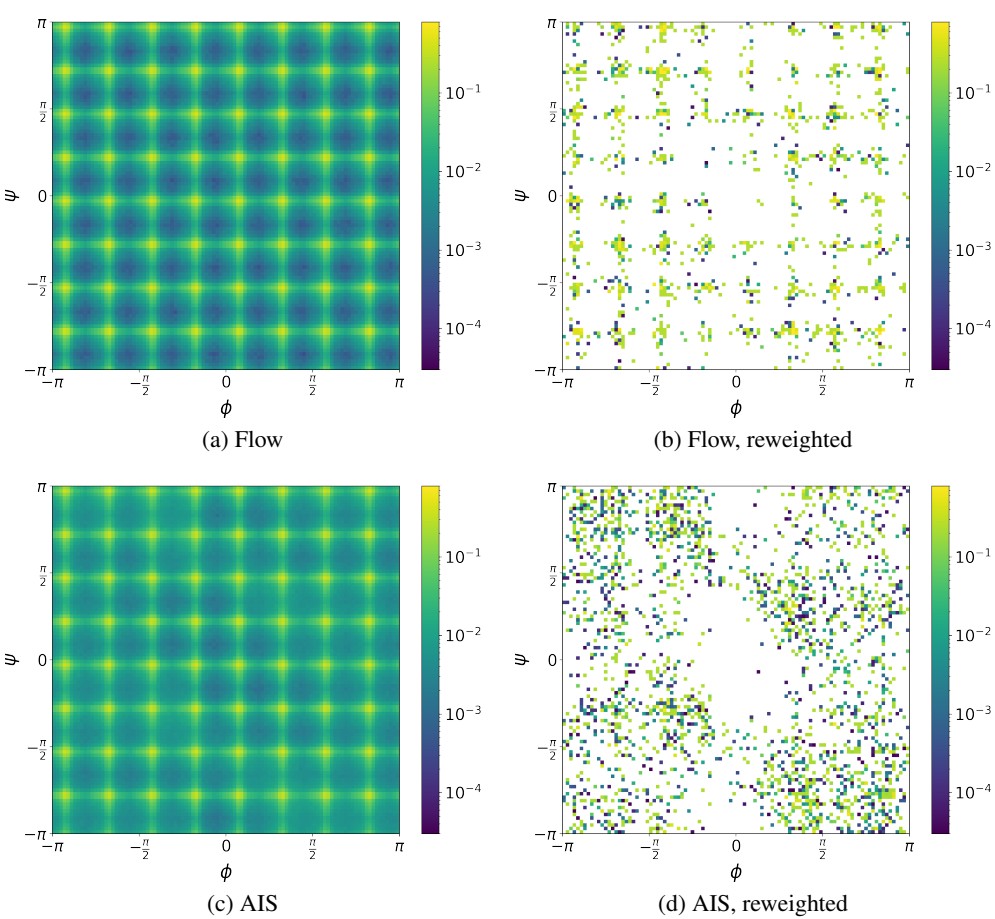

(a) Flow

(b) Flow, reweighted

(c) AIS

(d) AIS, reweighted

Figure 20: Ramachandran plots of a flow trained with the $\alpha = 2$-divergence.

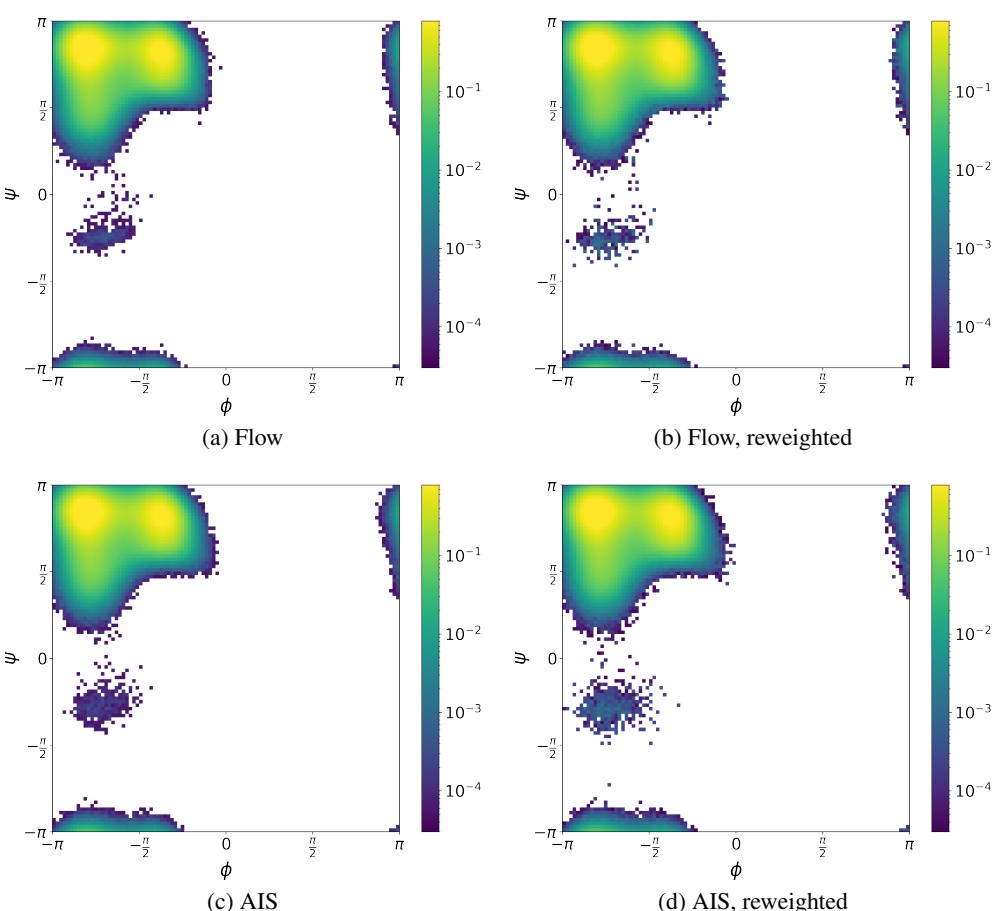

Figure 21: Ramachandran plots of a flow trained with the KL divergence.

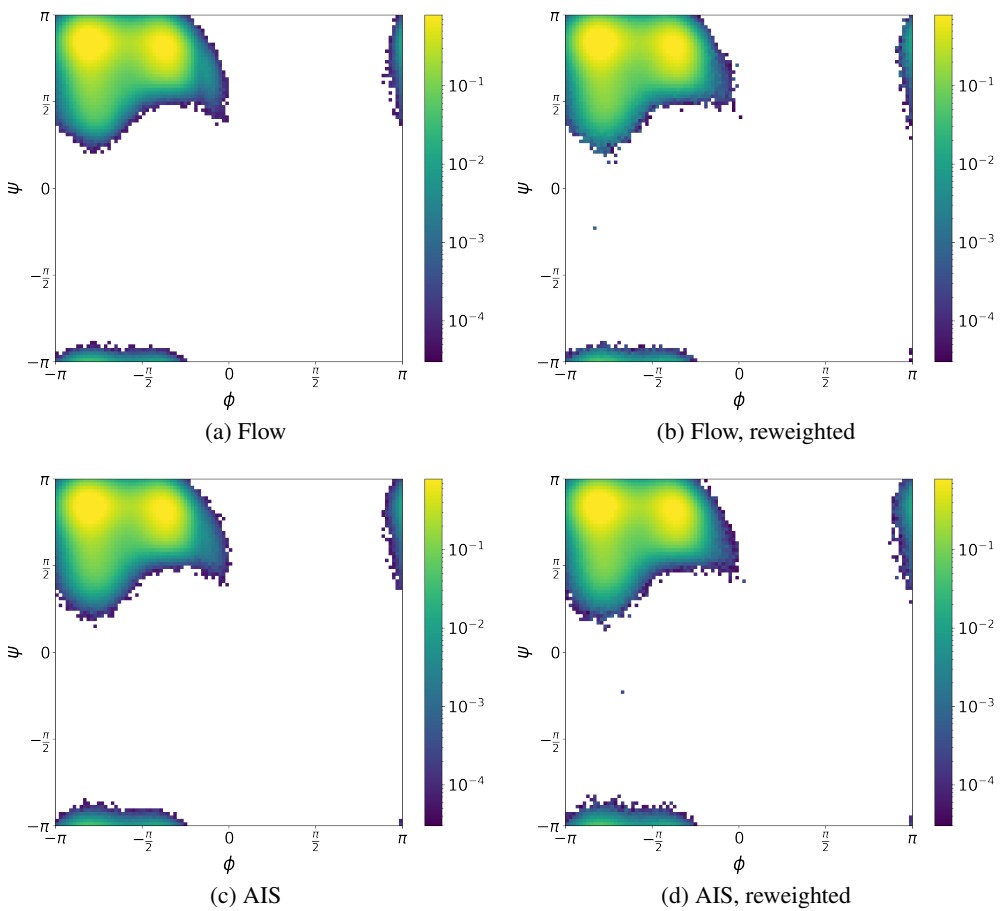

Figure 22: Ramachandran plots of a flow with a resampled base distribution trained with the KL divergence.

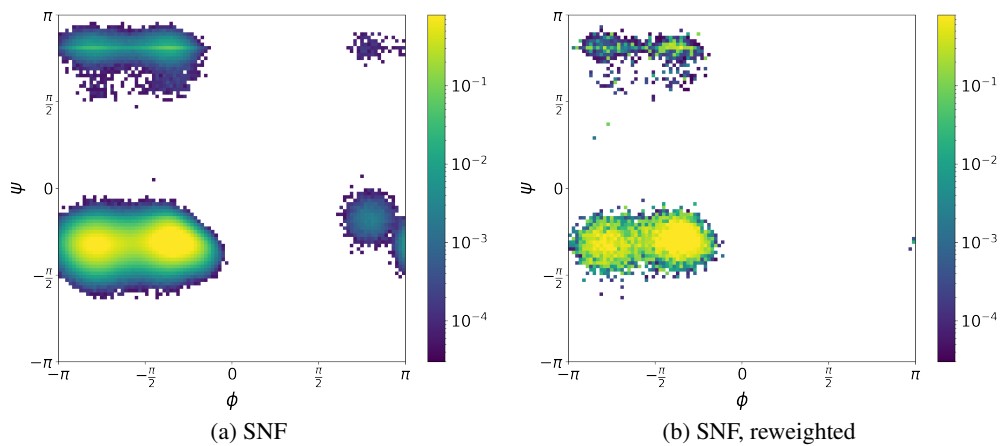

Figure 23: Ramachandran plots of a SNF trained with the KL divergence. Since the SNF already has layers which do sampling, we did not do AIS with it.

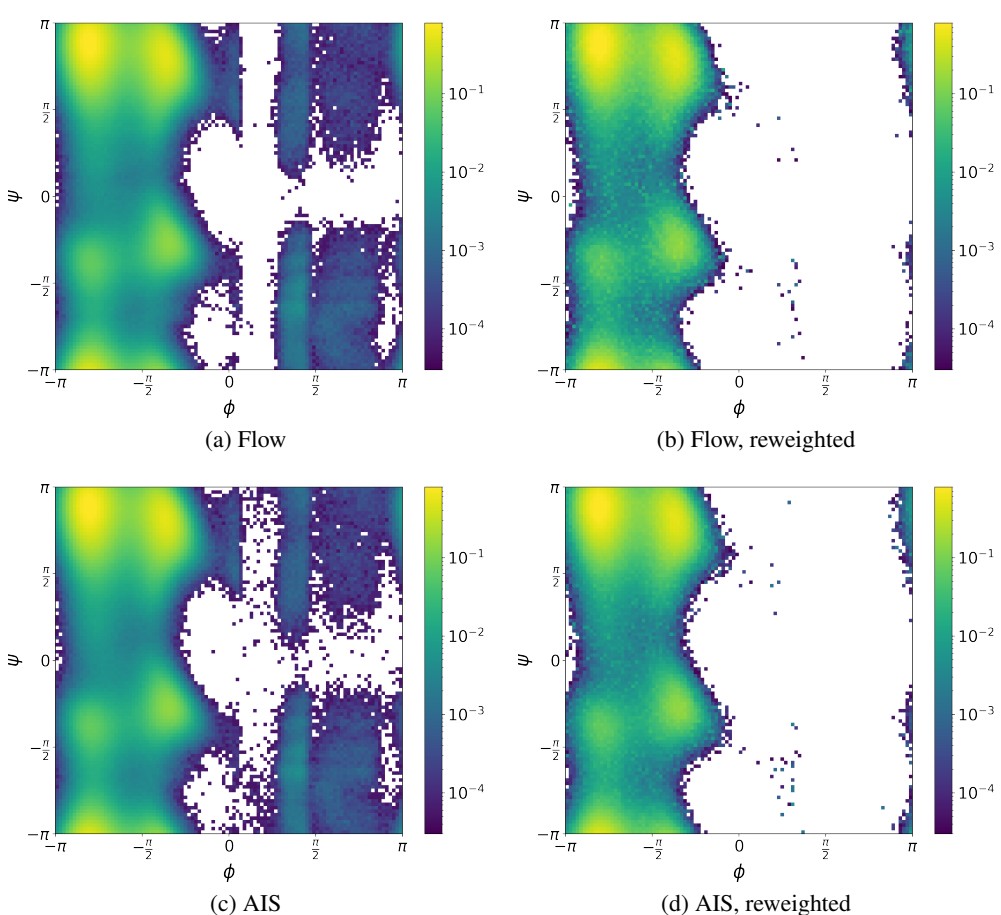

Figure 24: Ramachandran plots of the flows trained with ML.

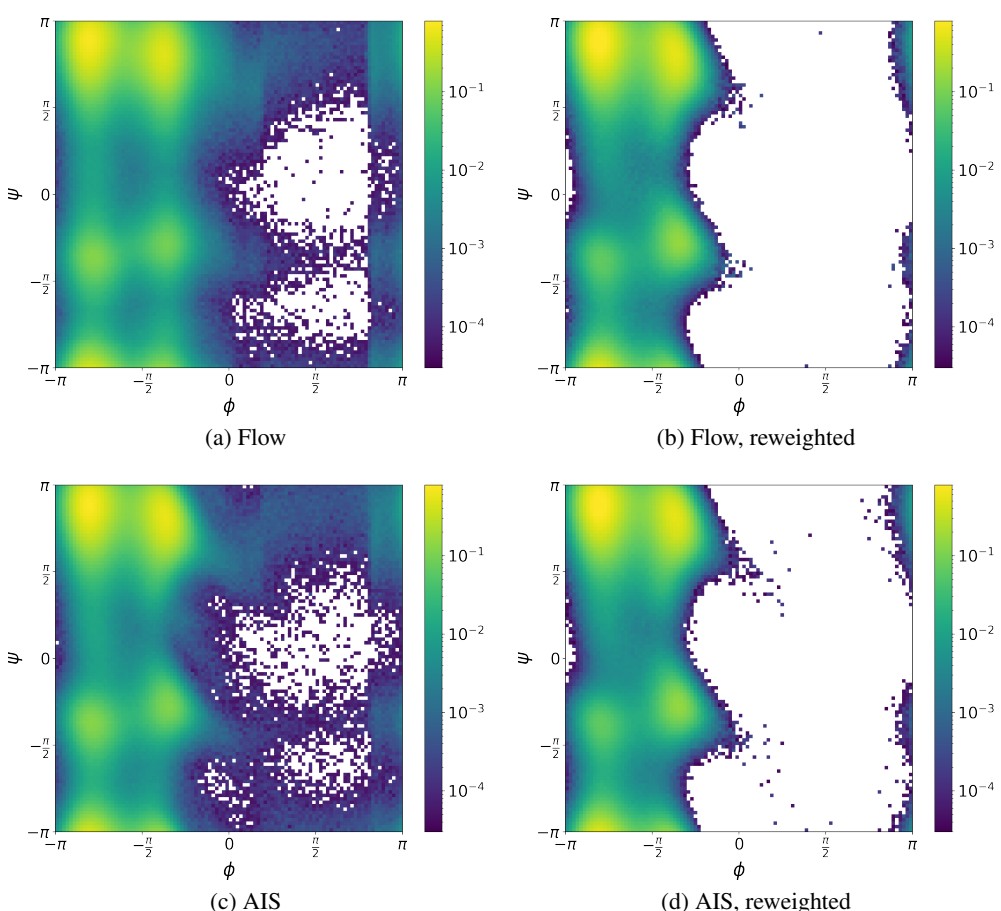

Figure 25: Ramachandran plots of the flows trained with FAB without the use of a replay buffer.

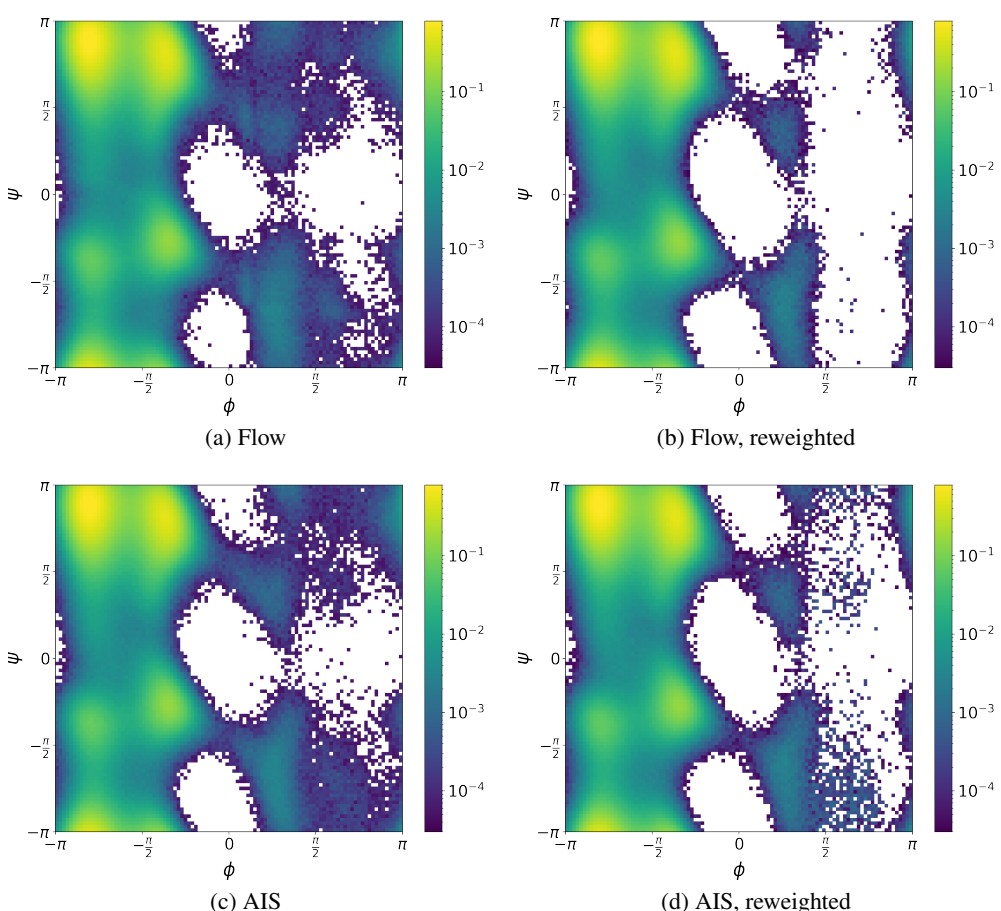

Figure 26: Ramachandran plots of the flows trained with FAB with the use of a replay buffer.

