# OpenReview forum: "Flow Annealed Importance Sampling Bootstrap"
_ICLR.cc/2023/Conference — ICLR 2023 notable top 25%_

### Official Review · Reviewer_xTnc · 2022-10-22

**Confidence:** 3
**Correctness:** 4
**Technical Novelty And Significance:** 3
**Empirical Novelty And Significance:** 3
**Recommendation:** 6

**Clarity, Quality, Novelty And Reproducibility:**

The idea of combining AIS with flows and introducing a replay buffer to reduce cost is new. The paper is well structured and clearly written. The paper provides enough details of the experiments.


**Strength And Weaknesses:**

**Strength**
- The proposed algorithm introduces AIS in the training of normalizing flows. With importance sampling, one is able to sample from the region with larger loss. The computation cost of AIS is reduced by introducing a replay buffer.
- The algorithm is competitive in the reported numerical experiments.

**Weakness**
- There are a lot of tuning parameters in the AIS part of the algorithm: the number of intermediate distributions in AIS, $\beta_i$ for each intermediate distribution; number of MCMC steps used to sample from each intermediate distribution; number of gradient steps per AIS step $L$; $M$. The paper does not discuss how sensitive the performance is to these parameters or how should one choose the parameters.
- It seems that the algorithm should work for other $\alpha$. It is not clear why the paper only considers $\alpha=2$. Some discussion or empirical results with different $\alpha$ would help.
- The AIS step is trying to fit $p^2/q$, which is as challenging as the original problem of fitting $p$. One might wonder how the proposed method compares with pure AIS. For example, flow without AIS has bad performance as shown in Figure 8. So one might ask what's the performance of AIS alone.
- The "bootstrap" in the paper is not the bootstrap commonly seen. There is only one line of explanation in Section 3.1. I think the paper could benefit from more elaborate explanations of why the procedure is called "bootstrap", since it's in the name of the algorithm.

**Minor comments**
- The notation $N$ is abused. In Equation (3), $N$ is the number of intermediate distributions. In other places, $N$ is the number of samples generated by AIS.
- One might expect a plot with $\alpha=2$ in Figure 1 since the whole paper is about $\alpha=2$.

**Summary Of The Paper:**

This paper proposes to train normalizing flows with annealed importance sampling (AIS). The loss function, which is the 2-divergence, is approximated using importance weighted AIS samples targeted at the distribution with the zero-variance IS estimator. A replay buffer is introduced to reduce the computational burden of the AIS step. The algorithm has good performance on approximating a 2-d Gaussian mixture distribution and the Boltzmann distribution of alanine dipeptide.

**Summary Of The Review:**

The paper provides a novel algorithm for training normalizing flows. The algorithm is competitive in the reported experiments. The writing is clear. The weakness includes the complexity of tuning parameters and the lack of comparison with the baseline of AIS alone.

---

> ### Author Response · Authors · 2022-11-11
> **Reply to Reviewer xTnc**
>
> The number of AIS distributions improves the quality of the gradient estimation at each step, at the cost of extra compute.
> In general we aimed to pick a small number of intermediate distributions to keep the cost per iteration low.
> We found that the training procedure was robust even when using a low number of AIS distributions.
> We used only 1 intermediate distribution for the GMM problem, and 4 for the Many Well and Dipeptide problems.
> We agree that obtaining a principled way of selecting these hyper-parameters would be useful.
> One step we make towards this is the analysis of the number of intermediate distributions required as we scale the dimensionality of the problem in Appendix C.2 - where we suggest increasing the number of AIS distributions linearly with the dimensionality of the problem.
> However it would be advantageous to study this in further work.
>
> We use a relatively simple version of AIS/MCMC and cite literature for more advanced ways of selecting/optimizing these hyper-parameters.
> In general we expect that these more advanced techniques should further improve FAB.
> For example for $B_i$ we chose linear spacing for simplicity.
> [(Brekelmans et.al 2020)](https://arxiv.org/abs/2012.07823) provides a more advanced techniques for selecting this.
> Also notably [(Doucet et.al 2022)](https://arxiv.org/abs/2208.07698) provides a method of tuning the full extended target distribution of AIS.
>
>
> In the prioritized experience replay paper [(Schaul et al., 2016)](https://arxiv.org/abs/1511.05952) they sample each experience 8 times on average, which is the value of $L$ that we used for the alanine dipeptide problem.
> For the GMM problem in some preliminary experiments we found that setting $L=4$ tended to perform better so we left it at this value.
> In general we expect that setting L=8 should provide a good choice.
> Increasing $L$ decreases the amount of computation spent on AIS for training, however setting $L$ high may result in the flow "overfitting" on the buffer experience.
>
> AIS requires a proposal. When trying to benchmark the performance of AIS alone, we can use for instance the base distribution as proposal. For alanine dipeptide, this is a uniform distribution for the dihedral angles along rotatable bonds and Gaussian distributions for all other coordinates. With this setting, we only achive an ESS of $1.26\times10^{-4}$ and the KL divergence of the Ramachandran plots is $1.96$, which is comparable to the worst models we considered.
>
> The meaning of the word *bootstrap* we had in mind is “using one's existing resources to improve oneself”. This is a generic interpretation of the word bootstrap. We acknowledge that the word is overloaded and, hence, can be confusing. In statistics, this term is often used for a resampling method, and there exist other interpretations in other disciplines.
> FAB is a bootstrapping method according to the generic notion of the word since it uses the flow in combination with AIS to estimate a loss in order to improve the flow. While improving the flow, we get a better initial distribution for AIS, and so on.
> We included a definition of the word in the introduction of our manuscript.
>
> We have applied the fix for the abuse of notation with N - thanks!

---

> > ### Author Response · Authors · 2022-11-17
> > **Clarification**
> >
> > Reviewer 8zPW [noted](https://openreview.net/forum?id=XCTVFJwS9LJ&noteId=aaDlKD_3bE) that our response here has an inaccuracy as the method from [Brekelmans et.al 2020](https://arxiv.org/abs/2012.07823) is not for selection of $\beta_i$. Thus we add the following correction:
> >
> > The following text
> > > For example for $\beta_i$ we choose linear spacing for simplicity. [Brekelmans et.al 2020](https://arxiv.org/abs/2012.07823) provides a more advanced techniques for selecting this.
> >
> > should instead be
> > > For example, for the intermediate distributions we use a geometric path; $\log p_i(\mathbf{x}) = \beta_i \log p_0(\mathbf{x}) + (1 - \beta_i) \log p_M(\mathbf{x})$, with linear spacing between 0 and 1 for $\beta_i$ which we choose for simplicity. An alternative is to use the geometric path with $\beta_i$ spaced geometrically rather than linearly. [Masrani et.al 2020](https://arxiv.org/abs/2107.00745) provides a more advanced/general technique for selecting the intermediate distributions than the geometric path.

---

### Official Review · Reviewer_tRrj · 2022-10-22

**Confidence:** 5
**Clarity, Quality, Novelty And Reproducibility:** The paper is clear.
**Correctness:** 4
**Technical Novelty And Significance:** 2
**Empirical Novelty And Significance:** 2
**Recommendation:** 5

**Strength And Weaknesses:**

Strength: The paper is written well with clear mathematics.

Weakness: The authors need to demonstrate (motivate) why alpha=2 works better.  Does the author try other f-divergences?

**Summary Of The Paper:**

The authors use the 2-divergence (Chi^2) as the objective function for sampling target distribution. And they parameterize the distribution by a normalizing flow. They use importance sampling ideas to approximate the Chi^2 divergence. By solving the proposed optimization method numerically, they directly simulate the Gibbs distribution from Molecular Dynamics. Numerical results demonstrate the simplicity of the method.

**Summary Of The Review:**

Overall, the paper is written well. Some analytical examples of Chi^2 divergences can be studied. This could explain or motivate why it works better in numerics.

---

> ### Author Response · Authors · 2022-11-11
> **Reply to Reviewer tRrj**
>
> We thank you for your review and appreciate that you like the clarity of our manuscript.
>
> Your main criticism regarding the justification for using the $\alpha=2$-divergence is shared by several of the other reviewers, which is why we addressed them in a generic reply. If you have any other concerns, please feel free to leave a comment here.

---

### Official Review · Reviewer_1QZD · 2022-10-27

**Confidence:** 3
**Correctness:** 4
**Technical Novelty And Significance:** 3
**Empirical Novelty And Significance:** 2
**Recommendation:** 6

**Clarity, Quality, Novelty And Reproducibility:**

The question of how to estimate the flow-based method is important, and the proposed method is clear.

**Strength And Weaknesses:**

1. In section 3.1, they claim “p^2/q is the minimum variance importance sampling distribution for D_{α=2}(p||q)”. Can the authors explain further about this point?
2. In Many Well Experiments(Appendix E), it seems that their method can not beat the Flow w/ ML method based on Table 4. The Ep(x)[log q(x)], Mean log q (xmodes ) and DKL[p||q] results of those two methods are exactly the same.
3. Could the authors provide computation time for different methods?


**Summary Of The Paper:**

The normalizing Flow-based method is used to estimate complex densities.  However, current methods for training flows have some drawbacks, including expensive computing with MCMC simulations. In this work, the authors propose low AIS Bootstrap method to generate samples with a theoretical and numerical guarantees.





**Summary Of The Review:**


Overall, the writing is easy to follow and the proposed method seems to work well in 2-d case.
However, the method does not demonstrate improvements over the baseline method in higher
dimensional cases.

---

> ### Author Response · Authors · 2022-11-11
> **Reply to Reviewer 1QZD**
>
> Multiple reviewers also asked about the choice of $g=p^2/q$ - so we have added a comment describing the motivation behind this choice, and provide links to further analyses of alternatives to this that we included in the Appendix. We refer you to this comment.
>
> Training by ML requires being able to sample from the target. However, we are interested in the case when we cannot sample from the target. Hence, this is not for direct comparison, as the methods apply in different circumstances. Hence we use the --- in the tables to show that this method is "separate".
> In the case of the alanine dipeptide example, we have to run expensive MD simulations to obtain samples.
> After accounting for the target evaluations needed for MD, we see that training by ML is more expensive.
> The most common approach from literature for training without samples is the "Flow w/ KLD" baseline, see [(Noé et al., 2019)](https://www.science.org/doi/10.1126/science.aaw1147), that our method strongly outperforms in all of the experiments.
>
> Regarding the computational cost, the most relevant quantities are the number of flow and target evaluations, which are done during training. We listed them in the appendix, see e.g. Table 3, Table 5, and Table 6. How these numbers translate to a runtime depends on the available resources and the implementation. For the dipeptide experiments, we already gave a rough overview in the paragraph **Computational resources** in Appendix F.1. However, we agree that it is interesting to see how this information translates to the runtime for all methods. We report them in the table below, which we also added to our manuscript in Appendix F.1.
>
> | Training method | Runtime |
> |-----------------|---------|
> | Flow w/ ML      | 13.8h   |
> | Flow w/ $D_{\alpha=2}$ | 20.0h   |
> | Flow w/ KLD     | 20.0h   |
> | RBD w/ KLD      | 82.5h   |
> | SNF w/ KLD      | 170h    |
> | FAB w/o buffer  | 18.8h   |
> | FAB w/ buffer   | 15.7h   |
>
> Although training the flow with ML is faster than training with FAB, note that generating the data for ML training with MD requires an additional 9.4 kCPUh, which would take 131h when executed on one server.

---

### Official Review · Reviewer_8zPW · 2022-11-03

**Confidence:** 4
**Correctness:** 4
**Technical Novelty And Significance:** 3
**Empirical Novelty And Significance:** 4
**Recommendation:** 8

**Clarity, Quality, Novelty And Reproducibility:**

Most comments here are with respect to clarity.   Novelty and quality of the work is high and the appendix appears to have extensive experimental details, although I did not review.


**Sec 3**
On first pass, I found myself conflating two notions of 'minimum variance', which I think could be clarified by the exposition.    Minimizing the α=2 divergence corresponds to minimizing variance of IS weights when sampling under $q$, but $g$ is the 'minimum variance' is the minimum variance proposal for estimation of the α=2 divergence.     One way to emphasize / summarize might be: at test time, we'd like to sample from flow model $q$ (=> minimize α=2), but α=2 is hard to estimate during training (=> sample from $g$).

* The last sentence under Eq. 2 could more clearly specify that samples are drawn from $q$.

* In Sec. 3.1, I think it would be useful to discuss alternatives to the minimum variance proposal $g$, especially since sampling from $q$ and $p$ directly end up forming baselines later in the paper.    Only after rejecting these options might the reader fully appreciate the choice of $g$!



**Experiments**
I found the experiment sections difficult to follow.   It may be useful to demarcate "experimental design" details and "analysis of results".

* In particular,  I left Sec. 4.2 with very little sense for what Φ and ψ are, how they fit into the sampling/estimation task, etc.   More high-level definition of the problem (perhaps with fewer preprocessing or experimental details, for space) would be useful here.

* Please clarify which KL divergence is being calculated in Table 2 ($KL[q:p]$?).    I assume reweighting in Fig 4 is for samples from q, using importance weights wrt p (?)


Minor comments:

* A concrete expression for estimating expected values using AIS weights would be useful in Sec 2 or App A (e.g. to ease Eq 12).
* Probably best to use to term "perfect transitions" for exact, independent AIS samples


**Strength And Weaknesses:**

**Strengths:**  The paper is well-motivated in proposing a method for training mass-covering flows without access to target samples.   I am happy to see work utilizing the importance sampling variance-minimization interpretation of the α=2 divergence.

The authors essentially solve estimation of the Boltzmann dist. for alaine dipeptide molecule.   I do not fully appreciate the significance, but this does represent a 'state-of-the-art'-style result, in which the motivating aspects of the proposed method figure prominently.


**Weaknesses:**
One question is to what extent the proposed α=2 objective is preferred to MLE with $KL[p : q]$ in general / practice.   Similar derivations should apply using AIS to sample from p and minimize $KL[p : q]$(the only thing that should change is the AIS target and importance weights, if I understand correctly).

Results are nearly identical for α=2 vs. MLE with exact or MD samples (both essentially solve the problems), so it doesn't seem absolutely necessary to run additional baselines (or to open up the discussion to less justified values of α).     Still, the exposition should state the generality of the AIS bootstrap approach in applying to these divergences, especially since accurate sampling for higher values of α should be more challenging in practice.



**Summary Of The Paper:**

This paper proposes a method for training normalizing flows using a mass-covering objective which does not require samples from the target distribution.   Existing methods often optimize a mode-seeking objective when target samples are not available, although the experiments section highlights applications (such a learning mixtures of Gaussians or a molecular Boltzmann distribution) where mode-seeking optimization fails to match a multi-modal target.

The current work overcomes these limitations by evaluating flow density training gradients on approximate samples from the the optimal proposal for estimating the α-divergence with α=2.    Minimizing this divergence as a function of q corresponds to minimizing the variance of the importance weights when sampling under q.   Annealed Importance Sampling (with a replay buffer) is used to transform samples generated by the flow into samples from the optimal α=2 proposal, which emphasizes regions in the sample space where the current flow model does not match the target.



**Summary Of The Review:**

I am happy with this paper overall.   The experiments show success in proposing a mass-covering loss for normalizing flow training which can be used when target samples are not available.    Some concerns regarding presentation have been raised above.

---

> ### Author Response · Authors · 2022-11-11
> **Reply to Reviewer 8zPW**
>
> Thank you for the detailed review!
>
> You are correct that we can use AIS to estimate the forward KL divergence using $p$ as the target distribution.
> We note that $\mathrm{KL}[p || q]$ is a special case of alpha divergence where $\alpha=1$.
> We have now added a version of FAB that works for arbitrary values of $\alpha$to Appendix B.
> We have also added experiments for FAB with various values of $\alpha$ including $\alpha=1$ on the dipeptide problem to the Appendix.
> We found that $\alpha=2$ worked best.
> Questions regarding the choice of $\alpha=2$ were common across reviewers so we responded to these together.
> We refer you to this comment for further discussion.
>
> We agree that it is important to make the distinction clear between the two different notions of minimum variance and that it can be a bit confusing to the reader. We have updated the paper to include the 'test time' vs 'training time' distinction that you mentioned.
>
> We are sorry that the definition and role of $\phi$ and $\psi$ are still unclear. The molecule is parametrized in internal coordinates, i.e. bond lengths, bond angles, and dihedral angles. These coordinates are defined e.g. in the *Materials and methods* Section [(Noé et al., 2019)](https://www.science.org/doi/10.1126/science.aaw1147) or Appendix G.1 and Figure 11 in [(Stimper et al., 2022)](https://proceedings.mlr.press/v151/stimper22a/stimper22a.pdf). Two of these dihedral angles are $\phi$ and $\psi$. The marginal distribution of $(\phi, \psi)$ represented as a histogram given samples from the Boltzmann distribution (or a model of it) is called the Ramachandran plot. It plays an important role in analyzing proteins, as it can be used to determine how a protein locally folds.
> The KL divergences are computed by getting the Ramachandran plot from the test set as well as from samples from the desired model and then estimating it numerically on the histogram grid. As you said, reweighting is done through the importance weights given the model and the target. Thereby, we generated the rightmost plot in Figure 4, but we also computed the KL divergences of the Ramachandran plot histograms obtained with reweighting with respect to the ground truth.
> I hope we clarified what $\phi$ and $\psi$ are and which role they play in our analysis. We extended their description in our manuscript as well.
>
> Regarding the minor comments, we have added expressions for estimating expectations with AIS to both Section 2 and Appendix A. Moreover, we added "perfect transitions" for exact independent samples for MCMC transitions in AIS.

---

> > ### Comment · Reviewer_8zPW · 2022-11-16
> > **thanks for addressing concenrs**
> >
> > Thank you for the comprehensive reply!  This explanation of the Alanine Dipeptide molecule was particularly useful.   If the authors agree, it seems useful to include some of this detail in place of discussing preprocessing steps and molecule chirality in the first 2 paragraphs of Sec 4.2.    Thank you for the additional experiments as well.
> >
> > Beyond-the-scope of current review comments :
> >
> > - If extending the method to differentiable AIS methods, it may be interesting to consider the SNIS weights in the surrogate loss (Eq. 7) as arising from an IWAE-style objective (e.g. [Zhang et. al 2021 Eq. 23](https://arxiv.org/pdf/2107.10211.pdf))
> > $$ \mathcal{S}(\theta) = \log \frac{1}{K} \sum_k w_{AIS}^{(k)}(x_{0:T}, \theta) \implies \nabla_{\theta}  \mathcal{S}(\theta) = \frac{w_{AIS}^{(k)}(x_{0:T}, \theta) }{\sum_k w_{AIS}^{(k)}(x_{0:T}, \theta) } \nabla_{\theta}  \log w_{AIS}^{(k)}(x_{0:T}, \theta) $$
> >
> >
> > - In the response to Rev. xTnc, note that  [Brekelmans et.al 2020](https://arxiv.org/abs/2012.07823) (conference version [Masrani et. al 2021](https://arxiv.org/abs/2107.00745)]) changes the path of intermediate densities (moving away geometric averaging), and does not concern the spacing of densities along the geometric path.   For scheduling, see e.g. [Grosse et. al 2013](https://www.cs.toronto.edu/~rgrosse/nips2013-moment.pdf) or [Kiwaki 2015](https://arxiv.org/pdf/1502.05313.pdf), although these approaches may not be very practical.   I don't think this is a weakness of the current paper.

---

> > > ### Author Response · Authors · 2022-11-17
> > > **Thanks**
> > >
> > > Indeed that is relevant - thanks for the pointer! We have added a comment clarifying our response to Rev. xTnc.

---

### Official Review · Reviewer_itRu · 2022-11-04

**Confidence:** 3
**Correctness:** 4
**Technical Novelty And Significance:** 4
**Empirical Novelty And Significance:** 4
**Recommendation:** 8

**Clarity, Quality, Novelty And Reproducibility:**

Clarity: I find this paper to be well-written and clear.
Quality: See strength 4,5,6
Novelty: See strength 1,2,3
Reproducibility: The code is included, going through the code it seems all experiments are included. But I don’t have enough time to run the code unfortunately. However based on going through the appendix, it seems the results will be reproducible.


**Strength And Weaknesses:**

Strength:

1. I find FAB to be well-motivated. I wasn’t familiar with Boltzmann generators but I find the adoption of AIS to train Flows for this problem to be clever. AIS samples are usually pretty good, but mostly it was only used to estimate the partition function. I have previously thought that the AIS samples could be put to better use, and in this paper the alanine dipeptide problem is a perfect use case because we have access to only the unnormalized density, but to train an explicit generative model we need samples. Yet directly training with the ML objective has various problems, AIS comes to the rescue.

2. I find the use of Alpha divergence with alpha=2 to be clever and appropriate as it encourages mode covering and minimizes importance weight variance at the same time.

3. I find the decomposition of HMC step sizes to be clever (In Appendix E.3). I don’t recall seeing it in prior works, is it a novel contribution by the authors?

4. The FAB method is theoretically sound.

5. It seems authors have sufficiently explored various ways to empirically improve FAB, tricks including the replay buffer (and various nuanced decisions went into improving that including using mini batching with probability proportional to old AIS weights and reweighting etc) are well-motivated and empirically shown to be useful.

6. As a result, the FAB method is empirically strong and convincing. On the toy example, Figure2 is convincing at showing the effectiveness of FAB with buffer, without ground truth samples, it performs as well as Flow trained with ML objective with access to ground truth samples. Similarly Table 1, Table 2 and Figure 3, Figure 4 show strong evidence of the effectiveness of the method.

Weakness:

1. The step sizes tuning for the HMC is tricky, because to conform to mathematical correctness, it is necessary to load the step sizes from a given run and repeat the same run with different random seeds, so that the Markovian property is not violated. I don’t think the authors did this? Granted it probably doesn’t make a big difference empirically, but authors could consider at least verifying how much of a difference this would make. For more details, see Appendix C.3 of https://arxiv.org/pdf/2008.06653.pdf


**Summary Of The Paper:**

This paper proposed a method, Flow AIS Bootstrap (FAB) that uses AIS to train flows to approximate complex, unnormalized distributions. Authors used. Alpha divergence with alpha=2 as the objective as it encourages mode covering and minimizes importance weight variance. Authors further reduce the computational cost of FAB by introducing a replay buffer with various tricks, and the authors show empirical evidence of the effectiveness of FAB on toy distributions and on the more sophisticated Boltzmann distribution of alanine dipeptide.



**Summary Of The Review:**

In summary, I think this paper is novel both in terms of the method itself and in terms of the specific use case. And authors have shown strong empirical evidence of the superiority of the method.  I would recommend accept.

---

> ### Author Response · Authors · 2022-11-11
> **Reply to Reviewer itRu**
>
> Thank you for the detailed review. We are pleased that you appreciate our choice of setting $\alpha=2$, which we further empirically justify in a comment to all reviewers as well as in the Appendix of our updated manuscript.
>
> We agree that adapting shared parameters across AIS within a forward pass violates Markov property.
> The effect of this is probably minor as the step size changes are relatively small for each run.
> Also for evaluation we always fix the parameters of AIS, so this is only present during training - this similar to the "preliminary" versus "formal" run in the paper that you linked to.
> We have added a note in the Appendix to clarify this for the reader.

---

### Official Review · Reviewer_A5bQ · 2022-11-05

**Confidence:** 3
**Correctness:** 4
**Technical Novelty And Significance:** 4
**Empirical Novelty And Significance:** 4
**Recommendation:** 6

**Clarity, Quality, Novelty And Reproducibility:**

Overall this paper is clear to read, well written and the novelty is high. The code is included, so I assume that the reproducibility should be high.

**Details Of Ethics Concerns:**

I like this paper overall. Clearly sufficient novelty and the experimental part is also convincng. Therefore, I would recommend accept.

**Strength And Weaknesses:**

Strength: I like the idea of using AIS to cover for regions in the sample space where the current flow model struggle. The experimental results also supported the claim.

Weakness: My concern is over the particular choice of α=2 objective. Why does minimizing variance indicate that this objective works better for the flow?

**Summary Of The Paper:**

This paper proposes a strategy to augment flows with annealed importance sampling (AIS) with the objective to minimize importance weight variance. AIS is responsible to generate samples in regions where the flow struggles. The proposed FAB was shown to produce
better results than training via maximum likelihood on MD samples with less computational demand.

**Summary Of The Review:**

In summary, I think this paper is novel both in terms of the method itself and in terms of the specific use case. And authors have shown strong empirical evidence of the superiority of the method. I would recommend accept.

---

> ### Author Response · Authors · 2022-11-11
> **Reply to Reviewer A5bQ**
>
> We thank you for your review and are glad that you appreciate the novelty and clarity of our work.
>
> Your question regarding the motivation behind $\alpha=2$ was common across multiple reviewers - so we have responded to it along with the others in a single comment.

---

### Author Response · Authors · 2022-11-11
**Reply to all reviewers**

Thank you for your reviews. We appreciate your valuable comments. We identified two common question topics across reviews which we would like to address together.


Firstly, multiple reviewers asked for justification for our choice of setting $\alpha=2$ for the $\alpha$-divergence, and comparisons with alternative divergences.
The reason we picked this is for its theoretical properties: It is mass covering, and is equivalent to minimizing the variance in the importance weights or, equivalently, maximizing the effective sample size.
Our goal is to use the trained flow as an importance sampler and having low variance in the importance sampling weights will generally lower the variance in the importance sampling estimates.
This is discussed in Appendix E of [*Divergence measures and message passing* by Minka 2006](https://groups.seas.harvard.edu/courses/cs281/papers/minka-divergence.pdf).

As noted by reviewers, the FAB style of approach may be applied to $alpha$-divergence with other values of $\alpha$.
We have added the derivation of the FAB loss for arbitrary values of $\alpha$ to Appendix B.
Setting $\alpha=1$ is equivalent to using a maximum likelihood objective / minimising the forward kl - which was also asked about by reviewers.

We have also added further experiments where we run the FAB algorithm with different values of $\alpha$ to see whether our theoretical claims are empirically justified.
For alanine dipeptide, we trained models with FAB using $\alpha \in \{0.25, 0.5, 1, 1.5, 3\}$ in addition to the already existing experiments with $\alpha = 2$.
Below, the results for FAB with a replay buffer using different $\alpha$-values are reported. The results are averaged over 3 runs and the standard error is given as uncertainty. Best results are emphasized in **bold**.
| $\alpha$ | ESS (\%)         | $\mathrm{E}_{p(x)} \left[ \log q (x) \right]$ | KLD | KLD w/ RW |
|----------|-----------------|-----------------------------------------------|-----|-----------|
| $0.25$ | $0.021 \pm 0.000$ | $-546 \pm 14$ | $5.36 \pm 0.72$ | $10.9 \pm 1.4$ |
| $0.5$ | $0.023 \pm 0.001$ | $-606 \pm 20$ | $4.84 \pm 0.95$ | $10.4 \pm 1.9$ |
| $1$ | $0.027 \pm 0.000$ | $60.1 \pm 9.0$ | $2.79 \pm 0.81$ | $6.73 \pm 1.76$ |
| $1.5$ | $89.4 \pm 0.3$ | $211.49 \pm 0.02$ | $(1.44 \pm 0.90)\times 10^{-2}$ | $(1.52 \pm 0.93)\times 10^{-2}$ |
| $2$ | $\mathbf{92.8} \pm \mathbf{0.1}$ | $\mathbf{211.54} \pm \mathbf{0.00}$ | $\mathbf{(3.42} \pm \mathbf{0.45)\times 10^{-3}}$ | $\mathbf{(2.51} \pm \mathbf{0.39)\times 10^{-3}}$ |
| $3$ | $\mathbf{93.9} \pm \mathbf{1.1}$ | $211.49 \pm 0.03$ | $(2.58 \pm 0.96)\times 10^{-2}$ | $(2.19 \pm 0.82)\times 10^{-2}$ |

In the following table, the results for FAB without a replay buffer using different $\alpha$-values are reported. The results are averaged over 3 runs and the standard error is given as uncertainty. Best results are emphasized in **bold**.
| $\alpha$ | ESS (\%)         | $\mathrm{E}_{p(x)} \left[ \log q (x) \right]$ | KLD | KLD w/ RW |
|----------|-----------------|-----------------------------------------------|-----|-----------|
| $0.25$ | $1.7 \pm 0.3$ | $198.71 \pm 0.09$ | $3.94 \pm 0.40$ | $8.45 \pm 0.20$ |
| $0.5$ | $9.4 \pm 4.4$ | $-192 \pm 326$ | $7.70 \pm 5.59$ | $7.99 \pm 5.49$ |
| $1$ | $15.8 \pm 4.9$ | $210.16 \pm 0.40$ | $(1.09 \pm 0.19)\times 10^{-1}$ | $(5.60 \pm 0.75)\times 10^{-2}$ |
| $1.5$ | $34.8 \pm 3.6$ | $210.87 \pm 0.07$ | $(7.72 \pm 0.72)\times 10^{-2}$ | $(4.04 \pm 0.04)\times 10^{-2}$ |
| $2$ | $\mathbf{52.2} \pm \mathbf{1.3}$ | $\mathbf{211.13} \pm \mathbf{0.03}$ | $\mathbf{(6.28} \pm \mathbf{0.33)\times10^{-2}}$ | $\mathbf{(2.66} \pm \mathbf{0.90)\times 10^{-2}}$ |
| $3$ | $24.8 \pm 4.6$ | $210.48 \pm 0.14$ | $(1.31 \pm 0.05)\times 10^{-1}$ | $(3.88 \pm 0.03)\times 10^{-2}$ |

Setting $\alpha = 2$ turns out to be the best choices regarding almost all performance metrics, while values close to $2$ still perform better than the baselines not using samples from the target distribution. We added these results and the analysis in Appendix F.2 of our manuscript.

---

> ### Author Response · Authors · 2022-11-11
> **Reply to all reviewers (cont.)**
>
> Secondly, several reviewers asked why we selected $p^2/q$ as the AIS target distribution and how it compares to alternatives.
> This choice is justified as follows:
> Given an integral $\int f(x) dx$ where $f(x) \geq 0$, the minimum variance importance sampling distribution for estimation of this is given by $g(x) \propto f(x)$.
> This is a common result in importance sampling literature, that we note in a footnote on page 3 with citations to literature that explain this link in detail.
> For the $\alpha=2$-divergence loss we have $D_{\alpha=2}(p \| q) \propto p^2/q$, which gives the minimum variance importance sampling distribution $p^2/q$.
> Thus sampling approximately from this distribution using AIS provides a lower variance estimate of the loss which is important for obtaining a good training signal for the flow.
> One alternative choice is to run AIS targeting $p$, and then to write the loss as an expectation over $p$ (similarly to if we were using samples from $p$).
> In Appendix C.2, we provide a comparison of using $p$ versus $p^2/q$ as the target for AIS on a simple 1D Gaussian, and find that $p^2/q$ is superior.
> In Appendix C.2 we also provide analysis (theoretical and empirical) comparing the estimation of $\alpha=2$ divergence directly using samples from directly from $q$ and $p$.

---

> > ### Author Response · Authors · 2022-11-16
> > **Reply to all reviewers (cont.)**
> >
> > We have now also added a study of FAB varying $\alpha \in \{0.25, 0.5, 1, 1.5, 3\}$ for the GMM problem to Appendix D.3.
> > Generally, on the GMM problem the differences in performance between different choices of $\alpha$ was less stark than for the alanine dipeptide problem, and all methods with $\alpha \geq 1$ are able to obtain a reasonably good fit for the target.
> > The reason that the differences in performance are smaller is because the problem is easier.
> > For FAB without the replay buffer, $\alpha=2$ had slightly superior performance compared to FAB with other values of $\alpha$.
> > For FAB with the buffer $\alpha=2$ and $\alpha=3$ achieve the best performance.
> > Along with the results on the alanine dipeptide problem, this provides further support for the choice of $\alpha=2$.

---

### Decision · Program_Chairs · 2023-01-20

**Decision:**

Accept: notable-top-25%

**Justification For Why Not Higher Score:**

This work may meet the bar for an oral presentation, depending on the strength of other contenders.

**Justification For Why Not Lower Score:**

I think the paper is well above the cutoff for acceptance for the reasons outlined above. Note that the reviewers who read the paper more carefully gave it higher scores, so in spirit the average score is maybe a bit above 7.

**Metareview: Summary, Strengths And Weaknesses:**

This paper presents a method for learning flow-like models to match complex multimodal distributions which are typically hard to sample from with MCMC-like methods. It augments the flows with AIS and adapts them to minimize the alpha=2 divergence (which is beneficial for reducing the variance of importance weights). Experiments show the method improves upon existing flow methods for complex multimodal distributions.

The reviewers checked this paper pretty carefully and are impressed with the novelty and clarity. The method is innovative both in learning AIS-like samplers for inference (which is usually pretty hard, despite its effectiveness for partition function estimation), in the use of 2-divergence, and for carefully considering various details around HMC. They have some minor criticisms, mostly relating to the difficulty of tuning hyperparameters and whether the 2-divergence is really better than other choices such as MLE. These aren't critical issues, and indeed are the sort of thing that's more or less inevitable for papers with significant novelty. I believe the paper should be accepted.


**Note From Pc:**

if the above contains the word "oral" or "spotlight" please see: "oral" presentation means -> notable-top-5% and "spotlight" means -> notable-top-25%. As stated in our emails, we are disassociating presentation type from AC recommendations